# DistriBlock: Identifying adversarial audio samples by leveraging characteristics of the output distribution

**Matías P. Pizarro B.**[1]  **Dorothea Kolossa**[2]  **Asja Fischer**[1]

[1]Faculty of Computer Science, Ruhr University Bochum, Germany
[2]Electronic Systems of Medical Engineering, Technische Universität Berlin, Germany

## Abstract

Adversarial attacks can mislead automatic speech recognition (ASR) systems into predicting an arbitrary target text, thus posing a clear security threat. To prevent such attacks, we propose DistriBlock, an efficient detection strategy applicable to any ASR system that predicts a probability distribution over output tokens in each time step. We measure a set of characteristics of this distribution: the median, maximum, and minimum over the output probabilities, the entropy of the distribution, as well as the Kullback-Leibler and the Jensen-Shannon divergence with respect to the distributions of the subsequent time step. Then, by leveraging the characteristics observed for both benign and adversarial data, we apply binary classifiers, including simple threshold-based classification, ensembles of such classifiers, and neural networks. Through extensive analysis across different state-of-the-art ASR systems and language data sets, we demonstrate the supreme performance of this approach, with a mean area under the receiver operating characteristic curve for distinguishing target adversarial examples against clean and noisy data of 99% and 97%, respectively. To assess the robustness of our method, we show that adaptive adversarial examples that can circumvent DistriBlock are much noisier, which makes them easier to detect through filtering and creates another avenue for preserving the system's robustness.

## 1 INTRODUCTION

Voice recognition technologies are widely used in the devices that we interact with daily—in smartphones or virtual assistants—and are also being adapted for more safety-critical tasks like self-driving cars [Wu et al., 2022] and healthcare applications. Safeguarding these systems from malicious attacks thus plays a more and more critical role, since manipulated erroneous transcriptions can potentially lead to severe security harms.

State-of-the-art automated speech recognition systems are based on deep learning [Kahn et al., 2020, Chung et al., 2021, Chen et al., 2022, Radford et al., 2023]. Unfortunately, deep neural networks (NNs) are highly vulnerable to adversarial attacks, since the inherent properties of the model make it easy to generate inputs—referred to as adversarial examples (AEs)—that are necessarily mislabeled, simply by incorporating a low-level additive perturbation [Szegedy et al., 2014, Goodfellow et al., 2015, Ilyas et al., 2019, Du et al., 2020]. A well-established method to generate AEs also applicable to ASR systems, is the Carlini & Wagner (C&W) attack [Carlini and Wagner, 2018]. It aims to minimize a perturbation $\delta$ that—when added to a benign audio signal $x$—induces the system to recognize a phrase chosen by the attacker. Moreover, attacks specifically developed for ASR systems shape the perturbations to fall below the estimated time-frequency masking threshold of human listeners, rendering $\delta$ hardly perceptible, and sometimes even *inaudible* to humans [Schönherr et al., 2019, Qin et al., 2019]. This underlines the urgent need for approaches to automatically detect AE attacks on ASR systems.

Motivated by the intuition that attacks may result in a higher prediction uncertainty displayed by the ASR system, we develop a novel detection technique that allows to efficiently distinguish adversarial from benign audio signals and can be used for any ASR system that estimates a probability distribution over tokens at each step of generating the output sequence—which is the case for the vast majority of state-of-the-art systems. More precisely, our contributions are as follows:

1. We propose DistriBlock: binary classifiers that build on characteristics of the probability distribution over tokens, which can be interpreted as a simple proxy of the prediction uncertainty of the ASR system.

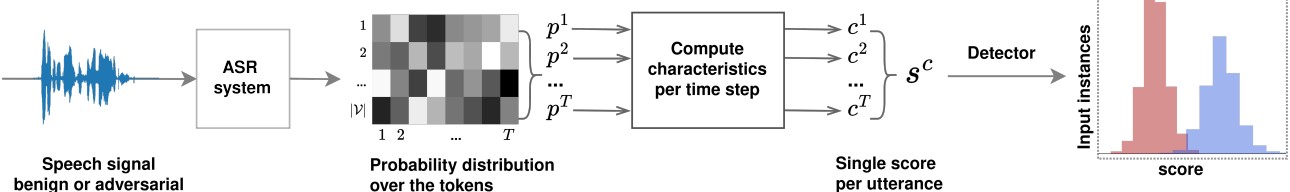

Figure 1: Proposed workflow to identify AEs: (1) compute output probability distribution characteristics per time step, (2) use a detector to tell benign data and AEs apart. $c^1$ to $c^T$ represent input characteristics, while $s^c$ denotes final scores.

2. We perform an extensive empirical analysis across diverse state-of-the-art ASR models, attack types, and datasets covering a range of languages, that demonstrate the superiority of DistriBlock over previous detection methods.

3. We propose adaptive attacks specifically designed to counteract DistriBlock, but show that the resulting adversarial examples contain a higher level of noise, which makes them easier to spot for human ears and identifiable using filtering techniques.

## 2 RELATED WORK

When it comes to mitigating the impact of adversarial attacks, there are two main research directions. On the one hand, there is a strand of research dedicated to enhancing the robustness of models. On the other hand, there is a separate research direction that focuses on designing detection mechanisms to recognize the presence of adversarial attacks.

Concerning the robustness of models, there are diverse strategies, one of which involves modifying the input data within the ASR system. This concept has been adapted from the visual to the auditory domain. Examples of input data modifications include quantization, temporal smoothing, down-sampling, low-pass filtering, slow feature analysis, and auto-encoder reformation [Meng and Chen, 2017, Guo et al., 2018, Pizarro et al., 2021]. However, these techniques become less effective once integrated into the attacker's deep learning framework [Yang et al., 2019]. Another strategy to mitigate adversarial attacks is to accept their existence and force them to be *perceivable* by humans [Eisenhofer et al., 2021], with the drawback that the AEs can continue misleading the system. Adversarial training [Madry et al., 2018], in contrast, involves employing AEs during training to enhance the NN's resiliency against adversarial attacks. Due to the impracticality of covering all potential attack classes through training, adversarial training has major limitations when applied to large and complex data sets, such as those commonly used in speech research [Zhang et al., 2019]. Additionally, this approach demands high computational costs and can result in reducing the accuracy of benign data. A recent method borrowed from the field of image recognition is adversarial purification, where generative models are employed to cleanse the input data prior to inference [Yoon et al., 2021, Nie et al., 2022]. However, only a few studies have investigated this strategy within the realm of audio. Presently, its ASR applications are confined to smaller vocabularies, and it necessitates substantial computational resources, while also resulting in decreased accuracy when applied to benign data [Wu et al., 2023].

In the context of improving the discriminative power against adversarial attacks, Rajaratnam and Kalita [2018] introduced a noise flooding (NF) detector method that quantifies the random noise needed to change the model's prediction, with smaller levels observed for AEs. Subsequently, they leverage this information to build binary classifiers. However, NF was only tested against a genetic untargeted attack [Alzantot et al., 2018] on a 10-word speech classification system. NF also needs to call the ASR system several times for making a prediction and is thus very time-consuming. A prominent non-differentiable approach uses the inherent temporal dependency (TD) in raw audio signals [Yang et al., 2019]. This strategy requires a minimal length of the audio stream for optimal performance. Unfortunately, Zhang et al. [2020] successfully evaded the detection mechanism of TD by preserving the necessary temporal correlations, leading to the generation of robust AEs once again. Däubener et al. [2020] proposed AEs detection based on uncertainty measures for hybrid ASR systems that utilize stochastic NNs. They applied their method to a limited vocabulary tailored for digit recognition. One of these uncertainty metrics—the mean-entropy—is also among those characteristics of the output distribution that we investigate, next to many others, for constructing defenses against AEs in this paper. It's worth noting that Meyer et al. [2016] utilized the averaged Kullback Leibler divergence between the output distributions of consecutive time steps (which they referred to as mean temporal distance), but in a different setting, namely to monitor the performance of ASR systems in noisy multi-channel environments. In the field of computer vision, research has explored techniques for detecting and eliminating anomalies in the input pixels that are perceptible to humans and referred to as adversarial patches. In this context, the work of Tarchoun et al. [2023] basis its detection approach on an entropy analysis of the inputs (i.e. of pixels in a certain window).

## 3 BACKGROUND

**Adversarial attacks**  Let $f(\cdot)$ be the ASR system's function, that maps an audio input to the sequence of words it most likely contains. Moreover, let $x$ be an audio signal with label transcript $y$ that is correctly predicted by the ASR, i.e., $y = f(x)$. A targeted AE can be created by finding a small perturbation $\delta$ that causes the ASR system to predict a desired transcript $\hat{y}$ given $x+\delta$, i.e., $f(x+\delta) = \hat{y} \neq y = f(x)$. This perturbation $\delta$ is usually estimated by gradient descent-based minimization of the following function

$$l(x, \delta, \hat{y}) = l_t(f(x + \delta), \hat{y}) + c \cdot l_a(x, \delta) \ , \qquad (1)$$

which includes two loss functions: (1) a task-specific loss, $l_t(\cdot)$, to find a distortion that induces the model to output the desired transcription target $\hat{y}$, and (2) an acoustic loss, $l_a(\cdot)$, that is used to minimize the energy and/or the perceptibility of the noise signal $\delta$. In the initial steps of the iterative optimization procedure, the weighting parameter $c$ is usually set to small values to first find a viable AE. Later, $c$ is often increased, in order to minimize the distortion, to render it as inconspicuous as possible.

The most prominent targeted attacks for audio are the C&W Attack and the *Imperceptible* also known as *Psychoacoustic Attack*, two well-established optimization-based adversarial algorithms. In the C&W attack [Carlini and Wagner, 2018], $l_t$ is the negative log-likelihood of the target phrase and $l_a = |\delta|_2^2$. Moreover, $|\delta|$ is constrained to be smaller than a predefined value $\epsilon$, which is decreased step-wise in an iterative process. The *Psychoacoustic Attack* [Schönherr et al., 2019, Qin et al., 2019] is divided into two stages. The first stage of the attack follows the approach outlined by C&W. The second stage of the algorithm aims to decrease the perceptibility of the noise by using frequency masking, following psychoacoustic principles. For untargeted adversarial attacks, the objective is to prevent models from predicting the correct output but no specific target transcription is given. Several untargeted attacks on ASR systems have been proposed. These include the projected gradient descent (PGD) [Madry et al., 2018], a well-known general attack method, as well as two black-box attacks—the Kenansville attack [Abdullah et al., 2020, 2021] utilizing signal processing methods to discard certain frequency components, and the genetic attack [Alzantot et al., 2018] that results from a gradient-free optimization algorithm.

**End-to-end ASR systems**  An E2E ASR system [Prabhavalkar et al., 2024] can be described as a unified ASR model that directly transcribes a speech waveform into text, as opposed to orchestrating a pipeline of separate ASR components. In general terms, the input of the system is a series of acoustic features extracted from overlapping speech frames. The model processes this series of acoustic features and predicts a probability distribution over tokens (e.g., phonemes, characters, or sub-word units) in each time step.

Subsequently, utilizing a decoding and alignment algorithm, it determines the output text. Ideally, E2E ASR models are fully differentiable and thus can be trained end-to-end by maximizing the conditional log-likelihood with respect to the desired output. Various E2E ASR models follow an encoder-only or an encoder-decoder architecture and typically are built using recurrent neural network (RNN) or transformer layers. Special care is taken of the unknown temporal alignments between the input waveform and output text, where the alignment can be modeled explicitly (e.g., CTC [Graves et al., 2006], RNN-T [Graves, 2012]), or implicitly using attention [Watanabe et al., 2017]. Furthermore, language models can be integrated to improve prediction accuracy by considering the most probable sequences [Toshniwal et al., 2018].

## 4 DETECTING ADVERSARIAL EXAMPLES WITH DISTRIBLOCK

We propose to leverage the probability distribution over the tokens from the output vocabulary to quantify uncertainty in order to identify adversarial attacks. This builds on the intuition that an adversarial input may lead to a higher uncertainty displayed by the ASR system. A schematic of our approach is displayed in Fig. 1. An audio clip—either benign or malicious—is fed to the ASR system. The system then generates probability distributions over the output tokens in each time step. The third step is to compute pertinent characteristics of these output distributions, as detailed below. Then, we use the mean, median, maximum, or minimum to aggregate the values of the characteristics into a single score per utterance. Lastly, we employ a binary classifier for differentiating adversarial instances from test data samples.

**Characteristics of the output distribution**  We assume that for each time step $t$ the ASR system produces a probability distribution $p^{(t)}$ over the tokens $i \in \mathcal{V}$ of a given output vocabulary $\mathcal{V}$. As a first characteristic, we use a common measure to quantify the total uncertainty in the predicted distribution of each time step, namely the **Shannon entropy**

$$H(p^{(t)}) = -\sum_{i=1}^{|\mathcal{V}|} p^{(t)}(i) \cdot \log p^{(t)}(i).$$

Moreover, since speech is processed sequentially from overlapping frames of the audio signal, acoustic features should transition gradually in the sequence (i.e., often the same token is predicted multiple times in a row). This should be displayed by a high similarity between probability distributions of subsequent time steps. However, the additional noise added to the speech signal of AEs potentially leads to larger changes. We, therefore, access the similarity between output distributions in two successive time steps in terms of

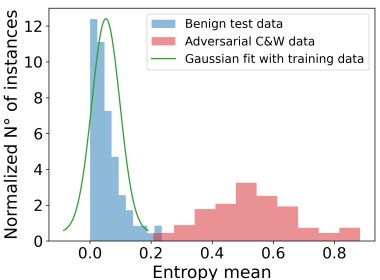
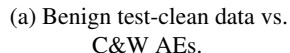
(a) Benign test-clean data vs. C&W AEs.

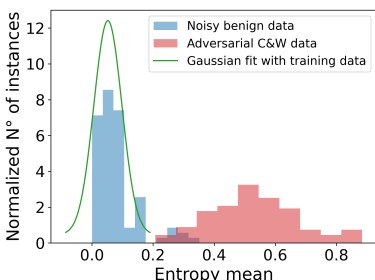
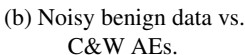
(b) Noisy benign data vs. C&W AEs.

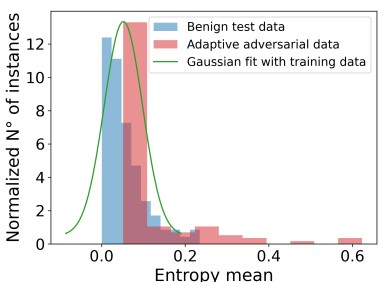
(c) Benign test-clean data vs. adaptive C&W AEs.

Figure 2: Histograms of the mean-entropy of the LSTM model's predictive distribution for 100 benign test samples vs. 100 C&W AEs.

the **Kullback–Leibler divergence (KLD)**,

$$D_{\mathrm{KL}}(p^{(t)}\|p^{(t+1)}) = \sum_{i=1}^{|\mathcal{V}|} p^{(t)}(i) \cdot \log \frac{p^{(t)}(i)}{p^{(t+1)}(i)},$$

as well as the **Jensen-Shannon divergence (JSD)**,

$$D_{\mathrm{JS}}(p^{(t)}, p^{(t+1)}) = \frac{1}{2} D_{\mathrm{KL}}(p^{(t)}\|M) + \frac{1}{2} D_{\mathrm{KL}}(p^{(t+1)}\|M),$$

with $M = \frac{1}{2}(p^{(t)} + p^{(t+1)})$, which can be seen as a symmetrized version of the KLD. In addition, we compute simple characteristics of the distributions for every $t \in \{1, \ldots, T\}$:

- the **median** of $p^{(t)}(i), i = 1, 2, \ldots, |\mathcal{V}|$.
- the **minimum** $\min_{i \in \{1,\ldots,|\mathcal{V}|\}} p^{(t)}(i)$.
- the **maximum** $\max_{i \in \{1,\ldots,|\mathcal{V}|\}} p^{(t)}(i)$.

Finally, we aggregate the step-wise values of the different characteristics into a single score, respectively, by taking the mean, median, minimum, or maximum of the values for different time steps.

**Binary classifier** The extracted characteristics of the output distribution can then be used as features for different binary classifiers. An option to obtain simple classifiers is to fit a Gaussian distribution to each score computed for the utterances from a held-out set of benign data. If the probability of a new audio sample is below a chosen threshold under the Gaussian model, this example is classified as adversarial. The threshold can be chosen on the benign examples as a value that guarantees a small number of false positives. For illustration, Fig. 2 displays example histograms of the mean-entropy values of the predictive distribution given benign and adversarial inputs, respectively. A more sophisticated approach is to employ ensemble models (EMs), in which multiple Gaussian distributions, fitted to a single score each, produce a unified decision by a majority vote. Another option is to construct an NN that takes all the characteristics described above as input.

**Adaptive attack** An adversary with complete knowledge of the defense strategy can implement so-called adaptive attacks. We implement such attacks, to challenge the robustness of DistriBlock. For this, we construct a new loss $l_k$ by adding a penalty $l_s$ to the loss function in Equation (1), weighted with some factor $\alpha$, that is

$$l_k(x, \delta, \hat{y}) = (1 - \alpha) \cdot l(x, \delta, \hat{y}) + \alpha \cdot l_s(x) . \quad (2)$$

When attacking a Gaussian classifier that is based on characteristic $c$, $l_s(x)$ corresponds to the $L_1$ norm of the difference between the mean $\overline{s}^c$ of the Gaussian fitted to the respective scores of benign data (resulting from aggregating $c$ over each utterance) and the score of $x$. When attacking an EM, $l_s$ is set to

$$l_s(x) = \sum_{i=1}^{I} |\overline{s}^{c_i} - s^{c_i}(x)| ,$$

where $c_1 \ldots c_I$ correspond to the characteristics used by the Gaussian classifiers of the ensemble is composed of. In the case of NNs, $l_s(x)$ is simply the $L_1$ norm, quantifying the difference between the NN's predicted outcome (a probability value) and one (indicating the highest probability for the benign category). We also investigated other options for choosing $l_s$, which are described in App. A.

## 5 EXPERIMENTS

This section provides information on experimental settings, assesses the quality of trained ASR systems, and evaluates both the strength of the adversarial attacks and the effectiveness of the proposed detection method.

### 5.1 EXPERIMENTAL SETTINGS

**Datasets** We used the LibriSpeech dataset [Panayotov et al., 2015] comprising approximately 1000 hours of English speech, sampled at a rate of 16 kHz, extracted from audiobooks. We further used Aishell [Bu et al., 2017], an open-source speech corpus for Mandarin Chinese. Since Chinese

Table 1: Performance of ASR systems on benign and noisy data, in terms of word and sentence error rate on 100 utterances. A cross in the LM column indicates that the ARS system integrates a language model.

| Model | Language | LM | Benign data WER | Benign data SER | Noisy data WER | Noisy data SER | Noisy data $SNR_{Seg}$ | $dB_x$ |
|-------|----------|-----|------|------|------|------|------|------|
| LSTM | Italian (It) | ✗ | 15.65% | 52% | 31.74% | 72% | -3.65 | 6.52 |
| LSTM | English (En) | ✗ | 5.37% | 31% | 8.46% | 45% | 2.75 | 6.67 |
| LSTM | English (En-LM) | ✓ | 4.23% | 24% | 5.68% | 30% | 2.75 | 6.67 |
| wav2vec | Mandarin (Ma) | ✓ | 4.37% | 28% | 8.49% | 43% | 5.25 | 4.50 |
| wav2vec | German (Ge) | ✗ | 8.65% | 33% | 16.08% | 51% | -2.66 | 7.85 |
| Trf | Mandarin (Ma) | ✗ | 4.79% | 29% | 7.40% | 40% | 5.25 | 4.50 |
| Trf | English (En) | ✓ | 3.10% | 20% | 11.87% | 44% | 2.75 | 6.67 |

Table 2: Quality of different targeted attacks. Results are averaged over 100 adversarial examples. WER and SER are measured w.r.t. the target utterance. The adaptive attack is customized for mean-median GCs.

| Model | C&W attack WER | C&W attack SER | C&W attack $SNR_{Seg}$ | C&W attack $dB_x$ | Psychoacoustic attack WER | Psychoacoustic attack SER | Psychoacoustic attack $SNR_{Seg}$ | Psychoacoustic attack $dB_x$ | Adaptive attack WER | Adaptive attack SER | Adaptive attack $SNR_{Seg}$ | Adaptive attack $dB_x$ |
|-------|------|------|------|------|------|------|------|------|------|------|------|------|
| LSTM (It) | 0.84% | 3.00% | 17.79 | 44.51 | 0.84% | 3.00% | 18.17 | 38.52 | 0.84% | 3.00% | -1.47 | 18.36 |
| LSTM (En) | 1.09% | 2.00% | 14.91 | 33.29 | 1.09% | 2.00% | 15.14 | 31.92 | 0.30% | 1.00% | 0.23 | 14.01 |
| LSTM (En-LM) | 1.19% | 2.00% | 17.50 | 36.46 | 1.19% | 2.00% | 17.82 | 33.93 | 1.19% | 2.00% | 3.81 | 17.45 |
| wav2vec (Ma) | 0.08% | 1.00% | 22.22 | 31.35 | 0.08% | 1.00% | 22.73 | 30.66 | 0.08% | 0.00% | -4.28 | 4.55 |
| wav2vec (Ge) | 0.00% | 0.00% | 20.58 | 50.86 | 0.00% | 0.00% | 21.08 | 41.46 | 0.00% | 0.00% | -12.12 | 11.23 |
| Trf (Ma) | 0.00% | 0.00% | 31.93 | 49.35 | 0.00% | 0.00% | 29.47 | 32.69 | 0.00% | 0.00% | 1.24 | 9.45 |
| Trf (En) | 0.00% | 0.00% | 27.85 | 53.54 | 0.00% | 0.00% | 25.70 | 37.68 | 0.00% | 0.00% | 1.99 | 15.52 |

is a tonal language, the speech of this corpus exhibits significant and meaningful variations in pitch. Additionally, we considerd the Common Voice (CV) corpus [Ardila et al., 2020], one of the largest multilingual open-source audio collections available in the public domain. Created through crowdsourcing, CV includes additional complexities within the recordings, such as background noise and reverberation.

**ASR systems** We analyzed a variety of fully integrated PyTorch-based state-of-the-art deep learning E2E speech engines. More specifically, we investigated three different models. The first employs a wav2vec2 encoder [Baevski et al., 2020] and a CTC decoder. The second integrates an encoder, a decoder, and an attention mechanism between them, as initially proposed with the Listen, Attend, and Spell (LAS) system [Chan et al., 2016], employing a CRDNN encoder and a LSTM for decoding [Chorowski et al., 2015]. The third model implements a transformer architecture relying on attention mechanisms for both encoding and decoding [Vaswani et al., 2017, Wolf et al., 2020]. The models are shortly referred to as wav2vec, LSTM, and Trf, respectively, in our tables. These models generate diverse output formats depending on their language and tokenizer selection, which can encode either characters or subwords. Specifically, they produce output structures with neuron counts of 32, 500, 1000, 5000, or 21128. In App. E, Tab. 18 we provide details regarding the specific tokenizer type employed by each ASR model. We trained these models on the data sets described above.

To improve generalization and make the classifiers more robust, we applied standard data augmentation techniques provided in SpeechBrain: corruption with random samples from a noise collection, removing portions of the audio, dropping frequency bands, and resampling the audio signal at a slightly different rate.

**Adversarial attacks** To generate the AEs, we utilized a repository that contains a PyTorch implementation of all considered attacks [Olivier and Raj, 2022]. For each data set, we randomly selected 200 samples that were not longer than five seconds from the test set [1]. Based on each sample, we generated adversarial examples for each adversarial attack type and ASR model. Then, all methods were tested on the task of distinguishing the first 100 test examples from the corresponding adversarial examples. So the same benign examples were used to test all different types of attacks. The additional 100 test examples and corresponding adversarial examples were used as a validation set for picking the best characteristic for the Gaussian classifiers for each model. For targeted attacks, each benign sample was assigned an adversarial target transcript sourced from the same dataset. Specifically, our selection process adhered to three guiding principles: (1) the audio file's original transcription cannot be used as the new target description, (2) there should be an equal number of tokens in both the original and target transcriptions, and (3) each audio file should receive a unique target transcription.

For the adaptive attack, 100 adaptive adversarial examples were generated starting from inputs that already mislead the system, i.e.by fine-tuning 100 adversarial examples used for testing. Fine-tuning was based on minimizing the loss function given in Equation (2) for 1000 steps of gradient descent. We kept $\alpha$ constant at a value of 0.3, while

---

[1]We reduced the audio clip length to save time and resources, as generating AEs for longer clips can take up to an hour [Carlini and Wagner, 2018], depending on the computer and model complexity. A 5-sec length was a favorable trade-off between time/resources, and the number of AEs created per model.

Table 3: Quality of different untargeted attacks. Results are averaged over 100 adversarial examples. WER and SER are measured w.r.t the predicted transcription from clean data given by the ASR system.

| Model | PGD attack | | | | Genetic attack | | | | Kenansville attack | | | |
|---|---|---|---|---|---|---|---|---|---|---|---|---|
| | WER | SER | $\text{SNR}_{Seg}$ | $\text{dB}_x$ | WER | SER | $\text{SNR}_{Seg}$ | $\text{dB}_x$ | WER | SER | $\text{SNR}_{Seg}$ | $\text{dB}_x$ |
| LSTM (It) | 121% | 100% | 7.39 | 25.76 | 41.6% | 83.0% | 3.04 | 35.13 | 73.2% | 95.0% | -6.1 | 6.32 |
| LSTM (En) | 95% | 100% | 15.13 | 25.91 | 24.5% | 85.0% | 6.49 | 33.59 | 49.8% | 85.0% | 1.32 | 7.4 |
| LSTM (En-LM) | 100% | 100% | 15.19 | 26.21 | 23.8% | 83.0% | 6.63 | 33.59 | 49.3% | 78.0% | 1.32 | 7.4 |
| wav2vec (Ma) | 90% | 100% | 20.09 | 23.68 | 36.2% | 94.0% | 6.24 | 23.74 | 62.4% | 99.0% | 6.18 | 6.33 |
| wav2vec (Ge) | 102% | 100% | 6.88 | 26.79 | 30.7% | 78.0% | 1.72 | 33.39 | 49.3% | 86.0% | -5.73 | 6.89 |
| Trf (Ma) | 126% | 100% | 19.49 | 26.41 | 44.1% | 96.0% | 4.36 | 23.74 | 73.8% | 98.0% | 6.18 | 6.33 |
| Trf (En) | 102% | 100% | 14.88 | 26.58 | 17.8% | 77.0% | 8.79 | 33.59 | 40.7% | 72.0% | 1.32 | 7.4 |

Table 4: AUROC for DistriBlock models (mean-median-GC ad NN trained on C&W AEs) compared to baselines (NF and TD) w.r.t the task of distinguishing different types of targeted AEs from clean and noisy test data.

| Model | Benign vs. C&W adversarial data | | | | Noisy vs. C&W adversarial data | | | | Benign vs. Psychoacoustic adversarial data | | | |
|---|---|---|---|---|---|---|---|---|---|---|---|---|
| | NF | TD | GC | NN | NF | TD | GC | NN | NF | TD | GC | NN |
| LSTM (It) | 0.8736 | 0.8564 | **0.9980** | 0.9955 | 0.9186 | 0.8237 | **0.9686** | 0.9103 | 0.8871 | 0.8557 | **0.9972** | 0.9943 |
| LSTM (En) | 0.8741 | 0.9695 | **0.9993** | 0.9982 | 0.9410 | 0.9694 | **0.9966** | 0.9940 | 0.8868 | 0.9697 | **0.9993** | 0.9982 |
| LSTM (En-LM) | 0.9345 | 0.9097 | 0.9508 | **0.9825** | 0.9680 | 0.8852 | 0.9454 | **0.9775** | 0.9447 | 0.9266 | 0.9557 | **0.9840** |
| wav2vec (Ma) | 0.8993 | **0.9937** | 0.9902 | 0.9852 | 0.9406 | **0.9817** | 0.9570 | 0.9427 | 0.9014 | **0.9935** | 0.9897 | 0.9853 |
| wav2vec (Ge) | 0.8725 | 0.9836 | **0.9982** | 0.9637 | 0.9372 | 0.9557 | **0.9863** | 0.9065 | 0.8749 | 0.9835 | **0.9973** | 0.9596 |
| Trf (Ma) | 0.9106 | 0.9828 | 0.9894 | **0.9990** | 0.9546 | 0.9790 | 0.9571 | **0.9798** | 0.9116 | 0.9910 | 0.9929 | **0.9974** |
| Trf (En) | 0.9100 | 0.9770 | **1.0000** | 0.9999 | 0.9728 | 0.9351 | **0.9769** | 0.9531 | 0.9098 | 0.9903 | **1.0000** | 0.9995 |
| Avg. AUROC | 0.8963 | 0.9532 | **0.9894** | 0.9892 | 0.9475 | 0.9328 | **0.9697** | 0.9520 | 0.9023 | 0.9586 | **0.9903** | 0.9883 |

$\delta$ remains unchanged during the initial 500 iterations, after which it is gradually reduced. This approach noticeably diminishes the discriminative capability of our defense across all models, but comes at the expense of generating noisy AEs. Other choices of hyperparameters resulted in less noisy AEs but also led to weaker attacks (most of the generated samples did not successfully deceive Distri-Block, see App. A.). A selection of benign, adversarial, and noisy data employed in our experiments, along with the code for our defense strategy, is available online at `https://github.com/matiuste/DistriBlock`.

**Adversarial example detectors** We construct three kinds of binary classifiers:

1. Based on the 24 scores (resulting from combing each of the 4 aggregation methods with the 6 characteristics) we obtain 24 simple Gaussian classifiers (GC) per model.

2. To construct an ensemble model, we implement a majority voting technique (i.e., the classification output that receives more than half of the votes), utilizing a total of 3, 5, 7, or 9 best-performing GCs. The choice of which GCs to incorporate is determined by evaluating the performance of each characteristic on the 100 C&W attacks (used for validation only) for each model and ranking them in descending order. The outcome of the ranking can be found in App. B.

3. The neural network architecture consists of three fully connected layers, each with 72 hidden nodes, followed by an output neuron with sigmoid activation function to generate a probability output in the range of 0 to 1. The network is trained on 80 C&W AEs and 80 test

samples using ADAM optimization [Kingma and Ba, 2015] with an initial learning rate of 0.0001 for 250 epochs.

Running the assessment with our detectors took approximately an extra 20 msec per sample, utilizing an NVIDIA A40 with a memory capacity of 48 GB, see App. C for more details.

## 5.2 QUALITY OF ASR SYSTEMS AND ADVERSARIAL ATTACKS

To assess the quality of the trained models as well as the performance of the AEs, we measured the word error rate (WER), the character error rate (CER), the sentence error rate (SER), a noise distortion metric defined by Carlini and Wagner [2018] and referred to as $\text{dB}_x$, and the Segmental Signal-to-Noise Ratio ($\text{SNR}_{Seg}$). Definitions of all metrics are available in App. D.

**Quality of ASR systems** Tab. 1 presents the performance of the ASR systems on 100 test samples. In addition, we evaluated the models on noisy audio data to determine performance in a situation that better mimics reality. To do so, we obtained noisy versions of the 100 test samples by adding noise utilizing SpeechBrain's environmental corruption function. The noise instances were randomly sampled from the Freesound section of the MUSAN corpus [Snyder et al., 2015, Ko et al., 2017], which includes room impulse responses, as well as 929 background noise recordings. The impact of noisy data on system performance is evident, resulting in a significant rise of the WER.

As a sanity check, we checked that the performances of all

Table 5: Average classification accuracies with classification thresholds guaranteeing a maximum 1% FPR (if possible) and a minimum 50% TPR. FPRs are indicated after the slash and accuracies were averaged over the tasks of distinguishing 100 C&W and Psychoacoustic AEs from 100 noisy and clean test samples. Results related to each task are given in App. H

| Model | NF | TD | GC | EM=3 | EM=5 | EM=7 | EM=9 | NN |
|---|---|---|---|---|---|---|---|---|
| LSTM (It) | 79.5% / 0.05 | 76.3% / 0.09 | **93.0% / 0.01** | 87.8% / 0.00 | 84.5% / 0.00 | 82.0% / 0.01 | 82.0% / 0.00 | 90.5% / 0.01 |
| LSTM (En) | 79.3% / 0.02 | 82.4% / 0.01 | **98.0% / 0.03** | 97.5% / 0.00 | 97.8% / 0.00 | 97.8% / 0.00 | 97.3% / 0.00 | 96.7% / 0.01 |
| LSTM (En-LM) | 86.3% / 0.00 | 81.3% / 0.06 | 81.3% / 0.01 | 88.8% / 0.02 | 90.5% / 0.01 | **95.0% / 0.01** | 91.3% / 0.02 | 92.0% / 0.01 |
| wav2vec (Ma) | 68.0% / 0.01 | **96.8% / 0.00** | 87.8% / 0.00 | 90.8% / 0.01 | 89.5% / 0.00 | 85.3% / 0.00 | 85.0% / 0.00 | 89.5% / 0.01 |
| wav2vec (Ge) | 90.5% / 0.01 | 96.0% / 0.00 | **97.3% / 0.00** | 92.3% / 0.03 | 91.8% / 0.04 | 90.0% / 0.04 | 92.3% / 0.03 | 90.1% / 0.01 |
| Trf (Ma) | **95.8% / 0.00** | 81.8% / 0.01 | 86.0% / 0.00 | 86.8% / 0.00 | 87.0% / 0.00 | 87.3% / 0.00 | 85.0% / 0.00 | 94.5% / 0.01 |
| Trf (En) | 95.8% / 0.01 | 92.8% / 0.04 | **98.0% / 0.01** | 96.8% / 0.01 | 97.3% / 0.01 | 97.3% / 0.01 | 97.0% / 0.01 | 95.0% / 0.01 |
| Avg. accuracy / FPR | 85.0% / 0.01 | 86.7% / 0.03 | 91.6% / 0.01 | 91.5% / 0.01 | 91.2% / 0.01 | 90.6% / 0.01 | 90.0% / 0.01 | **92.6% / 0.01** |

Table 6: AUROC for DistriBlock models (mean-median-GC ad NN trained on C&W AEs) compared to baselines (NF and TD) w.r.t the task of distinguishing different types of untargeted AEs from clean test data.

| | Benign vs. PGD adversarial data | | | | Benign vs. Genetic adversarial data | | | | Benign vs. Kenansville adversarial data | | | |
|---|---|---|---|---|---|---|---|---|---|---|---|---|
| Model | NF | TD | GC | NN | NF | TD | GC | NN | NF | TD | GC | NN |
| LSTM (It) | 0.7109 | 0.6516 | **0.9387** | 0.9156 | 0.5653 | 0.5283 | 0.5788 | **0.6791** | 0.8076 | 0.7331 | 0.8246 | **0.8992** |
| LSTM (En) | 0.7790 | 0.6742 | 0.9595 | **0.9643** | 0.5756 | 0.5053 | 0.4969 | **0.6814** | 0.7677 | 0.7554 | 0.8878 | **0.8972** |
| LSTM (En-LM) | 0.8745 | 0.8093 | **0.9984** | 0.9098 | 0.6879 | 0.5277 | 0.6384 | **0.7137** | 0.8026 | 0.7365 | 0.7881 | **0.8901** |
| wav2vec (Ma) | 0.8130 | 0.7178 | **0.8369** | 0.8357 | 0.6203 | 0.6009 | 0.6656 | **0.7067** | 0.8938 | 0.8539 | **0.9669** | 0.9525 |
| wav2vec (Ge) | 0.67075 | 0.74645 | **0.7785** | 0.4983 | 0.4750 | 0.5832 | **0.6482** | 0.6310 | 0.7514 | 0.7883 | **0.9273** | 0.8685 |
| Trf (Ma) | 0.8637 | 0.8592 | 0.8020 | **0.9311** | 0.6130 | 0.5854 | 0.6938 | **0.8402** | 0.9042 | 0.8964 | **0.9993** | **0.9993** |
| Trf (En) | **0.8590** | 0.7388 | 0.7499 | 0.5192 | **0.6185** | 0.5190 | 0.4555 | 0.6081 | 0.7826 | 0.7159 | 0.8942 | **0.9280** |
| Avg. AUROC | 0.7958 | 0.7425 | **0.8663** | 0.7963 | 0.5936 | 0.5500 | 0.5967 | **0.6943** | 0.8157 | 0.7828 | 0.8983 | **0.9193** |

trained models on the full test set (see Tab. 18 in the appendix) are consistent with those documented by Ravanelli et al. [2021], where you can also find detailed hyperparameter information for all these models.

**Quality of adversarial attacks** To estimate the effectiveness of the targeted adversarial attacks, we measured the word and sentence error w.r.t. the target utterances, reported in Tab. 2. We achieved nearly 100% success in generating targeted adversarial data for all attack types across all models. For the C&W attack, the lowest average distortion $dB_x$ achieved was $31.35$ dB, while the least distorted AEs had a $dB_x$ of $53.54$ dB. In a related study by Carlini and Wagner [2018], they reported a mean distortion of 31 dB over a different model. The AEs generated with the proposed adaptive adversarial attack also achieve a success rate of almost 100%. However, the AEs turned out to be much noisier, as displayed by a maximum average distortion $dB_x$ of $18.36$ dB over all models. This makes the perturbations more easily perceptible to humans.

For untargeted attacks, we measured the word and sentence error relative to the true label, i.e., the higher the WER or SER the stronger the attack. Results are presented in Tab. 3. In the case of a genetic attack, we observed a minimal effect on the WER, with the rate remaining below 50% for all models. PGD and Kenansville both restrict the magnitude of the perturbation of an AE based on a predefined factor value that we set to 25 for PGD and to 10 for Kenansville. Results for different choices are shown in App. F.

## 5.3 PERFORMANCE OF ADVERSARIAL EXAMPLE DETECTORS

**Detecting C&W and Psychoacoustic attacks** We investigated the performance of our binary classifiers constructed as described in Sec. 4 by measuring the area under the receiver operating characteristic (AUROC) curve for the task of distinguishing different targeted attacks from clean and noisy test data (where we used 100 test samples and 100 AEs in each setting). Results for the neural network trained on C&W attacks and the GC using the mean-median characteristic are presented in Tab. 4. We compare the results obtained by our classifiers with those obtained by noise flooding (NF) and temporal dependency (TD) as baseline methods. It's worth noting that the performance of TD on noisy data hasn't been analyzed before, and former investigations were limited to the English language [Yang et al., 2019]. Similarly, NF was solely tested against the untargeted genetic attack in a 10-word classification system. Our findings show that DistriBlock consistently outperforms NF and TD across all models but one, achieving an average AUROC score of 99% for clean and 97% for noisy data.

Note that the GC does not need any adversarial data and only requires estimating the characteristics on benign data. While the mean-median performs well as characteristic of the GC for all models, the performance of single models can be further pushed by picking another characteristic based on the performance of a validation set, as discussed in App. G. A ranking of the GCs using different characteristics based on the AUROCs for detecting C&W AEs is shown in App. B. For us, it was a bit surprising, that the mean-median lead

Table 7: Average classification accuracies with classification thresholds guaranteeing a maximum 1% FPR (if possible) and a minimum 50% TPR. FPRs are indicated after the slash and accuracies were averaged over the tasks of distinguishing 100 PGD, genetic and Kenansville AEs from 100 clean test samples, respectively.

| Model | NF | TD | GC | EM=3 | EM=5 | EM=7 | EM=9 | NN |
|---|---|---|---|---|---|---|---|---|
| LSTM (It) | 67.3% / 0.32 | 60.3% / 0.43 | 73.5% / 0.49 | **74.5% / 0.36** | 73.8% / 0.31 | 71.0% / 0.31 | 70.5% / 0.32 | 71.3% / 0.11 |
| LSTM (En) | 67.5% / 0.27 | 61.1% / 0.44 | 69.7% / 0.61 | 73.5% / 0.50 | 74.5% / 0.49 | **75.3% / 0.47** | 75.2% / 0.47 | 70.7% / 0.14 |
| LSTM (En-LM) | 76.7% / 0.25 | 66.5% / 0.37 | 70.8% / 0.25 | 76.3% / 0.31 | 78.2% / 0.31 | 77.3% / 0.40 | **79.7% / 0.31** | 69.3% / 0.10 |
| wav2vec (Ma) | 69.3% / 0.08 | 63.8% / 0.67 | 75.0% / 0.31 | 77.8% / 0.33 | **79.2% / 0.27** | 75.5% / 0.23 | 74.2% / 0.25 | 68.3% / 0.12 |
| wav2vec (Ge) | 54.5% / 0.71 | 63.5% / 0.69 | 65.8% / 0.66 | 65.8% / 0.60 | 65.2% / 0.62 | 66.3% / 0.56 | **66.7% / 0.58** | 62.0% / 0.25 |
| Trf (Ma) | 62.8% / 0.64 | 69.2% / 0.26 | 73.8% / 0.27 | **84.3% / 0.12** | 84.0% / 0.12 | 84.2% / 0.13 | 81.2% / 0.13 | 81.5% / 0.03 |
| Trf (En) | 55.3% / 0.80 | 60.5% / 0.75 | 57.8% / 0.84 | 59.3% / 0.80 | 59.3% / 0.81 | 59.2% / 0.82 | 59.5% / 0.81 | **60.8% / 0.27** |
| Avg. accuracy / FPR | 64.8% / 0.44 | 63.6% / 0.52 | 69.5% / 0.49 | 73.1% / 0.43 | **73.5% / 0.42** | 72.7% / 0.42 | 72.4% / 0.41 | 69.1% / 0.15 |

Table 8: Classification accuracy based on WER & CER values after LPF and SG filtering. Evaluated on 100 clean test set samples and 100 adaptive C&W AEs, using a threshold aiming for a maximum 1% FPR when feasible.

| Model | Pre-filtering | | | | | LPF+SG filtering. Results reported based on WER% and CER% values | | | | |
|---|---|---|---|---|---|---|---|---|---|---|
| | GC | EM=3 | EM=7 | EM=9 | NN | GC | EM=3 | EM=7 | EM=9 | NN |
| LSTM (It) | 50.0% | 46.0% | 43.5% | 48.5% | 49.5% | 94.0% / 97.0% | 91.5% / 95.0% | 88.5% / 90.5% | 87.0% / 88.5% | 97.5% / 99.0% |
| LSTM (En) | 51.5% | 54.5% | 65.5% | 73.0% | 65.5% | 98.0% / 96.5% | 98.0% / 96.5% | 97.0% / 97.5% | 97.0% / 97.0% | 99.5% / 100.0% |
| LSTM (En-LM) | 35.5% | 50.0% | 54.5% | 54.5% | 61.0% | 98.0% / 99.5% | 96.5% / 98.5% | 91.0% / 99.0% | 91.5% / 97.5% | 98.5% / 99.5% |
| wav2vec (Ma) | 43.5% | 62.0% | 57.5% | 69.5% | 59.0% | 92.5% / 92.5% | 93.5% / 93.5% | 90.5% / 90.5% | 93.0% / 93.0% | 94.5% / 94.5% |
| wav2vec (Ge) | 49.0% | 46.0% | 44.5% | 90.0% | 44.5% | 95.5% / 93.0% | 96.0% / 95.5% | 92.0% / 93.5% | 97.5% / 98.5% | 98.5% / 98.5% |
| Trf (Ma) | 38.0% | 41.5% | 40.5% | 38.0% | 49.5% | 75.0% / 75.0% | 78.0% / 78.0% | 73.5% / 73.5% | 71.0% / 71.0% | 89.0% / 89.0% |
| Trf (En) | 50.0% | 49.5% | 50.0% | 50.0% | 49.5% | 93.5% / 98.5% | 91.5% / 97.0% | 82.5% / 88.0% | 76.5% / 84.5% | 98.5% / 100.0% |
| Avg. accuracy | 45.4% | 49.9% | 50.9% | 60.5% | 54.1% | 92.4% / 93.1% | 92.1% / 93.4% | 87.9% / 90.4% | 87.6% / 90.0% | **96.6% / 97.2%** |

to the best results. However, the mean-entropy, which has a clear notion as uncertainty measure and was used in the context of AE detection before, is about as good. The metrics measuring the distance between distributions in successive steps (KLD and JSD) are overall less effective. Finally, we note that the neural networks perform surprisingly well at identifying Psychoacoustic AEs, although they were trained only on a small set of C&W attacks, thus showing a good transferability between attack types.

To evaluate the classification accuracy of the AE detectors, we adopted a conservative threshold selection criterion based on a validation set: we picked the threshold with the highest false positive rate (FPR) below 1% (if available) while maintaining a minimum true positive rate (TPR) of 50%. Next to GCs and NNs we now as well consider EMs build on a total of 3, 5, 7, or 9 best-performing GCs. The resulting classification accuracies (averaged over the tasks of distinguishing C&W and Psychoacoustic attacks from noisy and clean test samples, respectively) are shown in Tab. 5. DistriBlock again outperforms NF and TD on 5 of the 7 models. Detailed results can be found in App. H.

**Detecting untargeted attacks**   To assess the transferability of our detectors to untargeted attacks, we investigated the defense performance of GCs based on the mean-median characteristic and NNs trained on C&W AEs when exposed to PGD, genetic, or Kenansville attacks. Results are reported in Tab. 6. Although the detection performance decreases in comparison to targeted attacks, our methods are still way more efficient than NF and TD, with AUROCs even exceeding 90% for the Kenansville attack. While NF and TD

received (with thresholds picked as before) an average accuracy of 64,8% and 63,6% over all models and all three untargeted attacks, DistriBlock achieved with GC 69,5% and with an ensemble of 5 GCs 73,5% accuracy, as reported in Tab. 7. Detailed results can be found in App. I.

In general, AEs resulting from the genetic attack prove challenging to detect, which may be attributed to its limited impact on the output text as displayed by a relatively low WER (compare Tab. 3). Luckily, untargeted attacks are in general less threatening and all AEs we investigated are characterized by noise, making them easily noticeable by human hearing.

**Detecting adaptive adversarial attacks**   We evaluate the detection performance of our classifiers when challenged with the adaptive attack. Again, we calculated the classification accuracy based on a threshold aiming for a maximum FPR smaller than 1% (where feasible). The results shown on the left side of Tab. 8 demonstrate that the defense provided by DistriBlock can be easily mitigated if the adversary knows about the defense mechanism. However, one can leverage the fact that the adaptive attacks result in much noisier examples. To do so, we compare the predicted transcription of an input signal with the transcription of its filtered version using metrics like WER and CER. More specifically, we employed two filtering methods: a low-pass filter (LPF) eliminating high-frequency components with a 7 kHz cutoff frequency [Monson et al., 2014] and a PyTorch-based Spectral Gating (SG) [Sainburg, 2019, Sainburg et al., 2020], an audio-denoising algorithm that calculates noise thresholds for each frequency band and generates masks

Table 9: Classification accuracy based on WER & CER values after LPF and SG filtering. Evaluated on 100 noisy samples and 100 adaptive C&W AEs, using a threshold aiming for a maximum 1% FPR when feasible.

| Model | GC | EM=3 | EM=7 | EM=9 | NN |
|---|---|---|---|---|---|
| LSTM (It) | 89.5% / 90.5% | 87.0% / 88.5% | 84.0% / 84.0% | 82.5% / 82.0% | 93.0% / 92.5% |
| LSTM (En) | 96.5% / 95.0% | 96.5% / 95.0% | 95.5% / 96.0% | 95.5% / 95.5% | 98.0% / 98.5% |
| LSTM (En-LM) | 97.0% / 98.5% | 95.5% / 97.5% | 90.0% / 98.0% | 90.5% / 96.5% | 97.5% / 98.5% |
| wav2vec (Ma) | 91.5% / 91.5% | 92.5% / 92.5% | 89.5% / 89.5% | 92.0% / 92.0% | 93.5% / 93.5% |
| wav2vec (Ge) | 93.5% / 92.5% | 94.0% / 95.0% | 90.0% / 93.0% | 95.5% / 98.0% | 96.5% / 98.0% |
| Trf (Ma) | 73.0% / 73.0% | 76.0% / 76.0% | 71.5% / 71.5% | 69.0% / 69.0% | 87.0% / 87.0% |
| Trf (En) | 92.0% / 96.0% | 90.0% / 94.5% | 81.0% / 85.5% | 75.0% / 82.0% | 97.0% / 97.5% |
| Avg. accuracy | 90.4% / 91.0% | 90.2% / 91.3% | 85.9% / 88.2% | 85.7% / 87.9% | **94.6% / 95.1%** |

to suppress noise below these thresholds. We then tried to distinguish attacks from benign data based on the WER and CER resulting from comparing the transcription after filtering to that before filtering. For that we again constructed a simple Gaussian classifier, i.e., we performed threshold-based classification. The resulting accuracies are shown on the right side of Tab. 8, and demonstrate that the adaptive AEs can be efficiently detected due to their noisiness. We studied alternative ways to generate adaptive AEs, using different hyperparameters and loss functions, as outlined in App. A. In settings that lead to less noisy adaptive AEs, DistriBlock demonstrated high robustness in the detection of AEs. In summary, either DistriBlock could defend even adaptive attacks, or the attacks got that noisy, that the suggested filtering-based technique could be used for defense.

Finally, we test the robustness of our filtering methods when dealing with noisy data. We employed the same noisy data set used for testing ASR robustness, as detailed in subsection. 5.2. We then applied the WER and CER based classifiers described above to task of distinguishing noisy benign data from adaptive AEs. The resulting classification accuracies are shown in Tab. 9. Notably, the performance slightly decreases with noisy data compared to distinguishing benign data from adaptive AEs. However, the adaptive adversarial examples can still be detected with an average accuracy up to 95.1% in the case of NNs.

## 6  DISCUSSION & CONCLUSION

In this work we propose DistriBlock, a novel detection strategy for adversarial attacks on neural network-based state-of-the-art ASR systems. DistriBlock constructs simple classifiers based on features extracted from the distribution over tokens produced by the ASR models in each prediction step. We performed an extensive empirical analysis including 3 state-of-the-art neural ASR systems trained on 4 different languages, 5 prominent attack types (two targeted and one untargeted white-box attack, as well as two untargeted black-box attacks), and settings with clean and noisy benign data. Our results demonstrate that simple Gaussian classifiers based on the mean-median probability and mean-entropy of the distributions over all time steps are highly effective adversarial example detectors. Notably, their construction

is simple and only requires benign data, and the computational overhead during prediction is negligible. While the entropy is a measure of total prediction uncertainty and the results are in accordance with our intuition that attacks result in higher uncertainty, the effectiveness of the median as a decision criterion remains to be understood.

We also investigated neural network-based classifiers using a larger variety of distributional characteristics as input features. Although the neural networks were small and only trained on a tiny data set with only one kind of AEs, they generalized well to other kind of attacks. We suspect that the performance and robustness could be further increased by training larger models on bigger datasets containing different kind of AEs (maybe even resulting from adaptive attacks) and noisy data. DistriBlock clearly outperformed two existing detection methods that we used as baselines in the detection of all investigated attacks (and that to the best of our knowledge presented state of the art so far), showing an average accuracy increase of 5.89% for detection of targeted attacks and an increase of 8.67% for untargeted attacks compared to the better performing baseline.

To challenge our defense strategy, we proposed specific adaptive attacks that aim at generating AEs that are indistinguishable from benign data based on the induced distributional characteristics. These adaptive attacks can either be still detected by DistriBlock with high accuracy or are that noisy that they can be identified efficiently by comparing the difference in output transcripts before and after applying a noise-filter. In future work, it will be interesting to evaluate if the investigated characteristics of output distributions can also serve as indicators of other pertinent aspects, such as speech quality and intelligibility, which is a target for future work.

**Acknowledgements**

We thank our colleagues Michael Kamp, Jonas Ricker, Denis Lukovnikov, and Karla Pizzi, as well as our anonymous reviewers, for their valuable feedback. This work was supported by the German Academic Exchange Service - 57451854, the Chilean government (ANID/Doctorado acuerdo bilateral DAAD) - 62180013 and by the Deutsche Forschungsgemeinschaft (DFG, German Research Founda-

tion) under Germany's Excellence Strategy - EXC 2092 CASA – 390781972.

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

# DistriBlock: Identifying adversarial audio samples by leveraging characteristics of the output distribution
# (Supplementary Material)

**Matías P. Pizarro B.**[1]         **Dorothea Kolossa**[2]         **Asja Fischer**[1]

[1]Faculty of Computer Science, Ruhr University Bochum, Germany
[2]Electronic Systems of Medical Engineering, Technische Universität Berlin, Germany

This Supplementary Material contains additional experiments that support the findings presented in the main paper, these experiments are related to:

## A    ADAPTIVE ATTACK—ADDITIONAL SETTINGS

For an adaptive attack, we construct a new loss, $l_k$ explained in detail in Section 4.

$$l_k(x, \delta, \hat{y}) = (1 - \alpha) \cdot l(x, \delta, \hat{y}) + \alpha \cdot l_s(x) \ .$$

We perform 1,000 iterations on 100 randomly selected examples from the adversarial dataset, beginning with inputs that already mislead the system. We evaluated the adaptive attacks resulting from different settings:

1. We kept $\alpha$ constant at 0.3, while $\delta$ remained only unchanged during the initial 500 iterations. Afterward, $\delta$ is gradually reduced each time the perturbed signal successfully deceives the system.

2. We experimented with three fixed $\alpha$ values: 0.3, 0.6, and 0.9, while $\delta$ is gradually reduced each time the perturbed signal successfully deceives the system.

3. We increased the $\alpha$ value by 20% after each successful attack, while the $\delta$ factor remains unchanged during the initial 30 iterations.

4. We kept $\alpha$ constant at 0.3 and set the $l_s$ for attacking an EM, as defined in Section 4. We employed an EM-based on two characteristics: the median-mean and the mean-KLD.

5. We kept $\alpha$ constant at 0.3, and we redefined the $l_s$ term from the loss $l_k$ as follows:

$$l_s(x, \hat{x}) = \sum_{t=1}^{T-1} |D_{\mathrm{KL}}(p_x^{(t)} \| p_x^{(t+1)}) - D_{\mathrm{KL}}(p_{\hat{x}}^{(t)} \| p_{\hat{x}}^{(t+1)})| \ . \tag{3}$$

We assume that for each time step $t \in \{1, \dots, T\}$ the ASR system produces a probability distribution $p^{(t)}$ over the tokens $i \in \mathcal{V}$ of a given output vocabulary $\mathcal{V}$. Where $x$ represents the benign example, and $\hat{x}$ is its adversarial counterpart.

6. To minimize the statistical distance from the benign data distribution, we calculated for the same time step $t \in \{1, \dots, T\}$ the KLD between the output probability distribution $p^{(t)}$ related to the benign data $x$ and its adversarial counterpart $\hat{x}$. We kept $\alpha$ constant at 0.3, then we set the $l_s$ term to:

$$l_s(x, \hat{x}) = \sum_{t=1}^{T} |D_{\mathrm{KL}}(p_x^{(t)} \| p_{\hat{x}}^{(t)})| \ . \tag{4}$$

Results for the second setting are reported in Tab. 10. Regardless of the chosen $\alpha$ value, the second setting is unable to produce robust adversarial samples and has only a minimal effect on our proposed defense. This is due to the faster reduction of $\delta$, making it harder to generate an adaptive AE that circumvents DistriBlock. Similar outcomes are evident in the fifth and sixth settings, where the modified loss $l_s$ does not yield improvement, as illustrated in Tab. 13, and Tab. 14. In the third configuration, some models exhibit enhanced outcomes by diminishing the discriminative capability of our defense. Nevertheless, the adaptive AEs generated in this scenario are characterized by noise, as indicated in Tab. 11, with $\mathrm{SNR}_{Seg}$ values below 16 dB. The fourth setting presents a noise improvement with higher $\mathrm{SNR}_{Seg}$ values compared to prior settings, as shown in Tab. 12. However, detectors are still able to discriminate many AEs from benign data. We opt for the first setting in the main paper, as it substantially diminishes our defense's discriminative power across all models. But, this comes at the expense of generating noisy data, results are presented in Tab. 15. To mitigate this issue, we propose the use of filtering as an additional method to maintain the system's robustness. We demonstrate that adaptive AEs can be efficiently detected due to their noisiness, as described in subsection 5.3. In general, we observe that using an $\alpha$ value above 0.3 increases the difficulty of generating adaptive adversarial examples.

Table 10: Average SNR of 100 adaptive C&W AEs generated with different $\alpha$ values and AUROC with respect to these AEs and 100 test samples. $^{(*)}$ denotes best-performing score-characteristic.

| Model | Score-Characteristic$^{(*)}$ | $\alpha = 0.3$ | | $\alpha = 0.6$ | | $\alpha = 0.9$ | |
|---|---|---|---|---|---|---|---|
| | | $\mathrm{SNR}_{Seg}$ | GC AUROC | $\mathrm{SNR}_{Seg}$ | GC AUROC | $\mathrm{SNR}_{Seg}$ | GC AUROC |
| LSTM (It) | Mean-Median | 17.5 | 0.9635 | 17.13 | 0.8882 | 17.4 | 0.964 |
| LSTM (En) | Mean-Median | 14.78 | 0.9875 | 14.78 | 0.9741 | 14.75 | 0.9735 |
| LSTM (En-LM) | Max-Max | 17.42 | 0.9869 | 17.39 | 0.9762 | 17.37 | 0.9567 |
| wav2vec (Ma) | Mean-Entropy | 22.18 | 0.9912 | 22.16 | 0.9849 | 22.13 | 0.9779 |
| wav2vec (Ge) | Max-Min | 20.27 | 0.9774 | 19.82 | 0.8516 | 20.55 | 0.9803 |
| Trf (Ma) | Median-Max | 31.69 | 0.9893 | 31.74 | 0.9891 | 31.8 | 0.977 |
| Trf (En) | Max-Median | 27.54 | 1.000 | 27.61 | 1.000 | 27.65 | 1.000 |

Table 11: Average SNR of 100 adaptive C&W AEs generated with adapted $\alpha$ value and AUROC with respect to these AEs and 100 test samples. $^{(*)}$ denotes best-performing score-characteristic.

| Model | Score-Characteristic$^{(*)}$ | WER/CER | SER | $\mathrm{SNR}_{Seg}$ | $\mathrm{dB_x}$ | GC AUROC |
|---|---|---|---|---|---|---|
| LSTM (It) | Mean-Median | 0.84% | 3.00% | 6.64 | 27.29 | 0.4656 |
| LSTM (En) | Mean-Median | 0.20% | 1.00% | 9.51 | 24.2 | 0.5333 |
| LSTM (En-LM) | Max-Max | 0.40% | 1.00% | 12.91 | 29.77 | 0.7437 |
| wav2vec (Ma) | Mean-Entropy | 0.08% | 1.00% | 13.76 | 21.34 | 0.9146 |
| wav2vec (Ge) | Max-Min | 0.00% | 0.00% | 12.21 | 33.7 | 0.6666 |
| Trf (Ma) | Median-Max | 0.00% | 0.00% | 14.48 | 24.71 | 0.7857 |
| Trf (En) | Max-Median | 0.00% | 0.00% | 15.91 | 35.49 | 0.9835 |

Table 12: Average SNR of 100 adaptive C&W AEs generated with constant $\alpha = 0.3$ and $l_s$ aiming to attack an ensemble of GCs with mean-median and mean-KLD characteristics, and AUROC with respect to these AEs and 100 test samples.

| Model | WER/CER | SER | $SNR_{Seg}$ | $dB_x$ | GC AUROC |
|---|---|---|---|---|---|
| LSTM (It) | 0.84% | 3.00% | 16 | 38.77 | 0.8014 |
| LSTM (En) | 1.09% | 2.00% | 14.08 | 30.29 | 0.8509 |
| LSTM (En-LM) | 1.19% | 2.00% | 17.25 | 35.37 | 0.8701 |
| wav2vec (Ma) | 0.08% | 1.00% | 21.91 | 29.83 | 0.523 |
| wav2vec (Ge) | 0.00% | 0.00% | 17.61 | 41.65 | 0.4437 |
| Trf (Ma) | 0.00% | 0.00% | 27.58 | 35.52 | 0.5286 |
| Trf (En) | 0.00% | 0.00% | 22.84 | 39.82 | 0.7631 |

Table 13: Average SNR of 100 adaptive C&W AEs generated with constant $\alpha = 0.3$ and $l_s$ as defined in Equation (3), and AUROC with respect to these AEs and 100 test samples.

| Model | WER/CER | SER | $SNR_{Seg}$ | $dB_x$ | GC AUROC |
|---|---|---|---|---|---|
| LSTM (It) | 0.84% | 3.00% | 17.73 | 44.37 | 0.9976 |
| LSTM (En) | 1.09% | 2.00% | 14.91 | 33.29 | 0.9993 |
| LSTM (En-LM) | 1.19% | 2.00% | 17.5 | 36.45 | 0.957 |
| wav2vec (Ma) | 0.08% | 1.00% | 22.22 | 31.35 | 0.9904 |
| wav2vec (Ge) | 0.00% | 0.00% | 20.58 | 50.86 | 0.9982 |
| Trf (Ma) | 0.00% | 0.00% | 30.41 | 42.81 | 0.9921 |

Table 14: Average SNR of 100 adaptive C&W AEs generated with constant $\alpha = 0.3$ and $l_s$ as defined in Equation (4), and AUROC with respect to these AEs and 100 test samples.

| Model | WER/CER | SER | $SNR_{Seg}$ | $dB_x$ | GC AUROC |
|---|---|---|---|---|---|
| LSTM (It) | 0.84% | 3.00% | 17.76 | 44.5 | 0.9978 |
| LSTM (En) | 1.09% | 2.00% | 14.91 | 33.29 | 0.9993 |
| LSTM (En-LM) | 1.19% | 2.00% | 17.5 | 36.46 | 0.9551 |
| wav2vec (Ma) | 0.08% | 1.00% | 22.22 | 31.35 | 0.9902 |
| wav2vec (Ge) | 0.00% | 0.00% | 20.58 | 50.86 | 0.9982 |
| Trf (Ma) | 0.00% | 0.00% | 28.29 | 35.09 | 0.7562 |
| Trf (En) | 0.00% | 0.00% | 24.78 | 43.55 | 0.995 |

Table 15: Average SNR of 100 adaptive C&W AEs generated with constant $\alpha = 0.3$ and keeping $\delta$ unchanged during the initial 500 iterations, and AUROC with respect to these AEs and 100 test samples. $^{(*)}$ denotes best-performing score-characteristic.

| Model | Score-Characteristic$^{(*)}$ | WER/CER | SER | $SNR_{Seg}$ | $dB_x$ | GC AUROC |
|---|---|---|---|---|---|---|
| LSTM (It) | Mean-Median | 0.84% | 3.00% | -1.47 | 18.36 | 0.335 |
| LSTM (En) | Mean-Median | 0.30% | 1.00% | 0.23 | 14.01 | 0.295 |
| LSTM (En-LM) | Max-Max | 0.40% | 1.00% | 3.18 | 16.82 | 0.425 |
| wav2vec (Ma) | Mean-Entropy | 0.08% | 1.00% | -4.30 | 4.09 | 0.375 |
| wav2vec (Ge) | Max-Min | 0.00% | 0.00% | -12.96 | 10.88 | 0.255 |
| Trf (Ma) | Median-Max | 0.00% | 0.00% | -1.09 | 8.01 | 0.285 |
| Trf (En) | Max-Median | 0.00% | 0.00% | -0.19 | 14.69 | 0.250 |

# B CHARACTERISTIC RANKING

For the GCs, we determine the best-performing characteristics by ranking them according to the average AUROC on a validation set across all models. This ranking, which is shown in Tab. 16, determines the choice of characteristics to utilize for the EMs, where we implement a majority voting technique, using a total of 3, 5, 7, or 9 GCs.

Table 16: Ranking of best-performing characteristic. AUROC with respect to 100 benign data and 100 C&W AEs. $^{()}$ indicates the characteristic chosen for a specific EM.

| Score-Characteristic | Benign data vs. C&W AEs | Score-Characteristic | Benign data vs. C&W AEs |
|---|---|---|---|
| **Mean-Median**$^{(3,5,7,9)}$ | **0.9872** | Mean-KLD | 0.9162 |
| **Mean-Entropy**$^{(3,5,7,9)}$ | **0.9871** | Max-JSD | 0.8480 |
| **Max-Entropy**$^{(3,5,7,9)}$ | **0.9808** | Max-KLD | 0.8319 |
| **Median-Entropy**$^{(5,7,9)}$ | **0.9796** | Min-Median | 0.8242 |
| **Max-Median**$^{(5,7,9)}$ | **0.9759** | Max-Max | 0.7764 |
| **Median-Max**$^{(7,9)}$ | **0.9733** | Min-Min | 0.7751 |
| **Mean-Max**$^{(7,9)}$ | **0.9617** | Min-Entropy | 0.7703 |
| **Mean-Min**$^{(9)}$ | **0.9541** | Min-KLD | 0.7066 |
| **Min-Max**$^{(9)}$ | **0.9523** | Mean-JSD | 0.6717 |
| Median-Median | 0.9488 | Median-KLD | 0.6706 |
| Max-Min | 0.9365 | Min-JSD | 0.6669 |
| Median-Min | 0.9339 | Median-JSD | 0.6368 |

# C   COMPUTATIONAL OVERHEAD

The computational overhead assessment involves measuring the overall duration the system requires to predict 100 audio clips, utilizing an NVIDIA A40 with a memory capacity of 48 GB. As a result, running the assessment with our NN detectors took approximately an extra 20 msec per sample, therefore the proposed method is suitable for real-time usage. Results are reported in Tab. 17.

Table 17: Computational overhead to predict 100 audio clips measured in seconds.

| Model | Elapsed time | With NN detector | Overhead | Avg. time/sample |
|---|---|---|---|---|
| LSTM (It) | 53.005 | 53.452 | 0.447 | 0.004 |
| LSTM (En) | 55.742 | 58.038 | 2.295 | 0.023 |
| LSTM (En-LM) | 46.358 | 48.252 | 1.894 | 0.019 |
| wav2vec (Ma) | 13.339 | 14.772 | 1.432 | 0.014 |
| wav2vec (Ge) | 13.736 | 14.992 | 1.255 | 0.013 |
| Trf (Ma) | 32.070 | 34.656 | 2.586 | 0.026 |
| Trf (En) | 63.460 | 66.671 | 3.211 | 0.032 |
| Avg. across all models | 39.67 | 41.55 | 1.87 | 0.02 |

# D   PERFORMANCE INDICATORS OF ASR SYSTEMS

We used the following, standard performance indicators:

**WER**   The word error rate, is given by

$$\text{WER} = 100 \cdot \frac{S + D + I}{N} \quad ,$$

where $S$, $D$, and $I$ are the number of words that were substituted, deleted, and inserted, respectively, and $N$ is the reference text's total word count. When evaluating ARS systems, the reference text corresponds to the label utterance of the test sample, and when evaluating adversarial attacks it corresponds to the malicious target transcription of the AE. We aim for ASR models with low WER on the original data, i.e., modes that recognize the ground-truth transcript with high accuracy. From the attacker's standpoint, the aim of targeted attacks is to minimize the WER as well, but relative to the target transcription. In untargeted attacks, the objective is to have a model with a high WER w.r.t. the ground-truth text.

**CER**   The character error rate, is calculated like the WER, with the difference that instead of counting word errors, it counts character errors, with N representing the total character count of the reference text.

**SER**   The sentence error rate is defined as

$$\text{SER} = 100 \cdot \frac{N_E}{N_T} \quad ,$$

where $N_E$ is the number of audio clips that have at least one transcription error, and $N_T$ is the total number of examples. Again $N_T$ may correspond to either the number of samples in the test set or the number of adversarial examples.

**dB$_x$**   Carlini and Wagner [2018] quantified the relative loudness of an audio signal $x$ as

$$dB(x) = \max_{t \in \{0,\dots,|x|-1\}} 20 \cdot \log_{10} x(t).$$

Then, the level of distortion given by a perturbation $\delta$ to an audio signal $x$ is defined as

$$dB_x(\delta) = dB(x) - dB(\delta),$$

where a higher dB$_x$ indicates a lower level of added noise and where we assume equal lengths of $x$ and $\delta$.

**SNR$_{Seg}$**   The Segmental Signal-to-Noise Ratio measures the noise energy in Decibels and considers the entire audio signal. To obtain it, the energy ratios are computed segment by segment, which better reflects human perception than the non-segmental version [Mermelstein, 1979]. The results are then averaged:

$$\text{SNR}_{\text{Seg}} = \frac{10}{M} \cdot \sum_{m=0}^{M-1} \log_{10} \frac{\sum_{t=mF}^{mF+F-1} x(t)^2}{\sum_{t=mF}^{mF+F-1} \delta(t)^2} \, ,$$

where $M$ is the number of frames in a signal and $F$ is the frame length, $x$ represents the clean audio signal and $\delta$ the perturbation. Thus, a higher SNR$_{Seg}$ indicates less additional noise. We assume equal lengths of $x$ and $\delta$.

# E   PERFORMANCE OF ASR SYSTEMS

Tab. 18 presents the performance of the different models on the indicated datasets. The results align with those reported by Ravanelli et al. [2021], where detailed hyperparameter information for (the training of) all models can be found. Moreover, we provide the links to the detailed descriptions of each ASR system in the SpeechBrain repository.

Table 18: Performance of the ASR systems on benign data, in terms of word and sentence error rate, on the full test sets.

| Model-Language | Data | Pre-trained model | # Utterances | Subword-base Tokenizer | # of tokens | WER | SER |
|---|---|---|---|---|---|---|---|
| LSTM-Italian [1] | CV-Corpus | ✗ | 12,444 | Unigram: Converts words into subwords | 500 | 17.78% | 69.68% |
| LSTM-English | Librispeech | ✗ | 2,620 | Unigram: Converts words into subwords | 1,000 | 4.24% | 42.44% |
| LSTM-English [2] | Librispeech | RNN Language Model [3] | 2,620 | Unigram: Converts words into subwords | 5,000 | 2.91% | 32.06% |
| wav2vec-Mandarin [4] | Aishell | Large Fairseq Chinese wav2vec2 [5] | 7,176 | WordPiece: Converts words into chars | 21,128 | 5.05% | 39.30% |
| wav2vec-German [6] | CV-Corpus | Large Fairseq German wav2vec2 [7] | 15,415 | Char: Converts words into chars | 32 | 10.31% | 46.56% |
| Trf-Mandarin [8] | Aishell | ✗ | 7,176 | Unigram: Converts words into subwords | 5,000 | 6.23% | 42.35% |
| Trf-English [9] | Librispeech | Transformer Language Model [10] | 2,620 | Unigram: Converts words into subwords | 5,000 | 2.21% | 26.03% |

---

[1]CRDNN with CTC/Attention trained on CommonVoice Italian, see code:
`https://huggingface.co/speechbrain/asr-crdnn-commonvoice-it`
[2]CRDNN with CTC/Attention trained on LibriSpeech , see code:
`https://huggingface.co/speechbrain/asr-crdnn-rnnlm-librispeech`
[3]RNN language model, see code:
`https://speechbrain.readthedocs.io/en/latest/API/speechbrain.lobes.models.RNNLM.html`
[4]wav2vec 2.0 with CTC trained on Aishell, see code:
`https://huggingface.co/speechbrain/asr-wav2vec2-ctc-aishell`
[5]Chinese-wav2vec2-large-fairseq model, see code:
`https://huggingface.co/TencentGameMate/chinese-wav2vec2-large`
[6]wav2vec 2.0 with CTC trained on CommonVoice German, see code:
`https://huggingface.co/speechbrain/asr-wav2vec2-commonvoice-de`
[7]German-wav2vec2-large-fairseq model, see code:
`https://huggingface.co/jonatasgrosman/wav2vec2-large-xlsr-53-german`
[8]Transformer trained on Aishell, see code:
`https://huggingface.co/speechbrain/asr-transformer-aishell`
[9]Transformer trained on LibriSpeech, see code:
`https://huggingface.co/speechbrain/asr-transformer-transformerlm-librispeech`
[10]Transformer language model, see code:
`https://speechbrain.readthedocs.io/en/latest/API/speechbrain.lobes.models.transformer.TransformerLM.html`

# F   UNTARGETED ATTACKS

To expand the range of adversarial attacks, we explore three untargeted attacks: PGD, genetic, and Kenansville. The primary objective is to achieve a high WER w.r.t the ground-truth transcription. Each adversarial attack type is evaluated under distinct settings. Regarding PGD, the perturbation $\delta$ is limited to a predefined value $\epsilon$, calculated as $\epsilon = ||x||_2 / 10^{\frac{SNR}{20}}$. We experimented with SNR values of 10 and 25. For Kenansville, the perturbation $\delta$ is controlled by removing frequencies that have a magnitude below a certain threshold $\theta$, determined by scaling the power of a signal with an SNR factor given by $10^{\frac{-SNR}{10}}$. Subsequently, all frequencies that have a cumulative power spectral density smaller than $\theta$ are set to zero, and the reconstructed signal is formed using the remaining frequencies. Our Kenansville experiments involve a factor of 10, 15, and 25. Similar to PGD and Kenansville, the smaller perturbation is associated with higher SNR values. In genetic attack, the settings vary based on the number of iterations, we experimented with 1,000 and 2,000 iterations. The outcomes are detailed in Tab. 19 and Tab. 20 respectively.

In the main paper, in the case of PGD, we choose a factor of 25, as it induces a WER exceeding 50% across all models and maintains a higher segmental SNR. As for Kenansville, we opt for a factor of 10, as it is the only setting that demonstrates a genuine threat to the system by yielding a higher WER. In the context of a genetic attack, we choose 1,000 iterations, however adjusting the number of iterations doesn't result in a significant difference.

Table 19: Performance of PGD and genetic attacks. Results are averaged over 100 adversarial examples. WER and SER are measured w.r.t the ground truth transcription.

| Model | PGD: Factor values of 10 − 25 | | | | Genetic: Number of iterations 1,000 - 2,000 | | | |
|---|---|---|---|---|---|---|---|---|
| | WER | SER | $SNR_{Seg}$ | $dB_x$ | WER | SER | $SNR_{Seg}$ | $dB_x$ |
| LSTM (It) | 119% - 121% | 100% - 100% | -7.64 - 7.39 | 11.38 - 25.76 | 39.9% - 41.6% | 81% - 83% | 3.67 - 3.04 | 35.13 - 35.13 |
| LSTM (En) | 107% - 95% | 100% - 100% | -0.12 - 15.13 | 12.62 - 25.91 | 22.9% - 24.5% | 86% - 85% | 6.58 - 6.49 | 33.59 - 33.59 |
| LSTM (En-LM) | 108% - 100% | 100% - 100% | 0.01 - 15.19 | 12.59 - 26.21 | 20.7% - 23.8% | 79% - 83% | 7.00 - 6.63 | 33.59 - 33.59 |
| wav2vec (Ma) | 120% - 100% | 100% - 100% | 4.56 - 20.09 | 10.38 - 23.68 | 36.6% - 36.2% | 94% - 94% | 6.21 - 6.24 | 23.74 - 23.74 |
| wav2vec (Ge) | 118% - 102% | 100% - 100% | -8.28 - 6.88 | 12.45 - 26.79 | 28.4% - 30.7% | 76% - 78% | 2.44 - 1.72 | 33.39 - 33.39 |
| Trf (Ma) | 128% - 126% | 100% - 100% | 4.35 - 19.49 | 12.15 - 26.41 | 44.1% - 44.1% | 96% - 96% | 4.36 - 4.36 | 23.74 - 23.74 |
| Trf (En) | 109% - 102% | 100% - 100% | -0.49 - 14.88 | 13.10 - 26.58 | 15.9% - 17.8% | 74% - 77% | 9.16 - 8.79 | 33.59 - 33.59 |

Table 20: Performance of Kenansville attack. Results are averaged over 100 adversarial examples. WER and SER are measured w.r.t the ground truth transcription

| Model | Kenansville: Factor values of 10 − 15 − 25 | | | |
|---|---|---|---|---|
| | WER | SER | $SNR_{Seg}$ | $dB_x$ |
| LSTM (It) | 73.19% - 46.38% - 21.01% | 95.00% - 83.00% - 63.00% | -6.10 / -1.37 / 7.94 | 6.32 / 10.58 / 20.48 |
| LSTM (En) | 49.85% - 20.95% - 7.33% | 85.00% - 65.00% - 42.00% | 1.32 / 6.06 / 15.38 | 7.40 / 12.42 / 23.28 |
| LSTM (En-LM) | 49.33% - 19.92% - 5.26% | 78.00% - 56.00% - 29.00% | 1.32 / 6.06 / 15.38 | 7.40 / 12.42 / 23.28 |
| wav2vec (Ma) | 62.41% - 22.04% - 5.13% | 99.00% - 69.00% - 35.00% | 6.18 / 11.12 / 20.43 | 6.33 / 10.80 / 21.28 |
| wav2vec (Ge) | 49.32% - 26.71% - 10.62% | 86.00% - 67.00% - 35.00% | -5.73 / -1.22 / 7.83 | 6.89 / 11.93 / 22.83 |
| Trf (Ma) | 73.84% - 43.99% - 8.49% | 98.00% - 88.00% - 38.00% | 6.18 / 11.12 / 20.43 | 6.33 / 10.80 / 21.28 |
| Trf (En) | 40.66% - 13.73% - 4.02% | 72.00% - 46.00% - 24.00% | 1.32 / 6.06 / 15.38 | 7.40 / 12.42 / 23.28 |

# G   PERFORMANCE OF GAUSSIAN CLASSIFIERS

We fit 24 Gaussian classifiers for each model based on 24 characteristics. We then evaluate the performance of these GCs by measuring the AUROC and the area under the precision-recall curve (AUPRC) for the tasks of distinguishing AEs from clean and noisy data. For the calculation we used 100 samples from the clean test dataset, 100 C&W AEs, and 100 Psychoacoustic AEs. The results are presented in the following tables: Tab. 21 corresponds to the LSTM (It) model, Tab. 22 corresponds to the LSTM (En) model, Tab. 23 corresponds to the LSTM (En-LM) model, Tab. 24 corresponds to the wav2vec (Ma) model, Tab. 25 corresponds to the wav2vec (Ge) model, Tab. 26 corresponds to the Trf (Ma) model, and Tab. 27 corresponds to the Trf (En) model. The best AUROC values are shown in bold, as well as the top-performing score characteristic for detecting the C&W attack.

Table 21: Comparing GCs on clean and noisy data for the LSTM (It) model, assessing AUROC/AUPRC with 100 samples each from clean test data, C&W AEs, and Psychoacoustic AEs.

| Score-Characteristic | C&W attack | | Psychoacoustic attack | |
|---|---|---|---|---|
| | Noisy data vs. AEs | Benign data vs. AEs | Noisy data vs. AEs | Benign data vs. AEs |
| Mean-Entropy | 0.9360 / 0.9542 | 0.9812 / 0.9850 | 0.9301 / 0.9461 | 0.9769 / 0.9800 |
| Max-Entropy | 0.8122 / 0.8577 | 0.9571 / 0.9580 | 0.8072 / 0.8507 | 0.9513 / 0.9490 |
| Min-Entropy | 0.6496 / 0.5446 | 0.6602 / 0.5476 | 0.7240 / 0.6108 | 0.7372 / 0.6216 |
| Median-Entropy | 0.9092 / 0.8770 | 0.9276 / 0.8847 | 0.9011 / 0.8769 | 0.9196 / 0.8762 |
| Mean-Max | 0.9288 / 0.9414 | 0.9628 / 0.9709 | 0.9214 / 0.9244 | 0.9556 / 0.9535 |
| Max-Max | 0.7395 / 0.6594 | 0.7426 / 0.6606 | 0.7436 / 0.6637 | 0.7482 / 0.6659 |
| Min-Max | 0.8064 / 0.8377 | 0.9081 / 0.9077 | 0.7998 / 0.8285 | 0.8984 / 0.8924 |
| Median-Max | 0.8930 / 0.8459 | 0.9027 / 0.8259 | 0.8803 / 0.8254 | 0.8911 / 0.8168 |
| Mean-Min | 0.9457 / 0.9602 | 0.9879 / 0.9895 | 0.9460 / 0.9611 | **0.9887 / 0.9903** |
| Max-Min | 0.8090 / 0.8615 | 0.9706 / 0.9722 | 0.8084 / 0.8629 | 0.9720 / 0.9736 |
| Min-Min | 0.7582 / 0.7201 | 0.7492 / 0.6992 | 0.7675 / 0.7228 | 0.7565 / 0.7016 |
| Median-Min | 0.9285 / 0.9522 | 0.9855 / 0.9883 | 0.9292 / 0.9533 | 0.9860 / 0.9888 |
| **Mean-Median** | **0.9591 / 0.9681** | **0.9887 / 0.9900** | **0.9592 / 0.9687** | 0.9887 / 0.9902 |
| Max-Median | 0.8401 / 0.8829 | 0.9621 / 0.9698 | 0.8332 / 0.8782 | 0.9602 / 0.9687 |
| Min-Median | 0.8240 / 0.7814 | 0.8003 / 0.7435 | 0.8050 / 0.7384 | 0.7883 / 0.7163 |
| Median-Median | 0.9387 / 0.9573 | 0.9805 / 0.9851 | 0.9395 / 0.9585 | 0.9815 / 0.9860 |
| Mean-JSD | 0.4435 / 0.4813 | 0.4917 / 0.5387 | 0.4334 / 0.4666 | 0.4806 / 0.5375 |
| Max-JSD | 0.8556 / 0.8674 | 0.9479 / 0.9238 | 0.8433 / 0.8218 | 0.9337 / 0.8759 |
| Min-JSD | 0.5727 / 0.4897 | 0.5589 / 0.4916 | 0.5554 / 0.4803 | 0.5405 / 0.4818 |
| Median-JSD | 0.4651 / 0.4471 | 0.4316 / 0.4221 | 0.4630 / 0.4399 | 0.4305 / 0.4229 |
| Mean-KLD | 0.6444 / 0.6914 | 0.7538 / 0.7860 | 0.6472 / 0.6994 | 0.7550 / 0.7886 |
| Max-KLD | 0.6996 / 0.6959 | 0.7928 / 0.7890 | 0.7062 / 0.7210 | 0.7967 / 0.7953 |
| Min-KLD | 0.6573 / 0.5420 | 0.6487 / 0.5439 | 0.6419 / 0.5316 | 0.6302 / 0.5306 |
| Median-KLD | 0.4087 / 0.4123 | 0.4029 / 0.4131 | 0.4152 / 0.4145 | 0.4065 / 0.4147 |

Table 22: Comparing GCs on clean and noisy data for the LSTM (En) model, assessing AUROC/AUPRC with 100 samples each from clean test data, C&W AEs, and Psychoacoustic AEs.

| Score-Characteristic | C&W attack | | Psychoacoustic attack | |
|---|---|---|---|---|
| | Noisy data vs. AEs | Benign data vs. AEs | Noisy data vs. AEs | Benign data vs. AEs |
| Mean-Entropy | 0.9946 / 0.9943 | 0.9979 / 0.9979 | 0.9951 / 0.9949 | 0.9981 / 0.9981 |
| Max-Entropy | 0.9832 / 0.9833 | 0.9945 / 0.9939 | 0.9847 / 0.9855 | 0.9960 / 0.9959 |
| Min-Entropy | 0.6962 / 0.6150 | 0.6792 / 0.6143 | 0.6884 / 0.6119 | 0.6686 / 0.6079 |
| Median-Entropy | 0.9928 / 0.9938 | 0.9909 / 0.9946 | 0.9945 / 0.9954 | 0.9913 / 0.9950 |
| Mean-Max | 0.9819 / 0.9559 | 0.9854 / 0.9625 | 0.9824 / 0.9706 | 0.9863 / 0.9744 |
| Max-Max | 0.6823 / 0.6131 | 0.6855 / 0.6175 | 0.6847 / 0.6143 | 0.6881 / 0.6184 |
| Min-Max | 0.9618 / 0.9551 | 0.9805 / 0.9656 | 0.9623 / 0.9497 | 0.9808 / 0.9577 |
| Median-Max | 0.9920 / 0.9930 | 0.9900 / 0.9937 | 0.9933 / 0.9943 | 0.9907 / 0.9943 |
| Mean-Min | 0.9743 / 0.9783 | 0.9897 / 0.9902 | 0.9758 / 0.9790 | 0.9897 / 0.9899 |
| Max-Min | 0.9252 / 0.9311 | 0.9471 / 0.9508 | 0.9255 / 0.9304 | 0.9488 / 0.9521 |
| Min-Min | 0.5512 / 0.5324 | 0.5895 / 0.5646 | 0.5652 / 0.5470 | 0.5994 / 0.5726 |
| Median-Min | 0.9737 / 0.9790 | 0.9910 / 0.9919 | 0.9778 / 0.9825 | 0.9935 / 0.9940 |
| **Mean-Median** | **0.9981 / 0.9982** | **0.9998 / 0.9998** | **0.9982 / 0.9983** | **0.9996 / 0.9996** |
| Max-Median | 0.9883 / 0.9903 | 0.9994 / 0.9994 | 0.9879 / 0.9889 | 0.9992 / 0.9992 |
| Min-Median | 0.6964 / 0.7068 | 0.7100 / 0.7143 | 0.6719 / 0.6884 | 0.6854 / 0.6941 |
| Median-Median | 0.9928 / 0.9937 | 0.9981 / 0.9980 | 0.9920 / 0.9929 | 0.9974 / 0.9971 |
| Mean-JSD | 0.9306 / 0.9407 | 0.9312 / 0.9435 | 0.9305 / 0.9408 | 0.9314 / 0.9440 |
| Max-JSD | 0.9792 / 0.9784 | 0.9872 / 0.9808 | 0.9793 / 0.9784 | 0.9871 / 0.9807 |
| Min-JSD | 0.8813 / 0.7833 | 0.8855 / 0.8071 | 0.8715 / 0.7847 | 0.8775 / 0.8133 |
| Median-JSD | 0.2499 / 0.4439 | 0.2407 / 0.4258 | 0.2511 / 0.4472 | 0.2414 / 0.4302 |
| Mean-KLD | 0.9925 / 0.9943 | 0.9803 / 0.9887 | 0.9926 / 0.9943 | 0.9803 / 0.9887 |
| Max-KLD | 0.8043 / 0.7857 | 0.7832 / 0.7830 | 0.7790 / 0.7212 | 0.7580 / 0.7218 |
| Min-KLD | 0.8651 / 0.7926 | 0.8598 / 0.7856 | 0.8619 / 0.7783 | 0.8537 / 0.7693 |
| Median-KLD | 0.2970 / 0.4802 | 0.2591 / 0.4639 | 0.2960 / 0.4806 | 0.2571 / 0.4643 |

Table 23: Comparing GCs on clean and noisy data for the LSTM (En-LM) model, assessing AUROC/AUPRC with 100 samples each from clean test data, C&W AEs, and Psychoacoustic AEs.

| Score-Characteristic | C&W attack | | Psychoacoustic attack | |
| --- | --- | --- | --- | --- |
| | Noisy data vs. AEs | Benign data vs. AEs | Noisy data vs. AEs | Benign data vs. AEs |
| Mean-Entropy | 0.9889 / 0.9923 | 0.9895 / 0.9931 | 0.9899 / 0.9927 | 0.9906 / 0.9936 |
| Max-Entropy | 0.9604 / 0.9574 | 0.9775 / 0.9727 | 0.9587 / 0.9545 | 0.9745 / 0.9682 |
| Min-Entropy | 0.9922 / 0.9935 | 0.9913 / 0.9925 | 0.9908 / 0.9923 | 0.9895 / 0.9908 |
| Median-Entropy | 0.9846 / 0.9903 | 0.9935 / 0.9950 | 0.9858 / 0.9908 | **0.9943 / 0.9955** |
| Mean-Max | 0.9760 / 0.9838 | 0.9850 / 0.9914 | 0.9775 / 0.9846 | 0.9858 / 0.9917 |
| **Max-Max** | **0.9949 / 0.9954** | **0.9946 / 0.9950** | **0.9939 / 0.9946** | 0.9936 / 0.9941 |
| Min-Max | 0.9361 / 0.9053 | 0.9606 / 0.9245 | 0.9316 / 0.8820 | 0.9543 / 0.8989 |
| Median-Max | 0.9829 / 0.9891 | 0.9867 / 0.9921 | 0.9843 / 0.9899 | 0.9877 / 0.9925 |
| Mean-Min | 0.7303 / 0.7413 | 0.7314 / 0.7406 | 0.7405 / 0.7486 | 0.7405 / 0.7480 |
| Max-Min | 0.6505 / 0.6348 | 0.6734 / 0.6645 | 0.6444 / 0.6030 | 0.6639 / 0.6477 |
| Min-Min | 0.7277 / 0.7242 | 0.7140 / 0.6860 | 0.7293 / 0.7233 | 0.7140 / 0.6817 |
| Median-Min | 0.6071 / 0.5909 | 0.6065 / 0.5799 | 0.6152 / 0.5908 | 0.6137 / 0.5798 |
| Mean-Median | 0.9603 / 0.9648 | 0.9557 / 0.9598 | 0.9631 / 0.9669 | 0.9601 / 0.9629 |
| Max-Median | 0.9294 / 0.9210 | 0.9519 / 0.9481 | 0.9311 / 0.9265 | 0.9537 / 0.9524 |
| Min-Median | 0.8821 / 0.8876 | 0.8650 / 0.8693 | 0.8726 / 0.8657 | 0.8538 / 0.8477 |
| Median-Median | 0.7448 / 0.7329 | 0.7455 / 0.7365 | 0.7636 / 0.7668 | 0.7651 / 0.7686 |
| Mean-JSD | 0.9808 / 0.9866 | 0.9845 / 0.9903 | 0.9818 / 0.9874 | 0.9853 / 0.9909 |
| Max-JSD | 0.9706 / 0.9727 | 0.9756 / 0.9742 | 0.9668 / 0.9643 | 0.9717 / 0.9668 |
| Min-JSD | 0.6186 / 0.5869 | 0.6481 / 0.5707 | 0.6304 / 0.5882 | 0.6587 / 0.5735 |
| Median-JSD | 0.9848 / 0.9901 | 0.9875 / 0.9920 | 0.9855 / 0.9904 | 0.9873 / 0.9920 |
| Mean-KLD | 0.9836 / 0.9909 | 0.9866 / 0.9922 | 0.9840 / 0.9910 | 0.9870 / 0.9924 |
| Max-KLD | 0.8095 / 0.7712 | 0.8165 / 0.7515 | 0.8102 / 0.7734 | 0.8183 / 0.7604 |
| Min-KLD | 0.8343 / 0.8542 | 0.8807 / 0.8805 | 0.8316 / 0.8520 | 0.8787 / 0.8753 |
| Median-KLD | 0.9808 / 0.9875 | 0.9873 / 0.9919 | 0.9826 / 0.9892 | 0.9887 / 0.9932 |

Table 24: Comparing GCs on clean and noisy data for the wav2vec (Ma) model, assessing AUROC/AUPRC with 100 samples each from clean test data, C&W AEs, and Psychoacoustic AEs.

| Score-Characteristic | C&W attack | | Psychoacoustic attack | |
| --- | --- | --- | --- | --- |
| | Noisy data vs. AEs | Benign data vs. AEs | Noisy data vs. AEs | Benign data vs. AEs |
| **Mean-Entropy** | 0.9304 / 0.9516 | **0.9847 / 0.9861** | **0.9359 / 0.9563** | **0.9877 / 0.9890** |
| Max-Entropy | 0.9191 / 0.9464 | 0.9656 / 0.9757 | 0.9168 / 0.9449 | 0.9651 / 0.9751 |
| Min-Entropy | 0.4755 / 0.4991 | 0.4801 / 0.4981 | 0.4738 / 0.4991 | 0.4786 / 0.4984 |
| Median-Entropy | 0.8618 / 0.8151 | 0.8742 / 0.8076 | 0.8602 / 0.8284 | 0.8725 / 0.8293 |
| Mean-Max | 0.8018 / 0.7473 | 0.8562 / 0.7748 | 0.8205 / 0.7615 | 0.8777 / 0.7978 |
| Max-Max | 0.4655 / 0.4928 | 0.4710 / 0.5065 | 0.4881 / 0.4987 | 0.4935 / 0.5146 |
| Min-Max | **0.9413 / 0.9488** | 0.9752 / 0.9700 | 0.9341 / 0.9404 | 0.9686 / 0.9623 |
| Median-Max | 0.6932 / 0.6398 | 0.7052 / 0.6190 | 0.6779 / 0.6409 | 0.6910 / 0.6198 |
| Mean-Min | 0.8056 / 0.7694 | 0.8624 / 0.8121 | 0.8098 / 0.7834 | 0.8665 / 0.8325 |
| Max-Min | 0.8944 / 0.9026 | 0.9587 / 0.9433 | 0.9044 / 0.9236 | 0.9688 / 0.9671 |
| Min-Min | 0.4487 / 0.4762 | 0.4448 / 0.4822 | 0.4693 / 0.4943 | 0.4634 / 0.4911 |
| Median-Min | 0.8804 / 0.8710 | 0.9144 / 0.8960 | 0.8889 / 0.8911 | 0.9215 / 0.9185 |
| Mean-Median | 0.9383 / 0.9526 | 0.9814 / 0.9820 | 0.9431 / 0.9575 | 0.9859 / 0.9866 |
| Max-Median | 0.8783 / 0.8795 | 0.9119 / 0.8954 | 0.8933 / 0.9180 | 0.9272 / 0.9385 |
| Min-Median | 0.7322 / 0.6711 | 0.7558 / 0.7082 | 0.7264 / 0.6731 | 0.7504 / 0.7058 |
| Median-Median | 0.8705 / 0.8471 | 0.8981 / 0.8685 | 0.8870 / 0.8770 | 0.9142 / 0.8931 |
| Mean-JSD | 0.8066 / 0.7671 | 0.8121 / 0.7649 | 0.7888 / 0.7128 | 0.7904 / 0.7022 |
| Max-JSD | 0.9295 / 0.8795 | 0.9422 / 0.8917 | 0.9357 / 0.9023 | 0.9497 / 0.9110 |
| Min-JSD | 0.5726 / 0.5591 | 0.6205 / 0.5930 | 0.5966 / 0.5652 | 0.6435 / 0.6042 |
| Median-JSD | 0.8090 / 0.7942 | 0.7787 / 0.7749 | 0.8008 / 0.7889 | 0.7691 / 0.7689 |
| Mean-KLD | 0.9226 / 0.8971 | 0.9505 / 0.9198 | 0.9240 / 0.9090 | 0.9533 / 0.9327 |
| Max-KLD | 0.6909 / 0.6416 | 0.7639 / 0.7198 | 0.7057 / 0.6566 | 0.7748 / 0.7335 |
| Min-KLD | 0.5710 / 0.5529 | 0.6131 / 0.5778 | 0.5926 / 0.5774 | 0.6330 / 0.5976 |
| Median-KLD | 0.8711 / 0.8661 | 0.8685 / 0.8733 | 0.8759 / 0.8824 | 0.8735 / 0.8885 |

Table 25: Comparing GCs on clean and noisy data for the wav2vec (Ge) model, assessing AUROC/AUPRC with 100 samples each from clean test data, C&W AEs, and Psychoacoustic AEs.

| Score-Characteristic | C&W attack | | Psychoacoustic attack | |
|---|---|---|---|---|
| | Noisy data vs. AEs | Benign data vs. AEs | Noisy data vs. AEs | Benign data vs. AEs |
| Mean-Entropy | 0.9659 / 0.9609 | 0.9782 / 0.9755 | 0.9394 / 0.9313 | 0.9577 / 0.9506 |
| Max-Entropy | 0.9587 / 0.9682 | 0.9832 / 0.9867 | 0.9314 / 0.9181 | 0.9612 / 0.9333 |
| Min-Entropy | 0.6523 / 0.5829 | 0.6677 / 0.6020 | 0.6885 / 0.6313 | 0.7070 / 0.6421 |
| Median-Entropy | 0.9707 / 0.9674 | 0.9782 / 0.9720 | 0.9592 / 0.9482 | 0.9713 / 0.9534 |
| Mean-Max | 0.9000 / 0.8961 | 0.9176 / 0.9178 | 0.8729 / 0.8650 | 0.8958 / 0.8915 |
| Max-Max | 0.6418 / 0.5844 | 0.6521 / 0.5904 | 0.6518 / 0.5915 | 0.6621 / 0.5975 |
| Min-Max | 0.9277 / 0.9422 | 0.9313 / 0.9528 | 0.8933 / 0.8756 | 0.9003 / 0.8866 |
| Median-Max | 0.9652 / 0.9483 | 0.9722 / 0.9522 | 0.9571 / 0.9413 | 0.9697 / 0.9493 |
| Mean-Min | 0.9842 / 0.9878 | 0.9916 / 0.9951 | 0.9661 / 0.9673 | 0.9849 / 0.9874 |
| **Max-Min** | 0.9653 / 0.9725 | **0.9955 / 0.9968** | 0.9430 / 0.9515 | 0.9871 / 0.9886 |
| Min-Min | 0.6702 / 0.6191 | 0.7755 / 0.7527 | 0.6530 / 0.6029 | 0.7652 / 0.7377 |
| Median-Min | 0.9820 / 0.9867 | 0.9907 / 0.9949 | 0.9632 / 0.9668 | 0.9863 / 0.9902 |
| Mean-Median | **0.9882 / 0.9916** | 0.9925 / 0.9957 | 0.9762 / 0.9796 | **0.9904 / 0.9936** |
| Max-Median | 0.9722 / 0.9786 | 0.9945 / 0.9963 | 0.9512 / 0.9587 | 0.9886 / 0.9909 |
| Min-Median | 0.7049 / 0.6920 | 0.7855 / 0.7429 | 0.6907 / 0.6369 | 0.7594 / 0.6767 |
| Median-Median | 0.9871 / 0.9906 | 0.9920 / 0.9954 | 0.9739 / 0.9777 | 0.9901 / 0.9937 |
| Mean-JSD | 0.5886 / 0.5690 | 0.5926 / 0.5952 | 0.5925 / 0.5803 | 0.5955 / 0.5980 |
| Max-JSD | 0.6457 / 0.5488 | 0.6500 / 0.5470 | 0.5896 / 0.5152 | 0.5940 / 0.5139 |
| Min-JSD | 0.4062 / 0.4272 | 0.4276 / 0.4314 | 0.3717 / 0.4086 | 0.3920 / 0.4135 |
| Median-JSD | 0.4632 / 0.4811 | 0.4463 / 0.4690 | 0.4833 / 0.4844 | 0.4714 / 0.4742 |
| Mean-KLD | 0.9848 / 0.9881 | 0.9842 / 0.9882 | **0.9769 / 0.9805** | 0.9758 / 0.9802 |
| Max-KLD | 0.8647 / 0.8146 | 0.8945 / 0.8332 | 0.8966 / 0.8950 | 0.9299 / 0.9218 |
| Min-KLD | 0.4115 / 0.4252 | 0.4535 / 0.4692 | 0.3802 / 0.4101 | 0.4160 / 0.4386 |
| Median-KLD | 0.7156 / 0.7853 | 0.5832 / 0.7004 | 0.7028 / 0.7222 | 0.5787 / 0.6448 |

Table 26: Comparing GCs on clean and noisy data for the Trf (Ma) model, assessing AUROC/AUPRC with 100 samples each from clean test data, C&W AEs, and Psychoacoustic AEs.

| Score-Characteristic | C&W attack | | Psychoacoustic attack | |
|---|---|---|---|---|
| | Noisy data vs. AEs | Benign data vs. AEs | Noisy data vs. AEs | Benign data vs. AEs |
| Mean-Entropy | 0.9640 / 0.9643 | 0.9845 / 0.9806 | 0.9652 / 0.9723 | **0.9902 / 0.9913** |
| Max-Entropy | 0.9449 / 0.9462 | 0.9714 / 0.9686 | 0.9405 / 0.9470 | 0.9710 / 0.9704 |
| Min-Entropy | 0.7269 / 0.6768 | 0.7623 / 0.6765 | 0.7659 / 0.7399 | 0.8011 / 0.7427 |
| Median-Entropy | **0.9758 / 0.9516** | 0.9888 / 0.9651 | **0.9768 / 0.9672** | 0.9900 / 0.9756 |
| Mean-Max | 0.9370 / 0.9383 | 0.9640 / 0.9596 | 0.9350 / 0.9459 | 0.9658 / 0.9697 |
| Max-Max | 0.6942 / 0.6329 | 0.7444 / 0.6672 | 0.6895 / 0.6258 | 0.7362 / 0.6587 |
| Min-Max | 0.9392 / 0.9287 | 0.9616 / 0.9456 | 0.9279 / 0.9254 | 0.9532 / 0.9420 |
| **Median-Max** | 0.9754 / 0.9686 | **0.9902 / 0.9800** | 0.9749 / 0.9630 | 0.9898 / 0.9720 |
| Mean-Min | 0.9574 / 0.9604 | 0.9877 / 0.9846 | 0.9617 / 0.9670 | 0.9898 / 0.9895 |
| Max-Min | 0.9312 / 0.9381 | 0.9734 / 0.9694 | 0.9272 / 0.9415 | 0.9772 / 0.9782 |
| Min-Min | 0.8524 / 0.8271 | 0.8882 / 0.8563 | 0.8686 / 0.8486 | 0.9013 / 0.8673 |
| Median-Min | 0.9304 / 0.9414 | 0.9744 / 0.9755 | 0.9309 / 0.9420 | 0.9743 / 0.9766 |
| Mean-Median | 0.9501 / 0.9526 | 0.9813 / 0.9778 | 0.9552 / 0.9594 | 0.9838 / 0.9826 |
| Max-Median | 0.9135 / 0.9084 | 0.9288 / 0.9194 | 0.9056 / 0.9123 | 0.9221 / 0.9243 |
| Min-Median | 0.8269 / 0.8071 | 0.8673 / 0.8325 | 0.8516 / 0.8230 | 0.8908 / 0.8554 |
| Median-Median | 0.9126 / 0.9240 | 0.9582 / 0.9587 | 0.9155 / 0.9259 | 0.9600 / 0.9610 |
| Mean-JSD | 0.5298 / 0.5220 | 0.5471 / 0.5346 | 0.5321 / 0.5281 | 0.5448 / 0.5420 |
| Max-JSD | 0.7353 / 0.6682 | 0.7415 / 0.6570 | 0.8301 / 0.7646 | 0.8450 / 0.7649 |
| Min-JSD | 0.9071 / 0.8349 | 0.9096 / 0.8314 | 0.8217 / 0.7458 | 0.8222 / 0.7262 |
| Median-JSD | 0.9445 / 0.9453 | 0.9653 / 0.9551 | 0.9407 / 0.9305 | 0.9601 / 0.9355 |
| Mean-KLD | 0.8082 / 0.7941 | 0.8447 / 0.8413 | 0.8027 / 0.7663 | 0.8415 / 0.8126 |
| Max-KLD | 0.7386 / 0.6619 | 0.7507 / 0.6879 | 0.7329 / 0.6543 | 0.7450 / 0.6851 |
| Min-KLD | 0.8820 / 0.8628 | 0.8943 / 0.8720 | 0.7920 / 0.7664 | 0.8091 / 0.7729 |
| Median-KLD | 0.9443 / 0.9482 | 0.9689 / 0.9578 | 0.9364 / 0.9161 | 0.9637 / 0.9315 |

Table 27: Comparing GCs on clean and noisy data for the Trf (En) model, assessing AUROC/AUPRC with 100 samples each from clean test data, C&W AEs, and Psychoacoustic AEs.

| | C&W attack | | Psychoacoustic attack | |
| Score-Characteristic | Noisy data vs. AEs | Benign data vs. AEs | Noisy data vs. AEs | Benign data vs. AEs |
|---|---|---|---|---|
| Mean-Entropy | **0.9897 / 0.9911** | 0.9999 / 0.9999 | 0.9794 / 0.9796 | 0.9959 / 0.9951 |
| Max-Entropy | 0.9509 / 0.9663 | 0.9989 / 0.9989 | 0.9397 / 0.9380 | 0.9895 / 0.9700 |
| Min-Entropy | 0.9352 / 0.9142 | 0.9635 / 0.9272 | 0.9271 / 0.9184 | 0.9629 / 0.9421 |
| Median-Entropy | 0.9846 / 0.9885 | 0.9997 / 0.9997 | 0.9789 / 0.9836 | 0.9984 / 0.9985 |
| Mean-Max | 0.9861 / 0.9880 | 0.9994 / 0.9994 | 0.9672 / 0.9635 | 0.9884 / 0.9844 |
| Max-Max | 0.9248 / 0.9088 | 0.9633 / 0.9428 | 0.9138 / 0.8991 | 0.9619 / 0.9395 |
| Min-Max | 0.9252 / 0.9097 | 0.9682 / 0.9417 | 0.9032 / 0.8830 | 0.9486 / 0.9134 |
| Median-Max | 0.9835 / 0.9878 | 0.9992 / 0.9992 | 0.9787 / 0.9834 | 0.9983 / 0.9984 |
| Mean-Min | 0.9733 / 0.9808 | 0.9985 / 0.9987 | 0.9666 / 0.9761 | 0.9979 / 0.9983 |
| Max-Min | 0.9005 / 0.9425 | 0.9999 / 0.9999 | 0.9009 / 0.9422 | 0.9999 / 0.9999 |
| Min-Min | 0.9054 / 0.8908 | 0.9338 / 0.9183 | 0.8948 / 0.8617 | 0.9248 / 0.8941 |
| Median-Min | 0.9717 / 0.9787 | 0.9984 / 0.9986 | 0.9618 / 0.9709 | 0.9975 / 0.9980 |
| Mean-Median | 0.9864 / 0.9900 | 0.9998 / 0.9998 | 0.9821 / 0.9867 | 0.9993 / 0.9993 |
| **Max-Median** | 0.9299 / 0.9566 | **1.0000 / 1.0000** | 0.9269 / 0.9541 | **1.0000 / 1.0000** |
| Min-Median | 0.9307 / 0.9256 | 0.9560 / 0.9516 | 0.9230 / 0.9137 | 0.9498 / 0.9435 |
| Median-Median | **0.9897 / 0.9919** | 0.9999 / 0.9999 | **0.9860 / 0.9892** | 0.9996 / 0.9996 |
| Mean-JSD | 0.5715 / 0.5876 | 0.5621 / 0.5674 | 0.5934 / 0.5940 | 0.5864 / 0.5836 |
| Max-JSD | 0.9664 / 0.9657 | 0.9840 / 0.9756 | 0.9660 / 0.9655 | 0.9851 / 0.9761 |
| Min-JSD | 0.8169 / 0.7506 | 0.8112 / 0.7362 | 0.7960 / 0.7053 | 0.7899 / 0.6875 |
| Median-JSD | 0.9180 / 0.8757 | 0.9400 / 0.8769 | 0.8828 / 0.8001 | 0.9039 / 0.8093 |
| Mean-KLD | 0.8612 / 0.9011 | 0.8797 / 0.9104 | 0.8683 / 0.9056 | 0.8868 / 0.9165 |
| Max-KLD | 0.8734 / 0.8825 | 0.8912 / 0.8982 | 0.8833 / 0.8808 | 0.8986 / 0.8947 |
| Min-KLD | 0.7115 / 0.6631 | 0.7557 / 0.6719 | 0.6937 / 0.6744 | 0.7463 / 0.6902 |
| Median-KLD | 0.8882 / 0.8065 | 0.9094 / 0.8030 | 0.8670 / 0.7941 | 0.8863 / 0.7789 |

# H  EVALUATION OF BINARY CLASSIFIERS FOR IDENTIFYING TARGETED ATTACKS

To evaluate the performance of our classifiers, we compute various metrics, including accuracy, false positive rate (FPR), true positive rate (TPR), precision, recall, and F1 score. These metrics are derived from the analysis of two types of errors: false positives (FP) and true negatives (TN) across all models. To calculate these metrics, we employ a conservative threshold chosen based on a validation set to achieve a maximum 1% FPR (if applicable) while maintaining a minimum 50% TPR. To determine the threshold, the validation set considers only benign data and C&W AEs, which is then applied to noisy data and Psychoacoustic AEs.

**LSTM (It) model performance**  Tab. 28: C&W AEs vs. benign data, Tab. 29: C&W AEs vs. noisy data, Tab. 30: Psychoacoustic AEs vs. benign data, Tab. 31: Psychoacoustic AEs vs. noisy data.

Table 28: LSTM (It) binary classifiers' metrics, using a threshold of maximum 1% FPR (if available) and a minimum 50% TPR, with 100 benign data and 100 C&W AEs.

| Classifier | Accuracy | TP | FP | TN | FN | FPR | TPR | Precision | Recall | F1 |
|---|---|---|---|---|---|---|---|---|---|---|
| NF | 80.50% | 67 | 6 | 94 | 33 | 0.06 | 0.67 | 0.9178 | 0.6700 | 0.7746 |
| TD | 77.50% | 64 | 9 | 91 | 36 | 0.09 | 0.64 | 0.8767 | 0.6400 | 0.7399 |
| GC | 98.00% | 96 | 0 | 100 | 4 | 0.00 | 0.96 | 1.0000 | 0.9600 | 0.9796 |
| EM=3 | 92.50% | 85 | 0 | 100 | 15 | 0.00 | 0.85 | 1.0000 | 0.8500 | 0.9189 |
| EM=5 | 89.50% | 79 | 0 | 100 | 21 | 0.00 | 0.79 | 1.0000 | 0.7900 | 0.8827 |
| EM=7 | 86.50% | 73 | 0 | 100 | 27 | 0.00 | 0.73 | 1.0000 | 0.7300 | 0.8439 |
| EM=9 | 86.50% | 73 | 0 | 100 | 27 | 0.00 | 0.73 | 1.0000 | 0.7300 | 0.8439 |
| NN | 96.00% | 92 | 0 | 100 | 8 | 0.00 | 0.92 | 1.0000 | 0.9200 | 0.9583 |

Table 29: LSTM (It) binary classifiers' metrics, using a threshold of maximum 1% FPR (if applicable) and a minimum 50% TPR, with 100 noisy data and 100 C&W AEs.

| Classifier | Accuracy | TP | FP | TN | FN | FPR | TPR | Precision | Recall | F1 |
|---|---|---|---|---|---|---|---|---|---|---|
| NF | 77.00% | 60 | 6 | 94 | 40 | 0.06 | 0.60 | 0.9091 | 0.6000 | 0.7229 |
| TD | 75.00% | 59 | 9 | 91 | 41 | 0.09 | 0.59 | 0.8676 | 0.5900 | 0.7024 |
| GC | 88.50% | 77 | 0 | 100 | 23 | 0.00 | 0.77 | 1.0000 | 0.7700 | 0.8701 |
| EM=3 | 83.00% | 66 | 0 | 100 | 34 | 0.00 | 0.66 | 1.0000 | 0.6600 | 0.7952 |
| EM=5 | 79.50% | 59 | 0 | 100 | 41 | 0.00 | 0.59 | 1.0000 | 0.5900 | 0.7421 |
| EM=7 | 78.00% | 56 | 0 | 100 | 44 | 0.00 | 0.56 | 1.0000 | 0.5600 | 0.7179 |
| EM=9 | 77.50% | 55 | 0 | 100 | 45 | 0.00 | 0.55 | 1.0000 | 0.5500 | 0.7097 |
| NN | 85.00% | 71 | 1 | 99 | 29 | 0.01 | 0.71 | 0.9861 | 0.7100 | 0.8256 |

Table 30: LSTM (It) binary classifiers' metrics, with a threshold of maximum 1% FPR (if exist) and a minimum 50% TPR, with 100 benign and 100 Psychoacoustic AEs.

| Classifier | Accuracy | TP | FP | TN | FN | FPR | TPR | Precision | Recall | F1 |
|---|---|---|---|---|---|---|---|---|---|---|
| NF | 82.00% | 67 | 3 | 97 | 33 | 0.03 | 0.67 | 0.9571 | 0.6700 | 0.7882 |
| TD | 77.50% | 64 | 9 | 91 | 36 | 0.09 | 0.64 | 0.8767 | 0.6400 | 0.7399 |
| GC | 97.50% | 96 | 1 | 99 | 4 | 0.01 | 0.96 | 0.9897 | 0.9600 | 0.9746 |
| EM=3 | 92.50% | 85 | 0 | 100 | 15 | 0.00 | 0.85 | 1.0000 | 0.8500 | 0.9189 |
| EM=5 | 89.50% | 79 | 0 | 100 | 21 | 0.00 | 0.79 | 1.0000 | 0.7900 | 0.8827 |
| EM=7 | 86.00% | 73 | 1 | 99 | 27 | 0.01 | 0.73 | 0.9865 | 0.7300 | 0.8391 |
| EM=9 | 86.50% | 73 | 0 | 100 | 27 | 0.00 | 0.73 | 1.0000 | 0.7300 | 0.8439 |
| NN | 96.00% | 92 | 0 | 100 | 8 | 0.00 | 0.92 | 1.0000 | 0.9200 | 0.9583 |

Table 31: LSTM (It) binary classifiers' metrics, with a threshold of maximum 1% FPR (if exist) and a minimum 50% TPR, with 100 noisy data and 100 Psychoacoustic AEs.

| Classifier | Accuracy | TP | FP | TN | FN | FPR | TPR | Precision | Recall | F1 |
|---|---|---|---|---|---|---|---|---|---|---|
| NF | 78.50% | 60 | 3 | 97 | 40 | 0.03 | 0.60 | 0.9524 | 0.6000 | 0.7362 |
| TD | 75.00% | 59 | 9 | 91 | 41 | 0.09 | 0.59 | 0.8676 | 0.5900 | 0.7024 |
| GC | 88.00% | 77 | 1 | 99 | 23 | 0.01 | 0.77 | 0.9872 | 0.7700 | 0.8652 |
| EM=3 | 83.00% | 66 | 0 | 100 | 34 | 0.00 | 0.66 | 1.0000 | 0.6600 | 0.7952 |
| EM=5 | 79.50% | 59 | 0 | 100 | 41 | 0.00 | 0.59 | 1.0000 | 0.5900 | 0.7421 |
| EM=7 | 77.50% | 56 | 1 | 99 | 44 | 0.01 | 0.56 | 0.9825 | 0.5600 | 0.7134 |
| EM=9 | 77.50% | 55 | 0 | 100 | 45 | 0.00 | 0.55 | 1.0000 | 0.5500 | 0.7097 |
| NN | 85.00% | 71 | 1 | 99 | 29 | 0.01 | 0.71 | 0.9861 | 0.7100 | 0.8256 |

**LSTM (En) model performance**  Tab. 32: C&W AEs vs. benign data, Tab. 33: C&W AEs vs. noisy data, Tab. 34: Psychoacoustic AEs vs. benign data, Tab. 35: Psychoacoustic AEs vs. noisy data.

Table 32: LSTM (En) binary classifiers' metrics, using a threshold of maximum 1% FPR (if available) and a minimum 50% TPR, with 100 benign data and 100 C&W AEs.

| Classifier | Accuracy | TP | FP | TN | FN | FPR | TPR | Precision | Recall | F1 |
|---|---|---|---|---|---|---|---|---|---|---|
| NF | 80.50% | 62 | 1 | 99 | 38 | 0.01 | 0.62 | 0.9841 | 0.6200 | 0.7607 |
| TD | 82.41% | 65 | 1 | 99 | 34 | 0.01 | 0.66 | 0.9848 | 0.6566 | 0.7879 |
| GC | 98.50% | 100 | 3 | 97 | 0 | 0.03 | 1.00 | 0.9709 | 1.0000 | 0.9852 |
| EM=3 | 98.50% | 97 | 0 | 100 | 3 | 0.00 | 0.97 | 1.0000 | 0.9700 | 0.9848 |
| EM=5 | 99.00% | 98 | 0 | 100 | 2 | 0.00 | 0.98 | 1.0000 | 0.9800 | 0.9899 |
| EM=7 | 99.00% | 98 | 0 | 100 | 2 | 0.00 | 0.98 | 1.0000 | 0.9800 | 0.9899 |
| EM=9 | 98.50% | 97 | 0 | 100 | 3 | 0.00 | 0.97 | 1.0000 | 0.9700 | 0.9848 |
| NN | 98.00% | 97 | 1 | 99 | 3 | 0.01 | 0.97 | 0.9898 | 0.9700 | 0.9798 |

Table 33: LSTM (En) binary classifiers' metrics, using a threshold of maximum 1% FPR (if applicable) and a minimum 50% TPR, with 100 noisy data and 100 C&W AEs.

| Classifier | Accuracy | TP | FP | TN | FN | FPR | TPR | Precision | Recall | F1 |
|---|---|---|---|---|---|---|---|---|---|---|
| NF | 78.50% | 58 | 1 | 99 | 42 | 0.01 | 0.58 | 0.9831 | 0.5800 | 0.7296 |
| TD | 82.41% | 65 | 1 | 99 | 34 | 0.01 | 0.66 | 0.9848 | 0.6566 | 0.7879 |
| GC | 97.50% | 98 | 3 | 97 | 2 | 0.03 | 0.98 | 0.9703 | 0.9800 | 0.9751 |
| EM=3 | 96.50% | 93 | 0 | 100 | 7 | 0.00 | 0.93 | 1.0000 | 0.9300 | 0.9637 |
| EM=5 | 96.50% | 93 | 0 | 100 | 7 | 0.00 | 0.93 | 1.0000 | 0.9300 | 0.9637 |
| EM=7 | 96.50% | 93 | 0 | 100 | 7 | 0.00 | 0.93 | 1.0000 | 0.9300 | 0.9637 |
| EM=9 | 96.00% | 92 | 0 | 100 | 8 | 0.00 | 0.92 | 1.0000 | 0.9200 | 0.9583 |
| NN | 95.50% | 92 | 1 | 99 | 8 | 0.01 | 0.92 | 0.9892 | 0.9200 | 0.9534 |

Table 34: LSTM (En) binary classifiers' metrics, with a threshold of maximum 1% FPR (if exist) and a minimum 50% TPR, with 100 benign and 100 Psychoacoustic AEs.

| Classifier | Accuracy | TP | FP | TN | FN | FPR | TPR | Precision | Recall | F1 |
|---|---|---|---|---|---|---|---|---|---|---|
| NF | 80.00% | 62 | 2 | 98 | 38 | 0.02 | 0.62 | 0.9688 | 0.6200 | 0.7561 |
| TD | 82.41% | 65 | 1 | 99 | 34 | 0.01 | 0.66 | 0.9848 | 0.6566 | 0.7879 |
| GC | 98.50% | 100 | 3 | 97 | 0 | 0.03 | 1.00 | 0.9709 | 1.0000 | 0.9852 |
| EM=3 | 98.50% | 97 | 0 | 100 | 3 | 0.00 | 0.97 | 1.0000 | 0.9700 | 0.9848 |
| EM=5 | 99.00% | 98 | 0 | 100 | 2 | 0.00 | 0.98 | 1.0000 | 0.9800 | 0.9899 |
| EM=7 | 99.00% | 98 | 0 | 100 | 2 | 0.00 | 0.98 | 1.0000 | 0.9800 | 0.9899 |
| EM=9 | 98.50% | 97 | 0 | 100 | 3 | 0.00 | 0.97 | 1.0000 | 0.9700 | 0.9848 |
| NN | 98.00% | 97 | 1 | 99 | 3 | 0.01 | 0.97 | 0.9898 | 0.9700 | 0.9798 |

Table 35: LSTM (En) binary classifiers' metrics, with a threshold of maximum 1% FPR (if exist) and a minimum 50% TPR, with 100 noisy data and 100 Psychoacoustic AEs.

| Classifier | Accuracy | TP | FP | TN | FN | FPR | TPR | Precision | Recall | F1 |
|---|---|---|---|---|---|---|---|---|---|---|
| NF | 78.00% | 58 | 2 | 98 | 42 | 0.02 | 0.58 | 0.9667 | 0.5800 | 0.7250 |
| TD | 82.41% | 65 | 1 | 99 | 34 | 0.01 | 0.66 | 0.9848 | 0.6566 | 0.7879 |
| GC | 97.50% | 98 | 3 | 97 | 2 | 0.03 | 0.98 | 0.9703 | 0.9800 | 0.9751 |
| EM=3 | 96.50% | 93 | 0 | 100 | 7 | 0.00 | 0.93 | 1.0000 | 0.9300 | 0.9637 |
| EM=5 | 96.50% | 93 | 0 | 100 | 7 | 0.00 | 0.93 | 1.0000 | 0.9300 | 0.9637 |
| EM=7 | 96.50% | 93 | 0 | 100 | 7 | 0.00 | 0.93 | 1.0000 | 0.9300 | 0.9637 |
| EM=9 | 96.00% | 92 | 0 | 100 | 8 | 0.00 | 0.92 | 1.0000 | 0.9200 | 0.9583 |
| NN | 95.50% | 92 | 1 | 99 | 8 | 0.01 | 0.92 | 0.9892 | 0.9200 | 0.9534 |

**LSTM (En-LM) model performance**  Tab. 36: C&W AEs vs. benign data, Tab. 37: C&W AEs vs. noisy data, Tab. 38: Psychoacoustic AEs vs. benign data, Tab. 39: Psychoacoustic AEs vs. noisy data.

Table 36: LSTM (En-LM) binary classifiers' metrics, using a threshold of maximum 1% FPR (if available) and a minimum 50% TPR, with 100 benign data and 100 C&W AEs.

| Classifier | Accuracy | TP | FP | TN | FN | FPR | TPR | Precision | Recall | F1 |
|---|---|---|---|---|---|---|---|---|---|---|
| NF | 89.00% | 78 | 0 | 100 | 22 | 0.00 | 0.78 | 1.0000 | 0.7800 | 0.8764 |
| TD | 81.50% | 70 | 7 | 93 | 30 | 0.07 | 0.70 | 0.9091 | 0.7000 | 0.7910 |
| GC | 83.00% | 67 | 1 | 99 | 33 | 0.01 | 0.67 | 0.9853 | 0.6700 | 0.7976 |
| EM=3 | 91.00% | 84 | 2 | 98 | 16 | 0.02 | 0.84 | 0.9767 | 0.8400 | 0.9032 |
| EM=5 | 93.00% | 87 | 1 | 99 | 13 | 0.01 | 0.87 | 0.9886 | 0.8700 | 0.9255 |
| EM=7 | 97.00% | 95 | 1 | 99 | 5 | 0.01 | 0.95 | 0.9896 | 0.9500 | 0.9694 |
| EM=9 | 94.00% | 90 | 2 | 98 | 10 | 0.02 | 0.90 | 0.9783 | 0.9000 | 0.9375 |
| NN | 92.50% | 86 | 1 | 99 | 14 | 0.01 | 0.86 | 0.9885 | 0.8600 | 0.9198 |

Table 37: LSTM (En-LM) binary classifiers' metrics, using a threshold of maximum 1% FPR (if applicable) and a minimum 50% TPR, with 100 noisy data and 100 C&W AEs.

| Classifier | Accuracy | TP | FP | TN | FN | FPR | TPR | Precision | Recall | F1 |
|---|---|---|---|---|---|---|---|---|---|---|
| NF | 83.50% | 67 | 0 | 100 | 33 | 0.00 | 0.67 | 1.0000 | 0.6700 | 0.8024 |
| TD | 80.00% | 67 | 7 | 93 | 33 | 0.07 | 0.67 | 0.9054 | 0.6700 | 0.7701 |
| GC | 79.00% | 59 | 1 | 99 | 41 | 0.01 | 0.59 | 0.9833 | 0.5900 | 0.7375 |
| EM=3 | 86.50% | 75 | 2 | 98 | 25 | 0.02 | 0.75 | 0.9740 | 0.7500 | 0.8475 |
| EM=5 | 88.00% | 77 | 1 | 99 | 23 | 0.01 | 0.77 | 0.9872 | 0.7700 | 0.8652 |
| EM=7 | 93.00% | 87 | 1 | 99 | 13 | 0.01 | 0.87 | 0.9886 | 0.8700 | 0.9255 |
| EM=9 | 88.00% | 78 | 2 | 98 | 22 | 0.02 | 0.78 | 0.9750 | 0.7800 | 0.8667 |
| NN | 91.50% | 84 | 1 | 99 | 16 | 0.01 | 0.84 | 0.9882 | 0.8400 | 0.9081 |

Table 38: LSTM (En-LM) binary classifiers' metrics, with a threshold of maximum 1% FPR (if exist) and a minimum 50% TPR, with 100 benign and 100 Psychoacoustic AEs.

| Classifier | Accuracy | TP | FP | TN | FN | FPR | TPR | Precision | Recall | F1 |
|---|---|---|---|---|---|---|---|---|---|---|
| NF | 89.00% | 78 | 0 | 100 | 22 | 0.00 | 0.78 | 1.0000 | 0.7800 | 0.8764 |
| TD | 82.50% | 70 | 5 | 95 | 30 | 0.05 | 0.70 | 0.9333 | 0.7000 | 0.8000 |
| GC | 83.50% | 67 | 0 | 100 | 33 | 0.00 | 0.67 | 1.0000 | 0.6700 | 0.8024 |
| EM=3 | 91.00% | 84 | 2 | 98 | 16 | 0.02 | 0.84 | 0.9767 | 0.8400 | 0.9032 |
| EM=5 | 93.00% | 87 | 1 | 99 | 13 | 0.01 | 0.87 | 0.9886 | 0.8700 | 0.9255 |
| EM=7 | 97.00% | 95 | 1 | 99 | 5 | 0.01 | 0.95 | 0.9896 | 0.9500 | 0.9694 |
| EM=9 | 94.50% | 90 | 1 | 99 | 10 | 0.01 | 0.90 | 0.9890 | 0.9000 | 0.9424 |
| NN | 92.50% | 86 | 1 | 99 | 14 | 0.01 | 0.86 | 0.9885 | 0.8600 | 0.9198 |

Table 39: LSTM (En-LM) binary classifiers' metrics, with a threshold of maximum 1% FPR (if exist) and a minimum 50% TPR, with 100 noisy data and 100 Psychoacoustic AEs.

| Classifier | Accuracy | TP | FP | TN | FN | FPR | TPR | Precision | Recall | F1 |
|---|---|---|---|---|---|---|---|---|---|---|
| NF | 83.50% | 67 | 0 | 100 | 33 | 0.00 | 0.67 | 1.0000 | 0.6700 | 0.8024 |
| TD | 81.00% | 67 | 5 | 95 | 33 | 0.05 | 0.67 | 0.9306 | 0.6700 | 0.7791 |
| GC | 79.50% | 59 | 0 | 100 | 41 | 0.00 | 0.59 | 1.0000 | 0.5900 | 0.7421 |
| EM=3 | 86.50% | 75 | 2 | 98 | 25 | 0.02 | 0.75 | 0.9740 | 0.7500 | 0.8475 |
| EM=5 | 88.00% | 77 | 1 | 99 | 23 | 0.01 | 0.77 | 0.9872 | 0.7700 | 0.8652 |
| EM=7 | 93.00% | 87 | 1 | 99 | 13 | 0.01 | 0.87 | 0.9886 | 0.8700 | 0.9255 |
| EM=9 | 88.50% | 78 | 1 | 99 | 22 | 0.01 | 0.78 | 0.9873 | 0.7800 | 0.8715 |
| NN | 91.50% | 84 | 1 | 99 | 16 | 0.01 | 0.84 | 0.9882 | 0.8400 | 0.9081 |

**wav2vec (Ma) model performance**  Tab. 40: C&W AEs vs. benign data, Tab. 41: C&W AEs vs. noisy data, Tab. 42: Psychoacoustic AEs vs. benign data, Tab. 43: Psychoacoustic AEs vs. noisy data.

Table 40: wav2vec (Ma) binary classifiers' metrics, using a threshold of maximum 1% FPR (if available) and a minimum 50% TPR, with 100 benign data and 100 C&W AEs.

| Classifier | Accuracy | TP | FP | TN | FN | FPR | TPR | Precision | Recall | F1 |
|---|---|---|---|---|---|---|---|---|---|---|
| NF | 73.00% | 47 | 1 | 99 | 53 | 0.01 | 0.47 | 0.9792 | 0.4700 | 0.6351 |
| TD | 97.50% | 95 | 0 | 100 | 5 | 0.00 | 0.95 | 1.0000 | 0.9500 | 0.9744 |
| GC | 90.50% | 81 | 0 | 100 | 19 | 0.00 | 0.81 | 1.0000 | 0.8100 | 0.8950 |
| EM=3 | 94.00% | 89 | 1 | 99 | 11 | 0.01 | 0.89 | 0.9889 | 0.8900 | 0.9368 |
| EM=5 | 92.50% | 85 | 0 | 100 | 15 | 0.00 | 0.85 | 1.0000 | 0.8500 | 0.9189 |
| EM=7 | 87.00% | 74 | 0 | 100 | 26 | 0.00 | 0.74 | 1.0000 | 0.7400 | 0.8506 |
| EM=9 | 86.50% | 73 | 0 | 100 | 27 | 0.00 | 0.73 | 1.0000 | 0.7300 | 0.8439 |
| NN | 93.50% | 88 | 1 | 99 | 12 | 0.01 | 0.88 | 0.9888 | 0.8800 | 0.9312 |

Table 41: wav2vec (Ma) binary classifiers' metrics, using a threshold of maximum 1% FPR (if applicable) and a minimum 50% TPR, with 100 noisy data and 100 C&W AEs.

| Classifier | Accuracy | TP | FP | TN | FN | FPR | TPR | Precision | Recall | F1 |
|---|---|---|---|---|---|---|---|---|---|---|
| NF | 63.00% | 27 | 1 | 99 | 73 | 0.01 | 0.27 | 0.9643 | 0.2700 | 0.4219 |
| TD | 96.00% | 92 | 0 | 100 | 8 | 0.00 | 0.92 | 1.0000 | 0.9200 | 0.9583 |
| GC | 85.00% | 70 | 0 | 100 | 30 | 0.00 | 0.70 | 1.0000 | 0.7000 | 0.8235 |
| EM=3 | 87.50% | 76 | 1 | 99 | 24 | 0.01 | 0.76 | 0.9870 | 0.7600 | 0.8588 |
| EM=5 | 86.50% | 73 | 0 | 100 | 27 | 0.00 | 0.73 | 1.0000 | 0.7300 | 0.8439 |
| EM=7 | 83.50% | 67 | 0 | 100 | 33 | 0.00 | 0.67 | 1.0000 | 0.6700 | 0.8024 |
| EM=9 | 83.50% | 67 | 0 | 100 | 33 | 0.00 | 0.67 | 1.0000 | 0.6700 | 0.8024 |
| NN | 85.50% | 72 | 1 | 99 | 28 | 0.01 | 0.72 | 0.9863 | 0.7200 | 0.8324 |

Table 42: wav2vec (Ma) binary classifiers' metrics, with a threshold of maximum 1% FPR (if exist) and a minimum 50% TPR, with 100 benign and 100 Psychoacoustic AEs.

| Classifier | Accuracy | TP | FP | TN | FN | FPR | TPR | Precision | Recall | F1 |
|---|---|---|---|---|---|---|---|---|---|---|
| NF | 73.00% | 47 | 1 | 99 | 53 | 0.01 | 0.47 | 0.9792 | 0.4700 | 0.6351 |
| TD | 97.50% | 95 | 0 | 100 | 5 | 0.00 | 0.95 | 1.0000 | 0.9500 | 0.9744 |
| GC | 90.50% | 81 | 0 | 100 | 19 | 0.00 | 0.81 | 1.0000 | 0.8100 | 0.8950 |
| EM=3 | 94.00% | 89 | 1 | 99 | 11 | 0.01 | 0.89 | 0.9889 | 0.8900 | 0.9368 |
| EM=5 | 92.50% | 85 | 0 | 100 | 15 | 0.00 | 0.85 | 1.0000 | 0.8500 | 0.9189 |
| EM=7 | 87.00% | 74 | 0 | 100 | 26 | 0.00 | 0.74 | 1.0000 | 0.7400 | 0.8506 |
| EM=9 | 86.50% | 73 | 0 | 100 | 27 | 0.00 | 0.73 | 1.0000 | 0.7300 | 0.8439 |
| NN | 93.50% | 88 | 1 | 99 | 12 | 0.01 | 0.88 | 0.9888 | 0.8800 | 0.9312 |

Table 43: wav2vec (Ma) binary classifiers' metrics, with a threshold of maximum 1% FPR (if exist) and a minimum 50% TPR, with 100 noisy data and 100 Psychoacoustic AEs.

| Classifier | Accuracy | TP | FP | TN | FN | FPR | TPR | Precision | Recall | F1 |
|---|---|---|---|---|---|---|---|---|---|---|
| NF | 63.00% | 27 | 1 | 99 | 73 | 0.01 | 0.27 | 0.9643 | 0.2700 | 0.4219 |
| TD | 96.00% | 92 | 0 | 100 | 8 | 0.00 | 0.92 | 1.0000 | 0.9200 | 0.9583 |
| GC | 85.00% | 70 | 0 | 100 | 30 | 0.00 | 0.70 | 1.0000 | 0.7000 | 0.8235 |
| EM=3 | 87.50% | 76 | 1 | 99 | 24 | 0.01 | 0.76 | 0.9870 | 0.7600 | 0.8588 |
| EM=5 | 86.50% | 73 | 0 | 100 | 27 | 0.00 | 0.73 | 1.0000 | 0.7300 | 0.8439 |
| EM=7 | 83.50% | 67 | 0 | 100 | 33 | 0.00 | 0.67 | 1.0000 | 0.6700 | 0.8024 |
| EM=9 | 83.50% | 67 | 0 | 100 | 33 | 0.00 | 0.67 | 1.0000 | 0.6700 | 0.8024 |
| NN | 85.50% | 72 | 1 | 99 | 28 | 0.01 | 0.72 | 0.9863 | 0.7200 | 0.8324 |

**wav2vec (Ge) model performance**  Tab. 44: C&W AEs vs. benign data, Tab. 45: C&W AEs vs. noisy data, Tab. 46: Psychoacoustic AEs vs. benign data, Tab. 47: Psychoacoustic AEs vs. noisy data.

Table 44: wav2vec (Ge) binary classifiers' metrics, using a threshold of maximum 1% FPR (if available) and a minimum 50% TPR, with 100 benign data and 100 C&W AEs.

| Classifier | Accuracy | TP | FP | TN | FN | FPR | TPR | Precision | Recall | F1 |
|---|---|---|---|---|---|---|---|---|---|---|
| NF | 89.50% | 80 | 1 | 99 | 20 | 0.01 | 0.80 | 0.9877 | 0.8000 | 0.8840 |
| TD | 98.00% | 96 | 0 | 100 | 4 | 0.00 | 0.96 | 1.0000 | 0.9600 | 0.9796 |
| GC | 99.00% | 98 | 0 | 100 | 2 | 0.00 | 0.98 | 1.0000 | 0.9800 | 0.9899 |
| EM=3 | 96.00% | 92 | 0 | 100 | 8 | 0.00 | 0.92 | 1.0000 | 0.9200 | 0.9583 |
| EM=5 | 96.00% | 92 | 0 | 100 | 8 | 0.00 | 0.92 | 1.0000 | 0.9200 | 0.9583 |
| EM=7 | 94.50% | 89 | 0 | 100 | 11 | 0.00 | 0.89 | 1.0000 | 0.8900 | 0.9418 |
| EM=9 | 95.50% | 91 | 0 | 100 | 9 | 0.00 | 0.91 | 1.0000 | 0.9100 | 0.9529 |
| NN | 94.00% | 89 | 1 | 99 | 11 | 0.01 | 0.89 | 0.9889 | 0.8900 | 0.9368 |

Table 45: wav2vec (Ge) binary classifiers' metrics, using a threshold of maximum 1% FPR (if applicable) and a minimum 50% TPR, with 100 noisy data and 100 C&W AEs.

| Classifier | Accuracy | TP | FP | TN | FN | FPR | TPR | Precision | Recall | F1 |
|---|---|---|---|---|---|---|---|---|---|---|
| NF | 91.50% | 84 | 1 | 99 | 16 | 0.01 | 0.84 | 0.9882 | 0.8400 | 0.9081 |
| TD | 94.00% | 88 | 0 | 100 | 12 | 0.00 | 0.88 | 1.0000 | 0.8800 | 0.9362 |
| GC | 95.50% | 91 | 0 | 100 | 9 | 0.00 | 0.91 | 1.0000 | 0.9100 | 0.9529 |
| EM=3 | 91.50% | 83 | 0 | 100 | 17 | 0.00 | 0.83 | 1.0000 | 0.8300 | 0.9071 |
| EM=5 | 91.50% | 83 | 0 | 100 | 17 | 0.00 | 0.83 | 1.0000 | 0.8300 | 0.9071 |
| EM=7 | 89.50% | 79 | 0 | 100 | 21 | 0.00 | 0.79 | 1.0000 | 0.7900 | 0.8827 |
| EM=9 | 91.50% | 83 | 0 | 100 | 17 | 0.00 | 0.83 | 1.0000 | 0.8300 | 0.9071 |
| NN | 88.00% | 77 | 1 | 99 | 23 | 0.01 | 0.77 | 0.9872 | 0.7700 | 0.8652 |

Table 46: wav2vec (Ge) binary classifiers' metrics, with a threshold of maximum 1% FPR (if exist) and a minimum 50% TPR, with 100 benign and 100 Psychoacoustic AEs.

| Classifier | Accuracy | TP | FP | TN | FN | FPR | TPR | Precision | Recall | F1 |
|---|---|---|---|---|---|---|---|---|---|---|
| NF | 89.50% | 80 | 1 | 99 | 20 | 0.01 | 0.80 | 0.9877 | 0.8000 | 0.8840 |
| TD | 98.00% | 96 | 0 | 100 | 4 | 0.00 | 0.96 | 1.0000 | 0.9600 | 0.9796 |
| GC | 99.00% | 98 | 0 | 100 | 2 | 0.00 | 0.98 | 1.0000 | 0.9800 | 0.9899 |
| EM=3 | 93.00% | 92 | 6 | 94 | 8 | 0.06 | 0.92 | 0.9388 | 0.9200 | 0.9293 |
| EM=5 | 92.00% | 92 | 8 | 92 | 8 | 0.08 | 0.92 | 0.9200 | 0.9200 | 0.9200 |
| EM=7 | 90.50% | 89 | 8 | 92 | 11 | 0.08 | 0.89 | 0.9175 | 0.8900 | 0.9036 |
| EM=9 | 93.00% | 91 | 5 | 95 | 9 | 0.05 | 0.91 | 0.9479 | 0.9100 | 0.9286 |
| NN | 92.50% | 86 | 1 | 99 | 14 | 0.01 | 0.86 | 0.9885 | 0.8600 | 0.9198 |

Table 47: wav2vec (Ge) binary classifiers' metrics, with a threshold of maximum 1% FPR (if exist) and a minimum 50% TPR, with 100 noisy data and 100 Psychoacoustic AEs.

| Classifier | Accuracy | TP | FP | TN | FN | FPR | TPR | Precision | Recall | F1 |
|---|---|---|---|---|---|---|---|---|---|---|
| NF | 91.50% | 84 | 1 | 99 | 16 | 0.01 | 0.84 | 0.9882 | 0.8400 | 0.9081 |
| TD | 94.00% | 88 | 0 | 100 | 12 | 0.00 | 0.88 | 1.0000 | 0.8800 | 0.9362 |
| GC | 95.50% | 91 | 0 | 100 | 9 | 0.00 | 0.91 | 1.0000 | 0.9100 | 0.9529 |
| EM=3 | 88.50% | 83 | 6 | 94 | 17 | 0.06 | 0.83 | 0.9326 | 0.8300 | 0.8783 |
| EM=5 | 87.50% | 83 | 8 | 92 | 17 | 0.08 | 0.83 | 0.9121 | 0.8300 | 0.8691 |
| EM=7 | 85.50% | 79 | 8 | 92 | 21 | 0.08 | 0.79 | 0.9080 | 0.7900 | 0.8449 |
| EM=9 | 89.00% | 83 | 5 | 95 | 17 | 0.05 | 0.83 | 0.9432 | 0.8300 | 0.8830 |
| NN | 86.00% | 73 | 1 | 99 | 27 | 0.01 | 0.73 | 0.9865 | 0.7300 | 0.8391 |

**Trf (Ma) model performance**  Tab. 48: C&W AEs vs. benign data, Tab. 49: C&W AEs vs. noisy data, Tab. 50: Psychoacoustic AEs vs. benign data, Tab. 51: Psychoacoustic AEs vs. noisy data.

Table 48: Trf (Ma) binary classifiers' metrics, using a threshold of maximum 1% FPR (if available) and a minimum 50% TPR, with 100 benign data and 100 C&W AEs.

| Classifier | Accuracy | TP | FP | TN | FN | FPR | TPR | Precision | Recall | F1 |
|---|---|---|---|---|---|---|---|---|---|---|
| NF | 95.00% | 90 | 0 | 100 | 10 | 0.00 | 0.90 | 1.0000 | 0.9000 | 0.9474 |
| TD | 81.00% | 64 | 2 | 98 | 36 | 0.02 | 0.64 | 0.9697 | 0.6400 | 0.7711 |
| GC | 87.50% | 75 | 0 | 100 | 25 | 0.00 | 0.75 | 1.0000 | 0.7500 | 0.8571 |
| EM=3 | 90.50% | 81 | 0 | 100 | 19 | 0.00 | 0.81 | 1.0000 | 0.8100 | 0.8950 |
| EM=5 | 90.00% | 80 | 0 | 100 | 20 | 0.00 | 0.80 | 1.0000 | 0.8000 | 0.8889 |
| EM=7 | 90.50% | 81 | 0 | 100 | 19 | 0.00 | 0.81 | 1.0000 | 0.8100 | 0.8950 |
| EM=9 | 87.50% | 75 | 0 | 100 | 25 | 0.00 | 0.75 | 1.0000 | 0.7500 | 0.8571 |
| NN | 98.50% | 98 | 1 | 99 | 2 | 0.01 | 0.98 | 0.9899 | 0.9800 | 0.9849 |

Table 49: Trf (Ma) binary classifiers' metrics, using a threshold of maximum 1% FPR (if applicable) and a minimum 50% TPR, with 100 noisy data and 100 C&W AEs.

| Classifier | Accuracy | TP | FP | TN | FN | FPR | TPR | Precision | Recall | F1 |
|---|---|---|---|---|---|---|---|---|---|---|
| NF | 96.50% | 93 | 0 | 100 | 7 | 0.00 | 0.93 | 1.0000 | 0.9300 | 0.9637 |
| TD | 81.50% | 65 | 2 | 98 | 35 | 0.02 | 0.65 | 0.9701 | 0.6500 | 0.7784 |
| GC | 84.50% | 69 | 0 | 100 | 31 | 0.00 | 0.69 | 1.0000 | 0.6900 | 0.8166 |
| EM=3 | 83.00% | 66 | 0 | 100 | 34 | 0.00 | 0.66 | 1.0000 | 0.6600 | 0.7952 |
| EM=5 | 84.00% | 68 | 0 | 100 | 32 | 0.00 | 0.68 | 1.0000 | 0.6800 | 0.8095 |
| EM=7 | 84.00% | 68 | 0 | 100 | 32 | 0.00 | 0.68 | 1.0000 | 0.6800 | 0.8095 |
| EM=9 | 82.50% | 65 | 0 | 100 | 35 | 0.00 | 0.65 | 1.0000 | 0.6500 | 0.7879 |
| NN | 91.50% | 84 | 1 | 99 | 16 | 0.01 | 0.84 | 0.9882 | 0.8400 | 0.9081 |

Table 50: Trf (Ma) binary classifiers' metrics, with a threshold of maximum 1% FPR (if exist) and a minimum 50% TPR, with 100 benign and 100 Psychoacoustic AEs.

| Classifier | Accuracy | TP | FP | TN | FN | FPR | TPR | Precision | Recall | F1 |
|---|---|---|---|---|---|---|---|---|---|---|
| NF | 95.00% | 90 | 0 | 100 | 10 | 0.00 | 0.90 | 1.0000 | 0.9000 | 0.9474 |
| TD | 82.00% | 64 | 0 | 100 | 36 | 0.00 | 0.64 | 1.0000 | 0.6400 | 0.7805 |
| GC | 87.50% | 75 | 0 | 100 | 25 | 0.00 | 0.75 | 1.0000 | 0.7500 | 0.8571 |
| EM=3 | 90.50% | 81 | 0 | 100 | 19 | 0.00 | 0.81 | 1.0000 | 0.8100 | 0.8950 |
| EM=5 | 90.00% | 80 | 0 | 100 | 20 | 0.00 | 0.80 | 1.0000 | 0.8000 | 0.8889 |
| EM=7 | 90.50% | 81 | 0 | 100 | 19 | 0.00 | 0.81 | 1.0000 | 0.8100 | 0.8950 |
| EM=9 | 87.50% | 75 | 0 | 100 | 25 | 0.00 | 0.75 | 1.0000 | 0.7500 | 0.8571 |
| NN | 97.50% | 96 | 1 | 99 | 4 | 0.01 | 0.96 | 0.9897 | 0.9600 | 0.9746 |

Table 51: Trf (Ma) binary classifiers' metrics, with a threshold of maximum 1% FPR (if exist) and a minimum 50% TPR, with 100 noisy data and 100 Psychoacoustic AEs.

| Classifier | Accuracy | TP | FP | TN | FN | FPR | TPR | Precision | Recall | F1 |
|---|---|---|---|---|---|---|---|---|---|---|
| NF | 96.50% | 93 | 0 | 100 | 7 | 0.00 | 0.93 | 1.0000 | 0.9300 | 0.9637 |
| TD | 82.50% | 65 | 0 | 100 | 35 | 0.00 | 0.65 | 1.0000 | 0.6500 | 0.7879 |
| GC | 84.50% | 69 | 0 | 100 | 31 | 0.00 | 0.69 | 1.0000 | 0.6900 | 0.8166 |
| EM=3 | 83.00% | 66 | 0 | 100 | 34 | 0.00 | 0.66 | 1.0000 | 0.6600 | 0.7952 |
| EM=5 | 84.00% | 68 | 0 | 100 | 32 | 0.00 | 0.68 | 1.0000 | 0.6800 | 0.8095 |
| EM=7 | 84.00% | 68 | 0 | 100 | 32 | 0.00 | 0.68 | 1.0000 | 0.6800 | 0.8095 |
| EM=9 | 82.50% | 65 | 0 | 100 | 35 | 0.00 | 0.65 | 1.0000 | 0.6500 | 0.7879 |
| NN | 90.50% | 82 | 1 | 99 | 18 | 0.01 | 0.82 | 0.9880 | 0.8200 | 0.8962 |

**Trf (En) model performance** Tab. 52: C&W AEs vs. benign data, Tab. 53: C&W AEs vs. noisy data, Tab. 54: Psychoacoustic AEs vs. benign data, Tab. 55: Psychoacoustic AEs vs. noisy data.

Table 52: Trf (En) binary classifiers' metrics, using a threshold of maximum 1% FPR (if available) and a minimum 50% TPR, with 100 benign data and 100 C&W AEs.

| Classifier | Accuracy | TP | FP | TN | FN | FPR | TPR | Precision | Recall | F1 |
|---|---|---|---|---|---|---|---|---|---|---|
| NF | 95.00% | 91 | 1 | 99 | 9 | 0.01 | 0.91 | 0.9891 | 0.9100 | 0.9479 |
| TD | 95.00% | 96 | 6 | 94 | 4 | 0.06 | 0.96 | 0.9412 | 0.9600 | 0.9505 |
| GC | 100.00% | 100 | 0 | 100 | 0 | 0.00 | 1.00 | 1.0000 | 1.0000 | 1.0000 |
| EM=3 | 99.50% | 99 | 0 | 100 | 1 | 0.00 | 0.99 | 1.0000 | 0.9900 | 0.9950 |
| EM=5 | 100.00% | 100 | 0 | 100 | 0 | 0.00 | 1.00 | 1.0000 | 1.0000 | 1.0000 |
| EM=7 | 100.00% | 100 | 0 | 100 | 0 | 0.00 | 1.00 | 1.0000 | 1.0000 | 1.0000 |
| EM=9 | 100.00% | 100 | 0 | 100 | 0 | 0.00 | 1.00 | 1.0000 | 1.0000 | 1.0000 |
| NN | 99.00% | 99 | 1 | 99 | 1 | 0.01 | 0.99 | 0.9900 | 0.9900 | 0.9900 |

Table 53: Trf (En) binary classifiers' metrics, using a threshold of maximum 1% FPR (if applicable) and a minimum 50% TPR, with 100 noisy data and 100 C&W AEs.

| Classifier | Accuracy | TP | FP | TN | FN | FPR | TPR | Precision | Recall | F1 |
|---|---|---|---|---|---|---|---|---|---|---|
| NF | 96.50% | 94 | 1 | 99 | 6 | 0.01 | 0.94 | 0.9895 | 0.9400 | 0.9641 |
| TD | 88.00% | 82 | 6 | 94 | 18 | 0.06 | 0.82 | 0.9318 | 0.8200 | 0.8723 |
| GC | 96.50% | 93 | 0 | 100 | 7 | 0.00 | 0.93 | 1.0000 | 0.9300 | 0.9637 |
| EM=3 | 94.50% | 89 | 0 | 100 | 11 | 0.00 | 0.89 | 1.0000 | 0.8900 | 0.9418 |
| EM=5 | 95.00% | 90 | 0 | 100 | 10 | 0.00 | 0.90 | 1.0000 | 0.9000 | 0.9474 |
| EM=7 | 95.00% | 90 | 0 | 100 | 10 | 0.00 | 0.90 | 1.0000 | 0.9000 | 0.9474 |
| EM=9 | 94.50% | 89 | 0 | 100 | 11 | 0.00 | 0.89 | 1.0000 | 0.8900 | 0.9418 |
| NN | 92.50% | 86 | 1 | 99 | 14 | 0.01 | 0.86 | 0.9885 | 0.8600 | 0.9198 |

Table 54: Trf (En) binary classifiers' metrics, with a threshold of maximum 1% FPR (if exist) and a minimum 50% TPR, with 100 benign and 100 Psychoacoustic AEs.

| Classifier | Accuracy | TP | FP | TN | FN | FPR | TPR | Precision | Recall | F1 |
|---|---|---|---|---|---|---|---|---|---|---|
| NF | 95.00% | 91 | 1 | 99 | 9 | 0.01 | 0.91 | 0.9891 | 0.9100 | 0.9479 |
| TD | 97.50% | 96 | 1 | 99 | 4 | 0.01 | 0.96 | 0.9897 | 0.9600 | 0.9746 |
| GC | 99.50% | 100 | 1 | 99 | 0 | 0.01 | 1.00 | 0.9901 | 1.0000 | 0.9950 |
| EM=3 | 99.00% | 99 | 1 | 99 | 1 | 0.01 | 0.99 | 0.9900 | 0.9900 | 0.9900 |
| EM=5 | 99.50% | 100 | 1 | 99 | 0 | 0.01 | 1.00 | 0.9901 | 1.0000 | 0.9950 |
| EM=7 | 99.50% | 100 | 1 | 99 | 0 | 0.01 | 1.00 | 0.9901 | 1.0000 | 0.9950 |
| EM=9 | 99.50% | 100 | 1 | 99 | 0 | 0.01 | 1.00 | 0.9901 | 1.0000 | 0.9950 |
| NN | 98.50% | 98 | 1 | 99 | 2 | 0.01 | 0.98 | 0.9899 | 0.9800 | 0.9849 |

Table 55: Trf (En) binary classifiers' metrics, with a threshold of maximum 1% FPR (if exist) and a minimum 50% TPR, with 100 noisy data and 100 Psychoacoustic AEs.

| Classifier | Accuracy | TP | FP | TN | FN | FPR | TPR | Precision | Recall | F1 |
|---|---|---|---|---|---|---|---|---|---|---|
| NF | 96.50% | 94 | 1 | 99 | 6 | 0.01 | 0.94 | 0.9895 | 0.9400 | 0.9641 |
| TD | 90.50% | 82 | 1 | 99 | 18 | 0.01 | 0.82 | 0.9880 | 0.8200 | 0.8962 |
| GC | 96.00% | 93 | 1 | 99 | 7 | 0.01 | 0.93 | 0.9894 | 0.9300 | 0.9588 |
| EM=3 | 94.00% | 89 | 1 | 99 | 11 | 0.01 | 0.89 | 0.9889 | 0.8900 | 0.9368 |
| EM=5 | 94.50% | 90 | 1 | 99 | 10 | 0.01 | 0.90 | 0.9890 | 0.9000 | 0.9424 |
| EM=7 | 94.50% | 90 | 1 | 99 | 10 | 0.01 | 0.90 | 0.9890 | 0.9000 | 0.9424 |
| EM=9 | 94.00% | 89 | 1 | 99 | 11 | 0.01 | 0.89 | 0.9889 | 0.8900 | 0.9368 |
| NN | 90.00% | 81 | 1 | 99 | 19 | 0.01 | 0.81 | 0.9878 | 0.8100 | 0.8901 |

# I  EVALUATION OF BINARY CLASSIFIERS FOR IDENTIFYING UNTARGETED ATTACKS

To evaluate the performance of our classifiers, we compute the same metrics as defined in Section H. To determine the threshold, the validation set considers only benign data and C&W AEs, which is then applied to PGD, genetic and Kenansville AEs.

**LSTM (It) model performance**  Tab. 56: PGD AEs vs. benign data, Tab. 57: genetic AEs vs. benign data, Tab. 58: Kenansville AEs vs. benign data.

Table 56: LSTM (It) binary classifiers' metrics, using a threshold of maximum 1% FPR (if available) and a minimum 50% TPR, with 100 benign data and 100 PGD AEs.

| Classifier | Accuracy | TP | FP | TN | FN | FPR | TPR | Precision | Recall | F1 |
|---|---|---|---|---|---|---|---|---|---|---|
| NF | 69.50% | 67 | 28 | 72 | 33 | 0.28 | 0.67 | 0.7053 | 0.6700 | 0.6872 |
| TD | 64.00% | 64 | 36 | 64 | 36 | 0.36 | 0.64 | 0.6400 | 0.6400 | 0.6400 |
| GC | 86.50% | 96 | 23 | 77 | 4 | 0.23 | 0.96 | 0.8067 | 0.9600 | 0.8767 |
| EM=3 | 85.00% | 85 | 15 | 85 | 15 | 0.15 | 0.85 | 0.8500 | 0.8500 | 0.8500 |
| EM=5 | 83.50% | 79 | 12 | 88 | 21 | 0.12 | 0.79 | 0.8681 | 0.7900 | 0.8272 |
| EM=7 | 81.00% | 73 | 11 | 89 | 27 | 0.11 | 0.73 | 0.8690 | 0.7300 | 0.7935 |
| EM=9 | 80.50% | 73 | 12 | 88 | 27 | 0.12 | 0.73 | 0.8588 | 0.7300 | 0.7892 |
| NN | 81.00% | 63 | 1 | 99 | 37 | 0.01 | 0.63 | 0.9844 | 0.6300 | 0.7683 |

Table 57: LSTM (It) binary classifiers' metrics, using a threshold of maximum 1% FPR (if available) and a minimum 50% TPR, with 100 benign data and 100 genetic AEs.

| Classifier | Accuracy | TP | FP | TN | FN | FPR | TPR | Precision | Recall | F1 |
|---|---|---|---|---|---|---|---|---|---|---|
| NF | 53.50% | 67 | 60 | 40 | 33 | 0.60 | 0.67 | 0.5276 | 0.6700 | 0.5903 |
| TD | 51.00% | 64 | 62 | 38 | 36 | 0.62 | 0.64 | 0.5079 | 0.6400 | 0.5664 |
| GC | 56.00% | 96 | 84 | 16 | 4 | 0.84 | 0.96 | 0.5333 | 0.9600 | 0.6857 |
| EM=3 | 55.50% | 85 | 74 | 26 | 15 | 0.74 | 0.85 | 0.5346 | 0.8500 | 0.6564 |
| EM=5 | 55.50% | 79 | 68 | 32 | 21 | 0.68 | 0.79 | 0.5374 | 0.7900 | 0.6397 |
| EM=7 | 52.00% | 73 | 69 | 31 | 27 | 0.69 | 0.73 | 0.5141 | 0.7300 | 0.6033 |
| EM=9 | 51.50% | 73 | 70 | 30 | 27 | 0.70 | 0.73 | 0.5105 | 0.7300 | 0.6008 |
| NN | 62.50% | 49 | 24 | 76 | 51 | 0.24 | 0.49 | 0.6712 | 0.4900 | 0.5665 |

Table 58: LSTM (It) binary classifiers' metrics, using a threshold of maximum 1% FPR (if available) and a minimum 50% TPR, with 100 benign data and 100 Kenansville AEs.

| Classifier | Accuracy | TP | FP | TN | FN | FPR | TPR | Precision | Recall | F1 |
|---|---|---|---|---|---|---|---|---|---|---|
| NF | 79.00% | 67 | 9 | 91 | 33 | 0.09 | 0.67 | 0.8816 | 0.6700 | 0.7614 |
| TD | 65.99% | 64 | 31 | 66 | 36 | 0.32 | 0.64 | 0.6737 | 0.6400 | 0.6564 |
| GC | 78.00% | 96 | 40 | 60 | 4 | 0.40 | 0.96 | 0.7059 | 0.9600 | 0.8136 |
| EM=3 | 83.00% | 85 | 19 | 81 | 15 | 0.19 | 0.85 | 0.8173 | 0.8500 | 0.8333 |
| EM=5 | 82.50% | 79 | 14 | 86 | 21 | 0.14 | 0.79 | 0.8495 | 0.7900 | 0.8187 |
| EM=7 | 80.00% | 73 | 13 | 87 | 27 | 0.13 | 0.73 | 0.8488 | 0.7300 | 0.7849 |
| EM=9 | 79.50% | 73 | 14 | 86 | 27 | 0.14 | 0.73 | 0.8391 | 0.7300 | 0.7807 |
| NN | 70.50% | 49 | 8 | 92 | 51 | 0.08 | 0.49 | 0.8596 | 0.4900 | 0.6242 |

**LSTM (En) model performance**    Tab. 59: PGD AEs vs. benign data, Tab. 60: genetic AEs vs. benign data, Tab. 61: Kenansville AEs vs. benign data.

Table 59: LSTM (En) binary classifiers' metrics, using a threshold of maximum 1% FPR (if available) and a minimum 50% TPR, with 100 benign data and 100 PGD AEs.

| Classifier | Accuracy | TP | FP | TN | FN | FPR | TPR | Precision | Recall | F1 |
|---|---|---|---|---|---|---|---|---|---|---|
| NF | 79.00% | 62 | 4 | 96 | 38 | 0.04 | 0.62 | 0.9394 | 0.6200 | 0.7470 |
| TD | 64.82% | 65 | 36 | 64 | 34 | 0.36 | 0.66 | 0.6436 | 0.6566 | 0.6500 |
| GC | 76.00% | 100 | 48 | 52 | 0 | 0.48 | 1.00 | 0.6757 | 1.0000 | 0.8065 |
| EM=3 | 84.00% | 97 | 29 | 71 | 3 | 0.29 | 0.97 | 0.7698 | 0.9700 | 0.8584 |
| EM=5 | 87.00% | 98 | 24 | 76 | 2 | 0.24 | 0.98 | 0.8033 | 0.9800 | 0.8829 |
| EM=7 | 88.50% | 98 | 21 | 79 | 2 | 0.21 | 0.98 | 0.8235 | 0.9800 | 0.8950 |
| EM=9 | 88.00% | 97 | 21 | 79 | 3 | 0.21 | 0.97 | 0.8220 | 0.9700 | 0.8899 |
| NN | 83.50% | 68 | 1 | 99 | 32 | 0.01 | 0.68 | 0.9855 | 0.6800 | 0.8047 |

Table 60: LSTM (En) binary classifiers' metrics, using a threshold of maximum 1% FPR (if available) and a minimum 50% TPR, with 100 benign data and 100 genetic AEs.

| Classifier | Accuracy | TP | FP | TN | FN | FPR | TPR | Precision | Recall | F1 |
|---|---|---|---|---|---|---|---|---|---|---|
| NF | 51.00% | 62 | 60 | 40 | 38 | 0.60 | 0.62 | 0.5082 | 0.6200 | 0.5586 |
| TD | 50.51% | 65 | 64 | 35 | 34 | 0.65 | 0.66 | 0.5039 | 0.6566 | 0.5702 |
| GC | 50.50% | 100 | 99 | 1 | 0 | 0.99 | 1.00 | 0.5025 | 1.0000 | 0.6689 |
| EM=3 | 53.00% | 97 | 91 | 9 | 3 | 0.91 | 0.97 | 0.5160 | 0.9700 | 0.6736 |
| EM=5 | 52.50% | 98 | 93 | 7 | 2 | 0.93 | 0.98 | 0.5131 | 0.9800 | 0.6735 |
| EM=7 | 53.50% | 98 | 91 | 9 | 2 | 0.91 | 0.98 | 0.5185 | 0.9800 | 0.6782 |
| EM=9 | 53.00% | 97 | 91 | 9 | 3 | 0.91 | 0.97 | 0.5160 | 0.9700 | 0.6736 |
| NN | 58.50% | 49 | 32 | 68 | 51 | 0.32 | 0.49 | 0.6049 | 0.4900 | 0.5414 |

Table 61: LSTM (En) binary classifiers' metrics, using a threshold of maximum 1% FPR (if available) and a minimum 50% TPR, with 100 benign data and 100 Kenansville AEs.

| Classifier | Accuracy | TP | FP | TN | FN | FPR | TPR | Precision | Recall | F1 |
|---|---|---|---|---|---|---|---|---|---|---|
| NF | 72.50% | 62 | 17 | 83 | 38 | 0.17 | 0.62 | 0.7848 | 0.6200 | 0.6927 |
| TD | 67.84% | 65 | 30 | 70 | 34 | 0.30 | 0.66 | 0.6842 | 0.6566 | 0.6701 |
| GC | 82.50% | 100 | 35 | 65 | 0 | 0.35 | 1.00 | 0.7407 | 1.0000 | 0.8511 |
| EM=3 | 83.50% | 97 | 30 | 70 | 3 | 0.30 | 0.97 | 0.7638 | 0.9700 | 0.8546 |
| EM=5 | 84.00% | 98 | 30 | 70 | 2 | 0.30 | 0.98 | 0.7656 | 0.9800 | 0.8596 |
| EM=7 | 84.00% | 98 | 30 | 70 | 2 | 0.30 | 0.98 | 0.7656 | 0.9800 | 0.8596 |
| EM=9 | 84.50% | 97 | 28 | 72 | 3 | 0.28 | 0.97 | 0.7760 | 0.9700 | 0.8622 |
| NN | 70.00% | 49 | 9 | 91 | 51 | 0.09 | 0.49 | 0.8448 | 0.4900 | 0.6203 |

**LSTM (En-LM) model performance**    Tab. 62: PGD AEs vs. benign data, Tab. 63: genetic AEs vs. benign data, Tab. 64: Kenansville AEs vs. benign data.

Table 62: LSTM (En-LM) binary classifiers' metrics, using a threshold of maximum 1% FPR (if available) and a minimum 50% TPR, with 100 benign data and 100 PGD AEs.

| Classifier | Accuracy | TP | FP | TN | FN | FPR | TPR | Precision | Recall | F1 |
|---|---|---|---|---|---|---|---|---|---|---|
| NF | 89.00% | 78 | 0 | 100 | 22 | 0.00 | 0.78 | 1.0000 | 0.7800 | 0.8764 |
| TD | 77.00% | 70 | 16 | 84 | 30 | 0.16 | 0.70 | 0.8140 | 0.7000 | 0.7527 |
| GC | 83.50% | 67 | 0 | 100 | 33 | 0.00 | 0.67 | 1.0000 | 0.6700 | 0.8024 |
| EM=3 | 87.50% | 84 | 9 | 91 | 16 | 0.09 | 0.84 | 0.9032 | 0.8400 | 0.8705 |
| EM=5 | 92.50% | 87 | 2 | 98 | 13 | 0.02 | 0.87 | 0.9775 | 0.8700 | 0.9206 |
| EM=7 | 90.00% | 95 | 15 | 85 | 5 | 0.15 | 0.95 | 0.8636 | 0.9500 | 0.9048 |
| EM=9 | 93.50% | 90 | 3 | 97 | 10 | 0.03 | 0.90 | 0.9677 | 0.9000 | 0.9326 |
| NN | 72.50% | 49 | 4 | 96 | 51 | 0.04 | 0.49 | 0.9245 | 0.4900 | 0.6405 |

Table 63: LSTM (En-LM) binary classifiers' metrics, using a threshold of maximum 1% FPR (if available) and a minimum 50% TPR, with 100 benign data and 100 genetic AEs.

| Classifier | Accuracy | TP | FP | TN | FN | FPR | TPR | Precision | Recall | F1 |
|---|---|---|---|---|---|---|---|---|---|---|
| NF | 63.50% | 78 | 51 | 49 | 22 | 0.51 | 0.78 | 0.6047 | 0.7800 | 0.6812 |
| TD | 53.00% | 70 | 64 | 36 | 30 | 0.64 | 0.70 | 0.5224 | 0.7000 | 0.5983 |
| GC | 58.50% | 67 | 50 | 50 | 33 | 0.50 | 0.67 | 0.5726 | 0.6700 | 0.6175 |
| EM=3 | 63.00% | 84 | 58 | 42 | 16 | 0.58 | 0.84 | 0.5915 | 0.8400 | 0.6942 |
| EM=5 | 63.50% | 87 | 60 | 40 | 13 | 0.60 | 0.87 | 0.5918 | 0.8700 | 0.7045 |
| EM=7 | 60.00% | 95 | 75 | 25 | 5 | 0.75 | 0.95 | 0.5588 | 0.9500 | 0.7037 |
| EM=9 | 65.00% | 90 | 60 | 40 | 10 | 0.60 | 0.90 | 0.6000 | 0.9000 | 0.7200 |
| NN | 63.50% | 49 | 22 | 78 | 51 | 0.22 | 0.49 | 0.6901 | 0.4900 | 0.5731 |

Table 64: LSTM (En-LM) binary classifiers' metrics, using a threshold of maximum 1% FPR (if available) and a minimum 50% TPR, with 100 benign data and 100 Kenansville AEs.

| Classifier | Accuracy | TP | FP | TN | FN | FPR | TPR | Precision | Recall | F1 |
|---|---|---|---|---|---|---|---|---|---|---|
| NF | 77.50% | 78 | 23 | 77 | 22 | 0.23 | 0.78 | 0.7723 | 0.7800 | 0.7761 |
| TD | 69.50% | 70 | 31 | 69 | 30 | 0.31 | 0.70 | 0.6931 | 0.7000 | 0.6965 |
| GC | 70.50% | 67 | 26 | 74 | 33 | 0.26 | 0.67 | 0.7204 | 0.6700 | 0.6943 |
| EM=3 | 78.50% | 84 | 27 | 73 | 16 | 0.27 | 0.84 | 0.7568 | 0.8400 | 0.7962 |
| EM=5 | 78.50% | 87 | 30 | 70 | 13 | 0.30 | 0.87 | 0.7436 | 0.8700 | 0.8018 |
| EM=7 | 82.00% | 95 | 31 | 69 | 5 | 0.31 | 0.95 | 0.7540 | 0.9500 | 0.8407 |
| EM=9 | 80.50% | 90 | 29 | 71 | 10 | 0.29 | 0.90 | 0.7563 | 0.9000 | 0.8219 |
| NN | 72.00% | 49 | 5 | 95 | 51 | 0.05 | 0.49 | 0.9074 | 0.4900 | 0.6364 |

**wav2vec (Ma) model performance** Tab. 65: PGD AEs vs. benign data, Tab. 66: genetic AEs vs. benign data, Tab. 67: Kenansville AEs vs. benign data.

Table 65: wav2vec (Ma) binary classifiers' metrics, using a threshold of maximum 1% FPR (if available) and a minimum 50% TPR, with 100 benign data and 100 PGD AEs.

| Classifier | Accuracy | TP | FP | TN | FN | FPR | TPR | Precision | Recall | F1 |
|---|---|---|---|---|---|---|---|---|---|---|
| NF | 73.50% | 47 | 0 | 100 | 53 | 0.00 | 0.47 | 1.0000 | 0.4700 | 0.6395 |
| TD | 59.00% | 95 | 77 | 23 | 5 | 0.77 | 0.95 | 0.5523 | 0.9500 | 0.6985 |
| GC | 77.50% | 81 | 26 | 74 | 19 | 0.26 | 0.81 | 0.7570 | 0.8100 | 0.7826 |
| EM=3 | 80.50% | 89 | 28 | 72 | 11 | 0.28 | 0.89 | 0.7607 | 0.8900 | 0.8203 |
| EM=5 | 84.00% | 85 | 17 | 83 | 15 | 0.17 | 0.85 | 0.8333 | 0.8500 | 0.8416 |
| EM=7 | 82.50% | 74 | 9 | 91 | 26 | 0.09 | 0.74 | 0.8916 | 0.7400 | 0.8087 |
| EM=9 | 80.00% | 73 | 13 | 87 | 27 | 0.13 | 0.73 | 0.8488 | 0.7300 | 0.7849 |
| NN | 68.00% | 49 | 13 | 87 | 51 | 0.13 | 0.49 | 0.7903 | 0.4900 | 0.6049 |

Table 66: wav2vec (Ma) binary classifiers' metrics, using a threshold of maximum 1% FPR (if available) and a minimum 50% TPR, with 100 benign data and 100 genetic AEs.

| Classifier | Accuracy | TP | FP | TN | FN | FPR | TPR | Precision | Recall | F1 |
|---|---|---|---|---|---|---|---|---|---|---|
| NF | 61.50% | 47 | 24 | 76 | 53 | 0.24 | 0.47 | 0.6620 | 0.4700 | 0.5497 |
| TD | 57.00% | 95 | 81 | 19 | 5 | 0.81 | 0.95 | 0.5398 | 0.9500 | 0.6884 |
| GC | 59.50% | 81 | 62 | 38 | 19 | 0.62 | 0.81 | 0.5664 | 0.8100 | 0.6667 |
| EM=3 | 60.50% | 89 | 68 | 32 | 11 | 0.68 | 0.89 | 0.5669 | 0.8900 | 0.6926 |
| EM=5 | 63.50% | 85 | 58 | 42 | 15 | 0.58 | 0.85 | 0.5944 | 0.8500 | 0.6996 |
| EM=7 | 60.00% | 74 | 54 | 46 | 26 | 0.54 | 0.74 | 0.5781 | 0.7400 | 0.6491 |
| EM=9 | 58.50% | 73 | 56 | 44 | 27 | 0.56 | 0.73 | 0.5659 | 0.7300 | 0.6376 |
| NN | 64.50% | 49 | 20 | 80 | 51 | 0.20 | 0.49 | 0.7101 | 0.4900 | 0.5799 |

Table 67: wav2vec (Ma) binary classifiers' metrics, using a threshold of maximum 1% FPR (if available) and a minimum 50% TPR, with 100 benign data and 100 Kenansville AEs.

| Classifier | Accuracy | TP | FP | TN | FN | FPR | TPR | Precision | Recall | F1 |
|---|---|---|---|---|---|---|---|---|---|---|
| NF | 73.00% | 47 | 1 | 99 | 53 | 0.01 | 0.47 | 0.9792 | 0.4700 | 0.6351 |
| TD | 75.50% | 95 | 44 | 56 | 5 | 0.44 | 0.95 | 0.6835 | 0.9500 | 0.7950 |
| GC | 88.00% | 81 | 5 | 95 | 19 | 0.05 | 0.81 | 0.9419 | 0.8100 | 0.8710 |
| EM=3 | 92.50% | 89 | 4 | 96 | 11 | 0.04 | 0.89 | 0.9570 | 0.8900 | 0.9223 |
| EM=5 | 90.00% | 85 | 5 | 95 | 15 | 0.05 | 0.85 | 0.9444 | 0.8500 | 0.8947 |
| EM=7 | 84.00% | 74 | 6 | 94 | 26 | 0.06 | 0.74 | 0.9250 | 0.7400 | 0.8222 |
| EM=9 | 84.00% | 73 | 5 | 95 | 27 | 0.05 | 0.73 | 0.9359 | 0.7300 | 0.8202 |
| NN | 72.50% | 49 | 4 | 96 | 51 | 0.04 | 0.49 | 0.9245 | 0.4900 | 0.6405 |

**wav2vec (Ge) model performance**  Tab. 68: PGD AEs vs. benign data, Tab. 69: genetic AEs vs. benign data, Tab. 70: Kenansville AEs vs. benign data.

Table 68: wav2vec (Ge) binary classifiers' metrics, using a threshold of maximum 1% FPR (if available) and a minimum 50% TPR, with 100 benign data and 100 PGD AEs.

| Classifier | Accuracy | TP | FP | TN | FN | FPR | TPR | Precision | Recall | F1 |
|---|---|---|---|---|---|---|---|---|---|---|
| NF | 46.00% | 80 | 88 | 12 | 20 | 0.88 | 0.80 | 0.4762 | 0.8000 | 0.5970 |
| TD | 64.50% | 96 | 67 | 33 | 4 | 0.67 | 0.96 | 0.5890 | 0.9600 | 0.7300 |
| GC | 63.50% | 98 | 71 | 29 | 2 | 0.71 | 0.98 | 0.5799 | 0.9800 | 0.7286 |
| EM=3 | 69.00% | 92 | 54 | 46 | 8 | 0.54 | 0.92 | 0.6301 | 0.9200 | 0.7480 |
| EM=5 | 67.00% | 92 | 58 | 42 | 8 | 0.58 | 0.92 | 0.6133 | 0.9200 | 0.7360 |
| EM=7 | 68.50% | 89 | 52 | 48 | 11 | 0.52 | 0.89 | 0.6312 | 0.8900 | 0.7386 |
| EM=9 | 69.00% | 91 | 53 | 47 | 9 | 0.53 | 0.91 | 0.6319 | 0.9100 | 0.7459 |
| NN | 53.00% | 49 | 43 | 57 | 51 | 0.43 | 0.49 | 0.5326 | 0.4900 | 0.5104 |

Table 69: wav2vec (Ge) binary classifiers' metrics, using a threshold of maximum 1% FPR (if available) and a minimum 50% TPR, with 100 benign data and 100 genetic AEs.

| Classifier | Accuracy | TP | FP | TN | FN | FPR | TPR | Precision | Recall | F1 |
|---|---|---|---|---|---|---|---|---|---|---|
| NF | 48.00% | 80 | 84 | 16 | 20 | 0.84 | 0.80 | 0.4878 | 0.8000 | 0.6061 |
| TD | 54.50% | 96 | 87 | 13 | 4 | 0.87 | 0.96 | 0.5246 | 0.9600 | 0.6784 |
| GC | 52.00% | 98 | 94 | 6 | 2 | 0.94 | 0.98 | 0.5104 | 0.9800 | 0.6712 |
| EM=3 | 51.50% | 92 | 89 | 11 | 8 | 0.89 | 0.92 | 0.5083 | 0.9200 | 0.6548 |
| EM=5 | 51.00% | 92 | 90 | 10 | 8 | 0.90 | 0.92 | 0.5055 | 0.9200 | 0.6525 |
| EM=7 | 54.00% | 89 | 81 | 19 | 11 | 0.81 | 0.89 | 0.5235 | 0.8900 | 0.6593 |
| EM=9 | 52.50% | 91 | 86 | 14 | 9 | 0.86 | 0.91 | 0.5141 | 0.9100 | 0.6570 |
| NN | 62.00% | 49 | 25 | 75 | 51 | 0.25 | 0.49 | 0.6622 | 0.4900 | 0.5632 |

Table 70: wav2vec (Ge) binary classifiers' metrics, using a threshold of maximum 1% FPR (if available) and a minimum 50% TPR, with 100 benign data and 100 Kenansville AEs.

| Classifier | Accuracy | TP | FP | TN | FN | FPR | TPR | Precision | Recall | F1 |
|---|---|---|---|---|---|---|---|---|---|---|
| NF | 69.50% | 80 | 41 | 59 | 20 | 0.41 | 0.80 | 0.6612 | 0.8000 | 0.7240 |
| TD | 71.50% | 96 | 53 | 47 | 4 | 0.53 | 0.96 | 0.6443 | 0.9600 | 0.7711 |
| GC | 82.00% | 98 | 34 | 66 | 2 | 0.34 | 0.98 | 0.7424 | 0.9800 | 0.8448 |
| EM=3 | 77.00% | 92 | 38 | 62 | 8 | 0.38 | 0.92 | 0.7077 | 0.9200 | 0.8000 |
| EM=5 | 77.50% | 92 | 37 | 63 | 8 | 0.37 | 0.92 | 0.7132 | 0.9200 | 0.8035 |
| EM=7 | 76.50% | 89 | 36 | 64 | 11 | 0.36 | 0.89 | 0.7120 | 0.8900 | 0.7911 |
| EM=9 | 78.50% | 91 | 34 | 66 | 9 | 0.34 | 0.91 | 0.7280 | 0.9100 | 0.8089 |
| NN | 71.00% | 49 | 7 | 93 | 51 | 0.07 | 0.49 | 0.8750 | 0.4900 | 0.6282 |

**Trf (Ma) model performance**  Tab. 71: PGD AEs vs. benign data, Tab. 72: genetic AEs vs. benign data, Tab. 73: Kenansville AEs vs. benign data.

Table 71: Trf (Ma) binary classifiers' metrics, using a threshold of maximum 1% FPR (if available) and a minimum 50% TPR, with 100 benign data and 100 PGD AEs.

| Classifier | Accuracy | TP | FP | TN | FN | FPR | TPR | Precision | Recall | F1 |
|---|---|---|---|---|---|---|---|---|---|---|
| NF | 55.00% | 90 | 80 | 20 | 10 | 0.80 | 0.90 | 0.5294 | 0.9000 | 0.6667 |
| TD | 74.50% | 64 | 15 | 85 | 36 | 0.15 | 0.64 | 0.8101 | 0.6400 | 0.7151 |
| GC | 70.50% | 75 | 34 | 66 | 25 | 0.34 | 0.75 | 0.6881 | 0.7500 | 0.7177 |
| EM=3 | 90.50% | 81 | 0 | 100 | 19 | 0.00 | 0.81 | 1.0000 | 0.8100 | 0.8950 |
| EM=5 | 90.00% | 80 | 0 | 100 | 20 | 0.00 | 0.80 | 1.0000 | 0.8000 | 0.8889 |
| EM=7 | 90.50% | 81 | 0 | 100 | 19 | 0.00 | 0.81 | 1.0000 | 0.8100 | 0.8950 |
| EM=9 | 87.50% | 75 | 0 | 100 | 25 | 0.00 | 0.75 | 1.0000 | 0.7500 | 0.8571 |
| NN | 75.00% | 51 | 1 | 99 | 49 | 0.01 | 0.51 | 0.9808 | 0.5100 | 0.6711 |

Table 72: Trf (Ma) binary classifiers' metrics, using a threshold of maximum 1% FPR (if available) and a minimum 50% TPR, with 100 benign data and 100 genetic AEs.

| Classifier | Accuracy | TP | FP | TN | FN | FPR | TPR | Precision | Recall | F1 |
|---|---|---|---|---|---|---|---|---|---|---|
| NF | 46.00% | 90 | 98 | 2 | 10 | 0.98 | 0.90 | 0.4787 | 0.9000 | 0.6250 |
| TD | 56.50% | 64 | 51 | 49 | 36 | 0.51 | 0.64 | 0.5565 | 0.6400 | 0.5953 |
| GC | 63.50% | 75 | 48 | 52 | 25 | 0.48 | 0.75 | 0.6098 | 0.7500 | 0.6726 |
| EM=3 | 72.00% | 81 | 37 | 63 | 19 | 0.37 | 0.81 | 0.6864 | 0.8100 | 0.7431 |
| EM=5 | 72.00% | 80 | 36 | 64 | 20 | 0.36 | 0.80 | 0.6897 | 0.8000 | 0.7407 |
| EM=7 | 71.50% | 81 | 38 | 62 | 19 | 0.38 | 0.81 | 0.6807 | 0.8100 | 0.7397 |
| EM=9 | 68.50% | 75 | 38 | 62 | 25 | 0.38 | 0.75 | 0.6637 | 0.7500 | 0.7042 |
| NN | 71.00% | 49 | 7 | 93 | 51 | 0.07 | 0.49 | 0.8750 | 0.4900 | 0.6282 |

Table 73: Trf (Ma) binary classifiers' metrics, using a threshold of maximum 1% FPR (if available) and a minimum 50% TPR, with 100 benign data and 100 Kenansville AEs.

| Classifier | Accuracy | TP | FP | TN | FN | FPR | TPR | Precision | Recall | F1 |
|---|---|---|---|---|---|---|---|---|---|---|
| NF | 87.50% | 90 | 15 | 85 | 10 | 0.15 | 0.90 | 0.8571 | 0.9000 | 0.8780 |
| TD | 76.50% | 64 | 11 | 89 | 36 | 0.11 | 0.64 | 0.8533 | 0.6400 | 0.7314 |
| GC | 87.50% | 75 | 0 | 100 | 25 | 0.00 | 0.75 | 1.0000 | 0.7500 | 0.8571 |
| EM=3 | 90.50% | 81 | 0 | 100 | 19 | 0.00 | 0.81 | 1.0000 | 0.8100 | 0.8950 |
| EM=5 | 90.00% | 80 | 0 | 100 | 20 | 0.00 | 0.80 | 1.0000 | 0.8000 | 0.8889 |
| EM=7 | 90.50% | 81 | 0 | 100 | 19 | 0.00 | 0.81 | 1.0000 | 0.8100 | 0.8950 |
| EM=9 | 87.50% | 75 | 0 | 100 | 25 | 0.00 | 0.75 | 1.0000 | 0.7500 | 0.8571 |
| NN | 98.50% | 98 | 1 | 99 | 2 | 0.01 | 0.98 | 0.9899 | 0.9800 | 0.9849 |

**Trf (En) model performance** Tab. 74: PGD AEs vs. benign data, Tab. 75: genetic AEs vs. benign data, Tab. 76: Kenansville AEs vs. benign data.

Table 74: Trf (En) binary classifiers' metrics, using a threshold of maximum 1% FPR (if available) and a minimum 50% TPR, with 100 benign data and 100 PGD AEs.

| Classifier | Accuracy | TP | FP | TN | FN | FPR | TPR | Precision | Recall | F1 |
|---|---|---|---|---|---|---|---|---|---|---|
| NF | 47.50% | 91 | 96 | 4 | 9 | 0.96 | 0.91 | 0.4866 | 0.9100 | 0.6341 |
| TD | 61.50% | 96 | 73 | 27 | 4 | 0.73 | 0.96 | 0.5680 | 0.9600 | 0.7138 |
| GC | 50.00% | 100 | 100 | 0 | 0 | 1.00 | 1.00 | 0.5000 | 1.0000 | 0.6667 |
| EM=3 | 50.00% | 99 | 99 | 1 | 1 | 0.99 | 0.99 | 0.5000 | 0.9900 | 0.6644 |
| EM=5 | 50.50% | 100 | 99 | 1 | 0 | 0.99 | 1.00 | 0.5025 | 1.0000 | 0.6689 |
| EM=7 | 50.50% | 100 | 99 | 1 | 0 | 0.99 | 1.00 | 0.5025 | 1.0000 | 0.6689 |
| EM=9 | 50.50% | 100 | 99 | 1 | 0 | 0.99 | 1.00 | 0.5025 | 1.0000 | 0.6689 |
| NN | 53.00% | 49 | 43 | 57 | 51 | 0.43 | 0.49 | 0.5326 | 0.4900 | 0.5104 |

Table 75: Trf (En) binary classifiers' metrics, using a threshold of maximum 1% FPR (if available) and a minimum 50% TPR, with 100 benign data and 100 genetic AEs.

| Classifier | Accuracy | TP | FP | TN | FN | FPR | TPR | Precision | Recall | F1 |
|---|---|---|---|---|---|---|---|---|---|---|
| NF | 50.00% | 91 | 91 | 9 | 9 | 0.91 | 0.91 | 0.5000 | 0.9100 | 0.6454 |
| TD | 51.50% | 96 | 93 | 7 | 4 | 0.93 | 0.96 | 0.5079 | 0.9600 | 0.6644 |
| GC | 50.00% | 100 | 100 | 0 | 0 | 1.00 | 1.00 | 0.5000 | 1.0000 | 0.6667 |
| EM=3 | 51.00% | 99 | 97 | 3 | 1 | 0.97 | 0.99 | 0.5051 | 0.9900 | 0.6689 |
| EM=5 | 51.00% | 100 | 98 | 2 | 0 | 0.98 | 1.00 | 0.5051 | 1.0000 | 0.6711 |
| EM=7 | 51.00% | 100 | 98 | 2 | 0 | 0.98 | 1.00 | 0.5051 | 1.0000 | 0.6711 |
| EM=9 | 51.00% | 100 | 98 | 2 | 0 | 0.98 | 1.00 | 0.5051 | 1.0000 | 0.6711 |
| NN | 56.50% | 49 | 36 | 64 | 51 | 0.36 | 0.49 | 0.5765 | 0.4900 | 0.5297 |

Table 76: Trf (En) binary classifiers' metrics, using a threshold of maximum 1% FPR (if available) and a minimum 50% TPR, with 100 benign data and 100 Kenansville AEs.

| Classifier | Accuracy | TP | FP | TN | FN | FPR | TPR | Precision | Recall | F1 |
|---|---|---|---|---|---|---|---|---|---|---|
| NF | 68.50% | 91 | 54 | 46 | 9 | 0.54 | 0.91 | 0.6276 | 0.9100 | 0.7429 |
| TD | 68.50% | 96 | 59 | 41 | 4 | 0.59 | 0.96 | 0.6194 | 0.9600 | 0.7529 |
| GC | 73.50% | 100 | 53 | 47 | 0 | 0.53 | 1.00 | 0.6536 | 1.0000 | 0.7905 |
| EM=3 | 77.00% | 99 | 45 | 55 | 1 | 0.45 | 0.99 | 0.6875 | 0.9900 | 0.8115 |
| EM=5 | 76.50% | 100 | 47 | 53 | 0 | 0.47 | 1.00 | 0.6803 | 1.0000 | 0.8097 |
| EM=7 | 76.00% | 100 | 48 | 52 | 0 | 0.48 | 1.00 | 0.6757 | 1.0000 | 0.8065 |
| EM=9 | 77.00% | 100 | 46 | 54 | 0 | 0.46 | 1.00 | 0.6849 | 1.0000 | 0.8130 |
| NN | 73.00% | 49 | 3 | 97 | 51 | 0.03 | 0.49 | 0.9423 | 0.4900 | 0.6447 |

## J  WORD SEQUENCE LENGTH IMPACT

In nearly all instances, we noticed no substantial decrease in performance, consistently maintaining an AUROC score exceeding 98% across all models, regardless of the word sequence length, supported by the results given in Table 77.

Table 77: Comparing AUROC scores across various word sequence lengths using a combined dataset of 100 benign samples and 100 C&W AEs.

| Model \ # of words | 2 | 3 | 4 | 5 | 6 | 7 | 8 | 9 | 10+ |
|---|---|---|---|---|---|---|---|---|---|
| LSTM (It) | - | 1.000 | 1.000 | 1.000 | 1.000 | 1.000 | 1.000 | 1.000 | 0.994 |
| LSTM (En) | 1.000 | - | 1.000 | 1.000 | 1.000 | 1.000 | 1.000 | 1.000 | 0.998 |
| LSTM (En-LM) | 1.000 | | 0.917 | 1.000 | 1.000 | 1.000 | 1.000 | 1.000 | 0.995 |
| wav2vec (Ma) | | - | 1.000 | 1.000 | 0.986 | 1.000 | 0.991 | 1.000 | 1.000 |
| wav2vec (Ge) | - | | 1.000 | 1.000 | 1.000 | 0.992 | 1.000 | 1.000 | 1.000 |
| Trf (Ma) | - | - | 1.000 | 1.000 | 1.000 | 1.000 | 1.000 | 1.000 | 1.000 |
| Trf (En) | 1.000 | - | 1.000 | 1.000 | 1.000 | 1.000 | 1.000 | 1.000 | 1.000 |

## K  TRANSFER ATTACK

To assess the transferability of targeted adversarial attacks between models, we tested whether the effectiveness of these attacks, specifically tailored to one model, remains consistent when tested on another ASR system. The results indicate a lack of transferability, as the WER in all cases is much closer to 100% than the expected 0%, as depicted in Tab. 78.

Table 78: WER performance of transfer attacks on chosen pairs of source and target models with 100 samples each from C&W AEs, and Psychoacoustic AEs.

| Source Model | Target model | C&W attack | Psychoacoustic attack |
|---|---|---|---|
| wav2vec (Ma) | Trf (Ma) | 99.33% | 99.24% |
| Trf (Ma) | wav2vec (Ma) | 99.41% | 99.41% |
| LSTM (En) | LSTM (En-LM) | 103.58% | 103.28% |
| LSTM (En) | Trf (En) | 104.48% | 104.58% |
| LSTM (En-LM) | LSTM (En) | 104.98% | 104.68% |
| LSTM (En-LM) | Trf (En) | 104.68% | 104.68% |
| Trf (En) | LSTM (En) | 106.17% | 105.37% |
| Trf (En) | LSTM (En-LM) | 104.68% | 104.78% |