# OpenReview forum: "DistriBlock: Identifying adversarial audio samples by leveraging characteristics of the output distribution"
_auai.org/UAI/2024/Conference — UAI 2024 poster_

### Official Review · Reviewer_eFx7 · 2024-03-20

**Q2-1 Originality-Novelty:** 4
**Q2-2 Correctness-Technical Quality:** 3
**Q2-5 Clarity Of Writing:** 3

**Q1 Summary And Contributions:**

Deep neural networks that perform automated speech recognition are known to be  highly vulnerable to adversarial attacks.  This paper describes a  strategy to detect such attacks by predicting the Evolution of the probability distribution over a string of output tokens.  Extensive empirical analysis across diverse ASR models, attack types, and datasets covering a range of languages.  Finally, an adaptive attack is designed to counteract the strategy described,  demṕiricallty showing it results in an output of high level noise that is easy for humans to spot.

**Q2-3 Extent To Which Claims Are Supported By Evidence:**

4: Excellent: all claims are supported by very convincing evidence (in the form of comprehensive experimental evaluation, rigorous mathematical proofs, detailed (pseudo-)code, precise references, well-motivated and realistic assumptions) and the authors deliver what they promise.

**Q2-4 Reproducibility:**

3: Good: key resources (e.g. proofs, code, data) are available and key details (e.g. proofs, experimental setup) are sufficiently well-described for competent researchers to confidently reproduce the main results.

**Q3 Main Strengths:**

Authors provide a  simple and elegant strategy  to detect attacks to ASR  models. The mathematical basis for the method is simply and clearly exposed,  and extensive tests support claims on the quality of the strategy. It is also demonstrated that an adversarial attack that takes into consideration the existence of the provided strategy  generates a  very noisy output, that would be easy for humans to detect.

**Q4 Main Weakness:**

There are some writing problems,  such as not mentioning NF and TD  before experiments  results are shown.

**Q5 Detailed Comments To The Authors:**

Simple fixes of writing problems mentioned above would make the paper more readable.

**Q9 Complying With Reviewing Instructions:**

Yes

---

> ### Author Rebuttal · Authors · 2024-04-08
>
> Many thanks for your feedback! We updated our paper accordingly.
>
> **>> There are some writing problems, such as not mentioning NF and TD before experiment results are shown. Simple fixes of writing problems mentioned above would make the paper more readable. <<**
>
> Thanks for this suggestion. Indeed, we do briefly discuss NF and TD in section 2 (page 2). To make this more clear, we will explicitly state in section 4 that NF and TD serve as baseline detectors for our comparison.

---

### Official Review · Reviewer_DZUB · 2024-03-22

**Q2-1 Originality-Novelty:** 2
**Q2-2 Correctness-Technical Quality:** 3
**Q2-5 Clarity Of Writing:** 4

**Q1 Summary And Contributions:**

This paper is proposing an efficient method for detecting adversarial attacks against automatic speech recognition (ASR) systems. The authors note that the noise added to speech signals in adversarial examples is likely to increase frame-to-frame variation in the distribution of tokens predicted by an ASR system. To this end, the proposed detector measures the divergence between the distributions of predictions for subsequent time steps, as well as summary statistics like median, minimum, and maximum likelihoods. A binary classifier is then used to predict whether the example is clean or adversarial, using either the sequence of divergences or summary statistics as input. The authors also develop an adaptive attack to circumvent their distinguisher, which produces adversarial examples which minimize the impact on the binary classifier of DistriBlock.

**Q2-3 Extent To Which Claims Are Supported By Evidence:**

3: Good: the main claims are supported by convincing evidence (in the form of adequate experimental evaluation, proofs, (pseudo-)code, references, assumptions).

**Q2-4 Reproducibility:**

3: Good: key resources (e.g. proofs, code, data) are available and key details (e.g. proofs, experimental setup) are sufficiently well-described for competent researchers to confidently reproduce the main results.

**Q3 Main Strengths:**

- The intuition for this detection method is easy to understand, and the paper explains why adversarial examples would be caught by this method in an understandable way.
- The evaluation section is particularly robust. Three ASR systems are tested which utilized different neural architectures, including transformers. Multilingual datasets are used for evaluation, reducing the risk of bias in the results driven by the use of a single language. The datasets also included other sources of noise and imperfect recording environments, helping to justify that DistriBlock is not just detecting noise. The defense is also tested against a new adaptive attack which is specifically designed to more closely mimic the relevant properties of the entropy of real speech.
- Analyzing the output distributions of ASR systems in this way may be useful in applications other than ML attacks/defenses, such as speech quality an intelligibility, as noted in the conclusion.
- The proposed detector has minimal computational overhead, as noted in appendix C.

**Q4 Main Weakness:**

- The drawbacks of previous approaches to AE detection (other than limited evaluations) are not made apparent in the related work section.
- Little guidance is provided on how to choose the best characteristics for use as features in the Gaussian classifier. The authors choose by testing the performance of all pairs on a collection of 100 attacked samples. This approach may become intractable if the user is considering a larger set of possible characteristic based on their own analysis of the input distribution.
- A major limitation of this defense is the lack of a formal guarantee of security. While it appears to be quite effective against existing attacks (and the adaptive attack described in the paper), given the rapid progress in adversarial examples it seems likely that new attacks will be developed to circumvent this defense.

**Q5 Detailed Comments To The Authors:**

- Based on appendices, there is are significant discrepancies in performance depending on the characteristic used for the Gaussian classifier. Are there any qualities of the dataset used that have an impact on the relative performance of each characteristic? Is there any dependency on language/accent?
- Is the filtering mechanism described in the "Detecting adaptive adversarial attacks" subsection robust to other sources of noise? I would hypothesize that a noisy recording environment might lead to similar changes in WER and CER after noise filtering is applied.
- What would you say are the biggest limitations of your approach to detecting adversarial samples?

**Q9 Complying With Reviewing Instructions:**

Yes

---

> ### Author Rebuttal · Authors · 2024-04-08
>
> Many thanks for your valuable feedback. We performed further investigations based on your comments and will update our paper accordingly.
>
> **>> Drawbacks of previous approaches <<**
>
> Temporal dependency has the drawback of requiring the audio to be at least of a certain length to show a reasonable performance, and it can be circumvented with adaptive attacks (as stated in the last paragraph of Section 2). A drawback of Noise Flooding we missed mentioning in the related work section is that it needs to call the ARS system several times to do prediction and is thus very time-consuming. We are happy to add this.
>
> **>> Little guidance on how to pick the best characteristic for the Gaussian classifier <<**
>
> We found that some characteristics (i.e. mean entropy and mean median) perform very well for all attack types and models. Therefore, these characteristics can be used out of the box without any further analysis.
>
> **>> Based on the appendices, there are significant discrepancies in performance depending on the characteristic used for the Gaussian classifier. Are there any qualities of the dataset used that have an impact on the relative performance of each characteristic? Is there any dependency on language/accent?<<**
>
> We did not find any pattern in the sense that some characteristics work well either for the same models independent from the dataset or for the same datasets independent from the employed model.
>
> **>> Is the filtering mechanism described in the "Detecting adaptive adversarial attacks" subsection robust to other sources of noise? I would hypothesize that a noisy recording environment might lead to similar changes in WER and CER after noise filtering is applied. <<**
>
> Thanks for the intriguing question. To test the performance on noisy data, we employed the same classifiers to distinguish (noisy) benign data from adaptive adversarial examples (AEs), as detailed in Table 7 (page 8) of the paper. This evaluation involved using 100 noisy data and 100 adaptive C&W AEs. The noisy data used is identical to that we use for testing ASR robustness by introducing environmental noise, described in section 5.2 (page 5). We compared the predicted transcription of an input signal with the transcription of its filtered version using metrics like WER and CER. The table below shows the resulting accuracy. Notably, the performance slightly decreases with noisy data compared to distinguishing benign data from adaptive AEs. However, the adaptive adversarial examples can still be detected with an average accuracy of about 95%. Results reported accuracy based on WER% - CER% values are described in the table below.
>
> | Model | GC | EM=3 | EM=7 | EM=9 | NN |
> | :----- | :----- | :----- | :----- | :----- | :----- |
> | LSTM (It) | 89.5% / 90.5% | 87.0% / 88.5% | 84.0% / 84.0% | 82.5% / 82.0% | 93.0% / 92.5% |
> | LSTM (En) | 96.5% / 95.0% | 96.5% / 95.0% | 95.5% / 96.0% | 95.5% / 95.5% | 98.0% / 98.5% |
> | LSTM (En-LM)  | 97.0% / 98.5% | 95.5% / 97.5% | 90.0% / 98.0% | 90.5% / 96.5% | 97.5% / 98.5% |
> | wav2vec (Ma) | 91.5% / 91.5% | 92.5% / 92.5%  | 89.5% / 89.5% | 92.0% / 92.0% | 93.5% / 93.5% |
> | wav2vec (Ge) | 93.5% / 92.5% | 94.0% / 95.0% | 90.0% / 93.0% | 95.5% / 98.0% | 96.5% / 98.0% |
> | Trf (Ma)  | 73.0% / 73.0% | 76.0% / 76.0% | 71.5% / 71.5% | 69.0% / 69.0% | 87.0% / 87.0% |
> | Trf (En) | 92.0% / 96.0% | 90.0% / 94.5% | 81.0% / 85.5% | 75.0% / 82.0% | 97.0% / 97.5% |
> | Avg. AUROC | 90.4% / 91.0% | 90.2% / 91.3% | 85.9% / 88.2% | 85.7% / 87.9% | **94.6% / 95.1%** |
>
> **>> What would you say are the biggest limitations of your approach to detecting adversarial samples? <<**
>
> We agree with the reviewer that the biggest limitation is that there is no formal guarantee that the detection can not be circumvented by novel attacks, as is the case for all existing defense strategies so far. But at least the straightforward adaptive attacks were not successful without inducing a lot of noise.

---

### Official Review · Reviewer_HLQe · 2024-03-23

**Q2-1 Originality-Novelty:** 3
**Q2-2 Correctness-Technical Quality:** 3
**Q2-5 Clarity Of Writing:** 4

**Q10 Ethical Concerns:**

None.

**Q1 Summary And Contributions:**

This paper presents DistriBlock, a novel method for detecting adversarial attacks targeting automatic speech recognition (ASR) systems. The method is based on comparing the probability distributions over output tokens in subsequent time steps. The assumption is that an adversarial attack will lead to higher uncertainty in the outputs of the ASR system whereas in cases of benign speech there should be high similarity between distributions of subsequent time steps. The authors use features from these distributions to build a binary classifier to distinguish between benign ASR outputs and ASR outputs resulting from adversarial attacks. The effectiveness of the method is illustrated with many experiments using ASR systems for different languages and employing common adversarial attacks (targeted, untargeted, adaptive). The proposed method also outperforms strong baselines (in most cases).

**Q2-3 Extent To Which Claims Are Supported By Evidence:**

4: Excellent: all claims are supported by very convincing evidence (in the form of comprehensive experimental evaluation, rigorous mathematical proofs, detailed (pseudo-)code, precise references, well-motivated and realistic assumptions) and the authors deliver what they promise.

**Q2-4 Reproducibility:**

3: Good: key resources (e.g. proofs, code, data) are available and key details (e.g. proofs, experimental setup) are sufficiently well-described for competent researchers to confidently reproduce the main results.

**Q3 Main Strengths:**

- Very interesting and timely topic.
- Very well written and clearly written paper especially given the complexity of the task and the plethora of experiments.
- The assumption that in benign ASR outputs there should be high similarity between distributions of subsequent time steps is intuitive but it's not trivial to use this information efficiently to build a robust classifier to identify ASR outputs of benign speech versus ASR outputs of speech resulting from adversarial attacks. It's also not trivial to build a method that performs well for different types of targeted and untargeted adversarial attacks.
- Different datasets, languages, and ASR systems are considered.
- There is thorough evaluation against strong baselines and using different types of adversarial attacks (targeted and untargeted). There are also experiments with adaptive attacks aiming to generate ASR output distributions that are very similar to the ASR output distributions from benign data. In these cases if the proposed method fails to detect the attack the resulting speech ends up being very noisy which would make it easier to detect through filtering.
- The paper provides a very detailed appendix with additional information and there is also a link to an anonymized GitHub repository with examples of benign, adversarial, and noisy data used in the experiments.

**Q4 Main Weakness:**

Even though the paper is very well written there are some points that would benefit from more information and elaboration (see detailed comments to the authors).

**Q5 Detailed Comments To The Authors:**

- It would be useful to have more information about the ASR systems, for example whether they output characters, phonemes, etc. and what kind of language models were used for having words as outputs. Why wasn't OpenAI Whisper used?
- In the "adversarial example detectors" section, 24 scores are mentioned resulting from the combination of 4 aggregation methods with 6 characteristics. The 6 characteristics for each time step (or pairs of subsequent steps) are the Shannon entropy, the Kullback-Leibler divergence, the Jensen-Shannon divergence, the median, the minimum, and the maximum. Then the 4 aggregation methods are the mean, median, maximum, and minimum of the 6 characteristics in order to get the score for the whole utterance. Why isn't the mean used as part of the characteristics?
- 200 samples were selected from the test data (for each dataset) but then the paper says "to create adaptive attacks, we started from 100 adversarial examples we already created". It would be useful to provide information about how the 200 samples were used. Were the same samples used to test all the different types of adversarial attacks?
- Please provide more information about how the Gaussian classifiers and the ensemble model are constructed.

**Q9 Complying With Reviewing Instructions:**

Yes

---

> ### Author Rebuttal · Authors · 2024-04-08
>
> We value your insightful comments and queries. To address your concerns, we provide explanations in the following.
>
> **>> More information about the ASR systems, for example, whether they output characters, phonemes, etc. and what kind of language models were used for having words as outputs <<**
>
> We analyze a variety of ASR systems. These models generate diverse output formats depending on their language and tokenizer selection, which can encode either characters or subwords. Specifically, the models we use produce output structures with neuron counts of 32, 500, 1000, 5000, or 21128. In Appendix E, table 16 (page 17) we provide details regarding the specific tokenizer type employed by each ASR model. We will update this table to contain additional information about the kind of language model, the number of token units, and the tokenizer type as shown below.
> Moreover, we will provide the links to the detailed descriptions of each ASR system in the SpeechBrain repository.
>
> | Model-Language | Pre-trained model | Subword-base Tokenizer | # Of tokenizer units |
> | :----- | :----- | :----- | :----- |
> | LSTM-Italian¹  | No    | Unigram that transforms words into subword units | 500 |
> | LSTM-English²  | No    | Unigram that transforms words into subword units | 1000 |
> | LSTM-English  | Yes: pre-trained Recurrent Neural Network Language Model⁷  | Unigram that transforms words into subword units | 5000 |
> | wav2vec-Mandarin³ | Yes: pre-trained large Fairseq Chinese wav2vec2 model⁸  | WordPiece (BERT from huggingface) that transforms words into chars  | 21128 |
> | wav2vec-German⁴  | Yes: pre-trained large Fairseq German wav2vec2 model⁹   | Tokenizer (char) that transforms words into chars | 32 |
> | Trf-Mandarin⁵  | No    | Unigram that transforms words into subword units | 5000 |
> | Trf-English⁶ | Yes: pre-trained Transformer Language Model¹⁰   | Unigram that transforms words into subword units | 5000 |
>
> **>> Why wasn't OpenAI Whisper used? <<**
>
> There is no specific reason that we did not take this model into account. Due to limited resources, we prioritized models that had been trained in various languages previously. Our approach is easily applicable to Whisper as well.
>
> **>> Why isn't the mean used as part of the characteristics? <<**
>
> The mean in the sense of the average of the probabilities is the same for all kinds of distributions, since the probabilities in each time step sum to one. This is why we did not take the mean into account.
>
> **>> 200 samples were selected from the test data (for each dataset) but then the paper says "to create adaptive attacks, we started from 100 adversarial examples we already created". It would be useful to provide information about how the 200 samples were used. Were the same samples used to test all the different types of adversarial attacks? <<**
>
> We randomly selected 200 samples from the test set for each dataset.  For all, we generated adversarial examples for each adversarial attack type and ASR model. Then, all methods were tested on the task of distinguishing the first 100 test examples from their corresponding adversarial examples.  So the same benign examples were used to test all different types of attacks.
> The additional 100 test examples and corresponding adversarial examples were used as a validation set for picking the best characteristic for the Gaussian classifiers for each model.
> In the case of adaptive attacks, a new set of 100 adversarial examples was generated. These adaptive attack instances were generated from inputs that had already proven to mislead the system in previous studies, i.e. the 100 adversarial examples used for testing.
> We make this clearer in the revised version of the paper.
>
> **>> Please provide more information about how the Gaussian classifiers and the ensemble model are constructed. <<**
>
> The extracted characteristics of the output distribution are used as features for different binary classifiers.
> We obtain simple Gaussian classifiers by fitting a Gaussian distribution to each score computed for the utterances from a held-out set of 100 benign data for each ASR system and characteristic. A new observation is then classified as an attack if the value of the prediction has a probability lower than a chosen threshold under the Gaussian model. The threshold can be chosen on the benign examples as a value that guarantees a small number of false positives.
> For the ensemble models we combine several of the above described Gaussian classifiers. That is, we take the decision of multiple Gaussian classifiers, that are based on different characteristics, simultaneously into account.  The final decision is then given by a majority vote (i.e., the classification output that receives more than half of the votes). We consider ensemble models built on a total of 3, 5, 7, or 9 Gaussian classifiers that perform best on the validation set.
> We measure the ROC of these classifiers for each model type and characteristic.

---

### Official Review · Reviewer_paiX · 2024-03-23

**Q2-1 Originality-Novelty:** 2
**Q2-2 Correctness-Technical Quality:** 3
**Q2-5 Clarity Of Writing:** 3

**Q1 Summary And Contributions:**

The paper introduces "DistriBlock," to enhance the security of Automatic Speech Recognition (ASR) systems against adversarial attacks. The authors propose to use the probability distribution of output tokens at each time step, including metrics like median, maximum, and minimum probabilities, entropy, and divergence measures as features for binary classifiers to distinguish between benign and malicious inputs.

**Q2-3 Extent To Which Claims Are Supported By Evidence:**

3: Good: the main claims are supported by convincing evidence (in the form of adequate experimental evaluation, proofs, (pseudo-)code, references, assumptions).

**Q2-4 Reproducibility:**

2: Fair: key resources (e.g. proofs, code, data) are unavailable but key details (e.g. proof sketches, experimental setup) are sufficiently well-described for an expert to confidently reproduce the main results.

**Q3 Main Strengths:**

+ proposes a defense against adversarial noise for ASR
+ The paper is well written
+ Considers adaptive attacker

**Q4 Main Weakness:**

- While analysing statistical/entropy propoerties might be novel when considering the ASR, these ideas have been investigated in the vision community, especially against adversarial patches

**Q5 Detailed Comments To The Authors:**

The paper introduces "DistriBlock," to enhance the security of Automatic Speech Recognition (ASR) systems against adversarial attacks. The authors propose to use the probability distribution of output tokens at each time step, including metrics like median, maximum, and minimum probabilities, entropy, and divergence measures as features for binary classifiers to distinguish between benign and malicious inputs.

The paper is well written and reads well. I also think that there is some novelty strictly from ASR perspective. I also like that the authors consider adaptive attacks, which is necessary from a methodological perspective. However, I think the novelty might be limited to this specific case, and that there not much surprise in terms of state-of-knowledge about adversarial attacks properties (and defenses against them). In fact, while analysing statistical/entropy propoerties might be novel when considering the ASR, these ideas have been investigated in the vision community, especially against adversarial patches both in the input space and in the latent space, such as :
[A]  Tarchoun et. al. ;" Jedi: Entropy-Based Localization and Removal of Adversarial Patches" Proceedings of the IEEE/CVF Conference on Computer Vision and Pattern Recognition (CVPR), 2023, pp. 4087-4095
[B] Naseer et. al. "gradients smoothing: Defense against localized adversarial attacks". In 2019 IEEE Winter Conference on Applications of
Computer Vision (WACV), pages 1300–1307, 2019.
[C] Jamie Hayes. "On visible adversarial perturbations & digital watermarking". In Proceedings of the IEEE Conference
on Computer Vision and Pattern Recognition (CVPR) Workshops, June 2018

**Q9 Complying With Reviewing Instructions:**

Yes

---

> ### Author Rebuttal · Authors · 2024-04-08
>
> Thanks for your valuable feedback. We appreciate your suggestions and will update our paper accordingly.
>
> **>> Highlight more the novelty, especially considering the state-of-the-art in the vision community. <<**
>
> Thanks for pointing out work from the computer vision community that also leverages entropy/statistics for defenses, specifically for the detection and removal of adversarial patches. We are happy to incorporate it in our related work section. Note, however, that only [A] utilizes entropy, while [B] and [C] are pre-processing methods that transform the input image before inference. Moreover, [A] employs entropy in a very different manner: they analyze the entropy of the input (i.e. of pixels in a certain window), [B] projects a scaled normalized gradient magnitude map onto the image to directly suppress high activation regions and then changing the input, [C] applies the Fast Marching method¹ to remove input pixel information, whereas we focus on analyzing the entropy/statistics of the output distribution. Furthermore, they use it to identify adversarial patches that are perceptible for humans, while we focus on the detection of imperceptible adversarial audio examples.
>
> ¹ A. Telea. An image inpainting technique based on the fast marching method. Journal of graphics tools, 9(1):23–34, 2004.

---

### Meta-Review · Area_Chair_t4bM · 2024-04-12

The paper proposes a method for identifying adversarial audio samples that can mislead automatic speech recognition (ASR) systems. They build a new representation of benign and corrupted audio samples with descriptive statistics of the distribution of output tokens issued by the ASR system.  They then train a classifier to discriminate between them.

Despite its simplicity, the procedure is shown to be effective through a series of experiments. There are no formal guarantees, but the evaluation is shown to be robust even in an informed case, i.e. when the attacker knows the defence and tries to minimise a loss that integrates the features of the defender's discriminative model.

There is a consensus to accept the paper.